# Accelerating Inference for Multilayer Neural Networks with Quantum Computers

**Arthur G. Rattew[1,3,*], Po-Wei Huang[2,3], Naixu Guo[4], Lirandë Pira[4], Patrick Rebentrost[4,5]**

[1] Department of Materials, University of Oxford, Oxford OX1 3PH, United Kingdom
[2] Mathematical Institute, University of Oxford, Oxford OX2 6GG, United Kingdom
[3] Quantum Motion, 9 Sterling Way, London N7 9HJ, United Kingdom
[4] Centre for Quantum Technologies, National University of Singapore, Singapore 117543
[5] Department of Computer Science, National University of Singapore, Singapore 117417

## Abstract

Fault-tolerant Quantum Processing Units (QPUs) promise to deliver exponential speed-ups in select computational tasks, yet their integration into modern deep learning pipelines remains unclear. In this work, we take a step towards bridging this gap by presenting the first fully-coherent quantum implementation of a multi-layer neural network with non-linear activation functions. Our constructions mirror widely used deep learning architectures based on ResNet, and consist of residual blocks with multi-filter 2D convolutions, sigmoid activations, skip-connections, and layer normalizations. We analyse the complexity of inference for networks under three quantum data access regimes. Without any assumptions, we establish a quadratic speedup over classical methods for shallow bilinear-style networks. With efficient quantum access to the weights, we obtain a quartic speedup over classical methods. With efficient quantum access to both the inputs and the network weights, we prove that a network with an $N$-dimensional vectorized input, $k$ residual block layers, and a final residual-linear-pooling layer can be implemented with an error of $\epsilon$ with $O(\text{polylog}(N/\epsilon)^k)$ inference cost.

## 1 Introduction

Within the past decade, deep learning methods (LeCun et al., 2015; Goodfellow et al., 2016) have become the mainstream methodology for tackling problems in machine learning and generative artificial intelligence, including tasks in computer vision (He et al., 2016; Ho et al., 2020; Dosovitskiy et al., 2021), natural language processing (Vaswani et al., 2017; Brown et al., 2020) and various other tasks with increasing applicability (Silver et al., 2016; Jumper et al., 2021; Fawzi et al., 2022). This progress is partly facilitated by advances in GPUs, which offer speed-ups for parallelizable operations such as matrix-vector arithmetic. However, as we approach the physical limits of Moore's law (Moore, 1965), the continuous upscaling of CPUs and GPUs may begin to plateau. Consequently, a natural question is whether quantum computing (Feynman, 1982; 1986; Nielsen & Chuang, 2010) and potential quantum processing units (QPUs) can offer further acceleration for deep learning.

The field of quantum machine learning (QML) (Biamonte et al., 2016; Schuld & Petruccione, 2021; Du et al., 2025), investigates this possibility. QML can broadly be separated into two main paradigms: (1) quantum algorithms tailored to the structure of near-term quantum hardware (Preskill, 2018) under assumptions of limited quantum resources, and (2) using quantum subroutines to obtain provable speed-ups for existing machine learning models, typically requiring large amounts of quantum resources necessitating error-corrected fault-tolerant quantum computers.

In the first paradigm, proposals of quantum neural networks (QNN) based on variational quantum algorithms (VQA) (Peruzzo et al., 2014; Cerezo et al., 2021) train parametrized quantum circuits (PQC) (Benedetti et al., 2019b) analogously to multi-layer neural networks. However, these algorithms face trainability issues in the form of poor local minima (Bittel & Kliesch, 2021; Anschuetz & Kiani, 2022) and vanishing gradients, or *barren plateaus* (McClean et al., 2018; Larocca et al., 2025).

---

*arthur.rattew.science@gmail.com

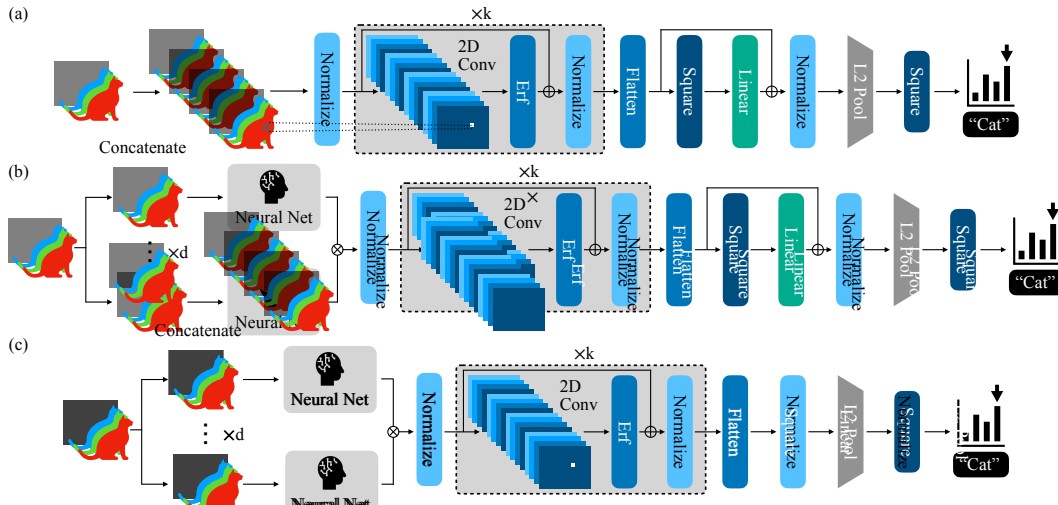

Figure 1: **Architecture for Convolutional Neural Networks.** This figure shows the architectures we consider with provable quantum complexity guarantees for inference under three regimes of quantum data access assumptions. (a) Depicts the architecture where both the inputs and network weights are provided in an efficient quantum data structure. (b) Only the network weights are provided in an efficient quantum data structure. (c) No input assumptions are made. In all architectures, the input is assumed to be a rank-3 tensor (e.g., images with 4 channels).

Moreover, techniques mitigating these issues often result in the algorithms being classically simulable (Cerezo et al., 2025; Bermejo et al., 2024). While alternate approaches such as quantum kernel methods (Havlíček et al., 2019; Schuld & Killoran, 2019) and others have been proposed (Benedetti et al., 2019a; Huang & Rebentrost, 2024), they often face similar trainability issues (Thanasilp et al., 2024; Rudolph et al., 2024).

The second paradigm focuses on the use of quantum subroutines (Harrow et al., 2009; Montanaro, 2016; Gilyén et al., 2019; Dalzell et al., 2025b) to provide asymptotic speed-ups in the underlying linear algebra of classical machine learning models, e.g., in matrix inversion, matrix-vector arithmetic, and sampling. Applications include support vector machines (Rebentrost et al., 2014), regression (Wiebe et al., 2012), feedforward neural networks (Allcock et al., 2020), convolutional neural networks (Kerenidis et al., 2020), transformers (Guo et al., 2024b), and other models (Lloyd et al., 2014; Wiebe et al., 2016; Rebentrost et al., 2018; Kapoor et al., 2016; Cherrat et al., 2024; Liu et al., 2021b; Yang et al., 2023; Ivashkov et al., 2024; Wang et al., 2025). Other works have also explored speeding up classical neural network training and inference (Kerenidis & Prakash, 2020; Abbas et al., 2023; Liu et al., 2024).

**Main Contributions.** In this paper, we propose a method that can be used to accelerate inference for multilayer residual networks (ResNets) (He et al., 2016) on quantum computers, given their significance in enabling deep networks (Xie et al., 2017; Dong et al., 2021). We provide core quantum subroutines and techniques for regularized multi-filter 2D convolutions, sigmoid activations, skip-connections, and layer normalizations – all of which we show can be coherently implemented on quantum computers. We list the main contributions as follows.

- In Section 2, we further develop a modular vector-encoding framework for quantum matrix-vector arithmetic. This is a special case of quantum block-encodings, with many useful properties.

- In Section 2.3, we derive a novel quantum algorithm for the multiplication of arbitrary full-rank and dense matrices with the element-wise square of a given vector, *without* incurring a rank-dependence. To the best of our knowledge, this is the first result which allows a quantum algorithm to utilize an arbitrary full-rank and dense matrix without a Frobenius norm complexity dependence.

- In Section 2.4, we provide a novel QRAM-free block-encoding for 2D multi-filter convolutions.

- In Section 4, to the best of our knowledge, we derive the *first coherent quantum implementations of multi-layer neural networks with non-linear activations*. We provide rigorous end-to-end complexity proofs for inference under three QRAM regimes:

  - **Regime 1 (inputs and weights provided via QRAM):** Assuming QRAM access to both inputs and weights, for a network with $k$ non-linear activations acting on $N$-dimensional inputs we prove $\tilde{O}(\text{polylog}(N/\epsilon)^k)$ inference cost. Moreover, we argue that existing techniques are insufficient to dequantize this result.

  - **Regime 2 (weights provided via QRAM):** When a cost linear in the dimension of the input must be paid (i.e., no QRAM for the input), but the network weights are stored in QRAM, we prove a quartic speedup over exact classical implementations for shallow architectures.

  - **Regime 3 (no QRAM):** In the absence of any QRAM, we prove a quadratic speedup over an exact classical implementation.

The relevant architectures in each regime can be seen in Figure 1. We derive a number of techniques and algorithms which have broad utility in implementing machine learning architectures on quantum computers. However, our main focus is on accelerating inference for classification, with our formal problem statement given in Definition 1. At a high-level, we assume that we are given a trained neural network which, given an input, outputs a probability distribution over possible outputs (e.g., over image classes). The goal is to draw a sample from this output distribution (thereby assigning a class to the input). We introduce an error parameter $\epsilon$, which allows the algorithm to sample from a distribution whose $\ell_2$ norm distance from the true distribution is bounded by at most $\epsilon$.

**Definition 1** (The Approximate Sampling-Based Classification Problem). *Let $0 \leq \epsilon \leq 1$. Given a neural network represented by function $h : \mathbb{R}^D \mapsto \mathbb{R}^C$ (i.e. with D-dimensional inputs and C-dimensional outputs) which returns a probability distribution as its output (i.e., for any $\boldsymbol{x} \in \mathbb{R}^D$, $\boldsymbol{y} := h(\boldsymbol{x})$ is all non-negative, and $\|\boldsymbol{y}\|_1 = 1$), then the sampling-based classification problem is to return a sample from some probability vector $\hat{\boldsymbol{y}}$ such that $\|\boldsymbol{y} - \hat{\boldsymbol{y}}\|_2 \leq \epsilon$.*

For example, in the case of CIFAR-10, $D = 3 \times 32 \times 32 = 3072$, and $C = 10$. Then, given some input $\boldsymbol{x} \in \mathbb{R}^{3072}$, $\boldsymbol{y} \in \mathbb{R}^{10}$ the entries of $\boldsymbol{y}$ correspond to the probability of assigning a given class (e.g., class $i$ is assigned with probability $y_i$, etc). This problem statement also naturally captures other applications, such as autoregressive next-token prediction, where the output distribution would instead be over the set of possible tokens rather than classes.

**Comparison to Prior Work.** In prior work, to achieve multi-layer architectures in feedforward and convolutional neural networks as well as transformers, intermediate measurements for inner products (Allcock et al., 2020) or quantum state tomography that read out the entire state (Kerenidis et al., 2020; Guo et al., 2024b) are required to extract information out to classical computers where data is required to be re-encoded into the quantum circuit for computation in the next layer, breaking the coherence of the quantum architecture and limiting potential speed-ups. We compare against the prior work in Table 1. To the best of our knowledge, *our work provides the first fully coherent quantum implementation of classical multi-layer neural networks* . Further, our work is also the first in works that accelerate classical deep learning algorithms to present an architecture which does not use QRAM. Moreover, we demonstrate that careful tracking on bounds of the vector norm (as it propagates through the forward-pass of a given network) is required to prevent arbitrary decay of the norm in multilayer structures, and subsequent unbounded runtimes. We provide rigorous proofs and develop tools to prove this norm preservation in our architectural blocks. Further, we make the observation that residual skip connections that enable deep networks classically are fundamental to the norm stability and preservation, enabling us to provide an efficient and coherent multilayer architecture not present in prior work.

**Notation.** We use standard big and small $O$ notations for asymptotics, using $\tilde{O}$ to hide polylogarithmic factors. The notation $[N]$ represents the set of integers $0, ..., N - 1$. We use kets to represent arbitrary (not necessarily normalized vectors). Logarithms are assumed to be base-2 unless otherwise stated. The subscript on the ket denotes the number of qubits it acts on (i.e., the log of the dimension), thus $|\psi\rangle_n \in \mathbb{C}^{2^n}$. When we assume a ket is normalized, we will explicitly state that it is. The one exception is with the definition of a vector-encoding (as defined subsequently in Definition 3). For example, an $(1, a, \epsilon)$-VE for $|\psi\rangle_n$ implicitly implies that $\||\psi\rangle_n\|_2 = 1$, and so we will not explicitly state the normalization of the encoded vector every time we introduce a VE. A bra is defined as

| | Architecture | Coherent Multi-Layer | Coherent Non-Linearity | QRAM-Free | Norm Preservation | Polylog $1/\epsilon$ | Polylog N |
|---|---|---|---|---|---|---|---|
| Cong et al. (2019)* | CNN Inspired PQC | ✗ | ✗ | ✓ | ✓ | N/A | N/A |
| Allcock et al. (2020) | Feed-forward | ✗ | ✗ | ✗ | ✗ | ✗ | ✗ |
| Kerenidis et al. (2020) | CNN | ✗ | ✗ | ✗ | ✗ | ✗ | ✗ |
| Guo et al. (2024b) | Transformer | ✗ | ✓ | ✗ | ✗ | ✗ | ✗ |
| Our work - Regime 1 | Residual CNN | ✓ | ✓ | ✗ | ✓ | ✓ | ✓ |
| Our work - Regime 2 | Bilinear Residual CNN | ✓ | ✓ | ✗ | ✓ | ✓ | ✗ |
| Our work - Regime 3 | Bilinear Residual CNN | ✓ | ✓ | ✓ | ✓ | ✓ | ✗ |

Table 1: **Comparison with prior work.** We briefly explain the meaning of each column. Coherent multi-layer refers to the construction of multi-layer architectures separated by non-linear activation functions without tomography. Coherent non-linearity refers to the implementation of non-linear transformations on the quantum computer without readout. Norm preservation refers to the preservation of vector norms throughout the network forward pass. Next, each quantum implementation of a classical architecture incurs some error over the exact classical implementation, and as such an entry ✓ in the polylog $1/\epsilon$ column indicates a $O(\text{polylog}(1/\epsilon))$ error-dependence, whilst a ✗ entry indicates a $O(\text{poly}(1/\epsilon))$ error-dependence. Finally, polylog $N$ refers to polylogarithmic complexity in the input dimension $N$. *Note: the architecture presented in Cong et al. (2019), is inspired by CNNs but is based on parameterized quantum circuits (PQC). As they do not aim to accelerate an existing classical architecture, it is not possible to provide an entry in the polylog $\epsilon$ column. Moreover, they do not provide complexities when considering classical input data, and so we do not give an entry in the column corresponding to polylog $N$.

the conjugate transpose of a ket, $\langle\psi|_n = |\psi\rangle_n^\dagger$. We use the notation $I_n$ to refer to an $n$-qubit (i.e., $2^n$-dimensional) identity matrix. We define the Kronecker product with the symbol $\otimes$, and will sometimes refer to this as a tensor product. We define basis functions both in vector notation and in ket notation, i.e., $|j\rangle \equiv \boldsymbol{e}_j$. E.g., $|0\rangle = \boldsymbol{e}_0 = (1 \quad 0 \quad \ldots \quad 0)^T$. When we define a function $f$ on scalars, i.e., $f : \mathbb{C} \mapsto \mathbb{C}$, given a vector $\boldsymbol{x} \in \mathbb{C}^N$ we sometimes use the notation $f(\boldsymbol{x}) := \sum_{j=0}^{N-1} f(x_j)\boldsymbol{e}_j$, i.e., $f(\boldsymbol{x})$ denotes an element-wise application of $f$ to $\boldsymbol{x}$.

## 2 QUANTUM MATRIX-VECTOR ARITHMETIC

In this section, we define and motivate the tools necessary to perform quantum matrix-vector arithmetic. These subroutines are essential for our subsequent results implementing classical neural networks on quantum computers. In Section 2.1, we provide a summary of quantum block-encodings and quantum vector encodings. **Novel contributions in this section:** In Section 2.2, we further develop the framework of vector-encodings, introducing straight-forward new quantum algorithms for vectors encoded as VEs, enabling vector sums, matrix-vector products, tensor products, and vector concatenations. In Section 2.3, we present a novel algorithm which applies an arbitrary full-rank and dense matrix to the element-wise square of a vector, without incurring a Frobenius norm dependence. Finally, in Section 2.4, we give a novel QRAM-free block-encoding for 2D multi-filter convolutions.

### 2.1 QUANTUM BLOCK-ENCODINGS AND VECTOR-ENCODINGS

A widely used tool in quantum algorithm design is the block-encoding (Gilyén et al., 2019), which can be viewed as a way to encode and manipulate matrices in quantum algorithms. A block-encoding is a unitary matrix $U$, specified by a quantum circuit, whose top left block contains a matrix $\tilde{A}$ (such that $\|\tilde{A}\|_2 \leq 1$) which is a scaled approximation to some matrix $A$. We give the formal definition in the following.

**Definition 2** (Block encoding (Gilyén et al., 2019)). *Suppose that $A$ is a $2^s \times 2^s$ matrix, $\alpha, \epsilon \in \mathbb{R}_+$ and $a \in \mathbb{N}$, then we say that the $2^{s+a} \times 2^{s+a}$ unitary matrix $U$ is an $(\alpha, a, \epsilon)$-block-encoding of $A$, if*

$$\|A - \alpha(\langle 0|^{\otimes a} \otimes I)U(|0\rangle^{\otimes a} \otimes I)\| \leq \epsilon. \tag{1}$$

Essentially, noting that $\langle 0|^{\otimes a} \otimes I = (I \quad 0 \quad \dots \quad 0)$, we see that $\langle 0|^{\otimes a} \otimes I$ selects the first $2^s$ rows of $U$, and then $|0\rangle^{\otimes a} \otimes I$ selects the first $2^s$ columns of $(\langle 0|^{\otimes a} \otimes I)U$, meaning that $(\langle 0|^{\otimes a} \otimes I)U(|0\rangle^{\otimes a} \otimes I)$ is simply the top-left $2^s \times 2^s$ block of $U$. Indeed, if $\epsilon = 0$, then $A/\alpha = (\langle 0|^{\otimes a} \otimes I)U(|0\rangle^{\otimes a} \otimes I)$. Additionally, $\alpha$ can be viewed as an upper-bound on the normalization factor of $A$, e.g., if $\epsilon = 0$, then $\|A/\alpha\|_2 \leq 1$. Any matrix encoded in a sub-block of a unitary matrix cannot have norm exceeding 1.

Analogously to how a quantum block-encoding encodes a general matrix in the top left block of a unitary, we can embed arbitrary (sub-normalized) $N$-dimensional vectors in the first $N$ rows of a larger vector corresponding to a normalized quantum state.

This naturally leads to the following definition of quantum vector-encodings (VEs), the definition of which we take nearly verbatim from Rattew & Rebentrost (2023), where they were called SPBEs.

**Definition 3** (Vector-Encoding (VE) (Rattew & Rebentrost, 2023)). *Let $\alpha \geq 1$, $a \in \mathbb{N}$, and $\epsilon \geq 0$. We call the $2^{a+n} \times 2^{a+n}$ unitary matrix $U_\psi$ an $(\alpha, a, \epsilon)-$VE for the $2^n$-dimensional quantum state $|\psi\rangle_n$, if*

$$\||\psi\rangle_n - \alpha(\langle 0|_a \otimes I_n) U_\psi |0\rangle_{a+n}\|_2 \leq \epsilon. \tag{2}$$

Note that $(\langle 0|_a \otimes I_n) U_\psi |0\rangle_{a+n}$ corresponds to the exact vector encoded by $U_\psi$, specifically encoded in the first $2^n$ rows of the first column of $U_\psi$. The parameter $\alpha$ is a measure of the norm of the encoded vector, e.g., if $\epsilon = 0$ then $\|(\langle 0|_a \otimes I_n) U_\psi |0\rangle_{a+n}\|_2 = 1/\alpha$. One of the most essential components of working with matrix-vector arithmetic in quantum algorithms is tracking the norm of the encoded vectors throughout the algorithm, as the quantum complexity is usually inversely proportional to the norm of the encoded vector. Vector encodings give a methodical way to track encoded vector norms when implementing various matrix-arithmetic operations on the encoded vectors.

In summary, block-encodings provide a formal framework for working with matrices in quantum algorithms, and vector-encodings provide a formal way for working with vectors.

## 2.2 New Operations on Vector Encodings

To enable our results on architectural blocks, we had to develop primitive operations on vector-encodings. These results are straight-forward modifications of existing techniques into the VE framework, but are necessary to allow easy tracking of the norm of encoded vectors, which is a crucial parameter dictating the complexity of quantum neural network accelerations.

**Lemma 1** (Vector Sum, Proof in Appendix B). *Let $0 \leq \tau \leq 1$. We are given unitary circuits $U_\psi$ and $U_\phi$ which are $(\alpha, a, \epsilon_0)$ and $(\beta, b, \epsilon_1)$ VEs for $|\psi\rangle_n$ and $|\phi\rangle_n$, respectively. Define $c := \max(a, b)$, $|\Gamma\rangle_n := \frac{\tau}{\alpha}|\psi\rangle_n + \frac{(1-\tau)}{\beta}|\phi\rangle_n$, $\mathcal{N} := \||\Gamma\rangle_n\|_2$ and $|\overline{\Gamma}\rangle_n := |\Gamma\rangle_n/\mathcal{N}$. Then, using one controlled $U_\psi$ circuit, one controlled $U_\phi$ circuit, and two additional single-qubit gates, we can construct a unitary matrix $V$ such that $V$ is a $(\mathcal{N}^{-1}, c+1, (\frac{\epsilon_0}{\alpha} + \frac{\epsilon_1}{\beta})/\mathcal{N})$-VE for $|\overline{\Gamma}\rangle$.*

**Lemma 2** (Matrix-Vector Product, Proof in Appendix B). *We are given an $(\alpha, a, \epsilon_0)$-block-encoding $U_A$ for the $n$-qubit operator $A$, and $U_\psi$ a $(\beta, b, \epsilon_1)$-VE for the $\ell_2$-normalized $n$-qubit quantum state $|\psi\rangle$. Let $\mathcal{N} := \|A|\psi\rangle_n\|_2$. $U_\psi$ has $T_\psi$ circuit complexity, and $U_A$ has $T_A$ circuit complexity. Then, we can obtain an $a + b + n$ qubit unitary $U$ with $O(T_\psi + T_A)$ circuit complexity such that $U$ is an $(\alpha\beta/\mathcal{N}, a + b, (\epsilon_0 + \alpha\epsilon_1)/\mathcal{N})$-VE for the quantum state state $A|\psi\rangle_n/\mathcal{N}$.*

**Lemma 3** (Tensor Product of Vector Encodings, Proof in Appendix B). *Given $U_\psi$ an $(\alpha, a, \epsilon)$-VE for $|\psi\rangle_n$ with $O(T_\psi)$ circuit complexity, and $U_\phi$ an $(\beta, b, \delta)$-VE for $|\phi\rangle_m$ with $O(T_\phi)$ circuit complexity, then we can obtain the circuit $V$ which is an $(\alpha\beta, a + b, \epsilon + \delta + \epsilon\delta)$-VE for $|\psi\rangle_n \otimes |\phi\rangle_m$ with $O(\max(T_\psi, T_\phi) + \max(n, b))$ circuit depth.*

**Lemma 4** (Concatenation of Vector Encodings, Proof in Appendix B). *Let $D = 2^d$, $N = 2^n$, and $0 \leq \epsilon < 1$. Assume that $d \leq n$. Suppose we are given a set of $D$ unitary circuits, $\{U_i\}_{i \in [d]}$ such that each $U_i$ is an $(\alpha_i, a, \epsilon)$-VE for the quantum state $|\psi_i\rangle_n$ with $O(T)$ circuit complexity.* [1] *Let*

---

[1] If $D$ is not a power of 2, padding can be added.

$|\Psi\rangle_{d+n} = \sum_{j=0}^{D-1} |j\rangle_d |\psi_j\rangle / \alpha_j$, *and let* $\mathcal{N} := \||\Psi\rangle_{d+n}\|_2 = \sqrt{\sum_{j=0}^{D-1} \frac{1}{\alpha_j^2}}$. *Then, we can obtain a* $(D/\mathcal{N}, d+a, \epsilon)$ *for* $\frac{|\Psi\rangle_{d+n}}{\mathcal{N}}$ *with* $O(dDT)$ *circuit complexity.*

## 2.3 Matrix Vector Squared Product

We are now ready to present the first key result of this section, showing how given a matrix $W$ (with $\|W\|_2 \le 1$) and a vector encoding of $\boldsymbol{x}$, we can obtain a vector encoding of $W(\boldsymbol{x})^2$. The key idea is to avoid obtaining a quantum block-encoding of the operator $W$ (which in general requires $W$ to be either low-rank, or sparse (Gilyén et al., 2019)). We then implement the product by using importance-weighting to coherently combine the columns of $W$ weighted by the corresponding elements of the input vector, and then apply the result to a modified version of the input vector.

**Theorem 1** (Product of Arbitrary Matrix with a Vector Element-wise Squared, Informal). *Let* $N = 2^n$. *We are given a matrix* $W \in \mathbb{C}^{N \times N}$, *provided via a pre-processed efficient quantum accessible data-structure. Additionally, we are given the unitary* $U_\psi$ *with circuit complexity* $O(T_\psi)$, *a* $(\alpha, a, \epsilon)$-VE *for the quantum state* $|\psi\rangle_n$. *Define the function* $g : \mathbb{C} \mapsto \mathbb{R}$ *as* $g(x) = |x|^2$, *and* $\mathcal{N} := \|Wg(|\psi\rangle_n)\|_2$. *Then we can construct the unitary* $U_f$ *which is a* $(\frac{\alpha^2}{\mathcal{N}}, 2a + 2n + 3, \frac{2\alpha\epsilon}{\mathcal{N}})$-VE *for* $Wg(|\psi\rangle_n)/\mathcal{N}$, *and has* $O(T_\psi + n^2)$ *circuit depth.*[2]

This result is stated formally and proven as Theorem B.1 in the Appendix, and we formally define one possible implementation of the quantum accessible data-structure assumption in Definition B.3. To use this to prepare the quantum state $Wg(|\psi\rangle_n)/\mathcal{N}$, the vector normalization result (Lemma B.8) can be directly applied to the output VE yielded by Theorem 1, preparing the state with $\tilde{O}(\alpha^2(T_\psi + n^2)/\mathcal{N})$ circuit complexity. This is the first such result *without a Frobenius norm dependence* on $A$.

We will now informally sketch the proof of this procedure. First, define the columns of $W$ as $W = (\boldsymbol{w}_0 \quad \dots \quad \boldsymbol{w}_{N-1})$. Define the normalized version as $|w_j\rangle = \boldsymbol{w}_j / \|\boldsymbol{w}_j\|_2$, and define $a_j := \|\boldsymbol{w}_j\|_2$. We assume access to three objects. (1) A block-encoding of $A := \text{diag}(a_0, \dots, a_{N_1})$. (2) An oracle implementing $U_W|0\rangle|j\rangle = |w_j\rangle|j\rangle$. (3) A vector-encoding for $|\psi\rangle = \sum_j \psi_j |j\rangle$. Then, by using our vector-encoding circuit, we can get an encoding of $|\phi\rangle := \sum_j \psi_j |j\rangle |w_j\rangle = (\psi_0\langle w_0| \quad \dots \quad \psi_{N-1}\langle w_1|)^\dagger$. Then, using our block-encoding of $A$, we can efficiently get a block-encoding of $\begin{pmatrix} a_0\psi_0 I_n & \dots & a_{N_1}\psi_{N-1}I_n \\ & \mathbf{0} & \end{pmatrix}$ (where $I_n$ is a $2^n$ dimensional identity matrix, and only the first $N$ rows are non-zero). We can then use the product of matrix-encoding with vector encoding result to take the product of $\begin{pmatrix} a_0\psi_0 I_n & \dots & a_{N_1}\psi_{N-1}I_n \\ & \mathbf{0} & \end{pmatrix}$ with $|\phi\rangle$ yielding the desired vector-encoding.

## 2.4 QRAM-Free Quantum Encoding of 2D Multi-Filter Convolutions

While the matrix-form of a $2D$ convolution has been given many times before in the literature, to the best of our knowledge the following is the first result giving a block-encoding of a QRAM-free 2D multi-filter convolution. We also stress that the following result can be highly optimized, especially if QRAM is used. We leave such optimizations to future work. The full proof is provided in Section B.2.

**Lemma 5** (QRAM-Free Block-Encoding of 2D Convolution With Filters). *Let* $M = 2^m$, *let* $n = 2m$, *let* $N = 2^n$, *and let* $D = 2^d$. *Define the matrix form of the 2D multi-filter convolution operation,* $\mathcal{C} \in \mathbb{R}^{CM^2 \times CM^2}$, *as per Lemma B.17. Here,* $C$ *represents the number of input and output channels, and* $D$ *represents the dimension of the kernel over rows and columns (i.e., the kernel is a rank$-4$ tensor containing* $C, C \times D \times D$ *filters). Then, after performing some one-time classical pre-computation, we can obtain a* $(1, 3 + 8D + 2\log(CD), 0)$- *block-encoding of* $\frac{\mathcal{C}}{2\|\mathcal{C}\|_2}$ *with* $O(m^2 C^3 D^4 \log(C) \log(D))$ *circuit depth.*

While the degrees on the number of channels and the filter size $D$ seem large, the filter size is usually quite small in practice (e.g., often 3). Moreover, there are straight-forward optimizations of this result which can substantially reduce the degrees on both $C$ and $D$. Convolutional layers are

---

[2]For simplicity, here we are assuming that the parameter $d$ (as defined in Theorem B.1) is set to $n$.

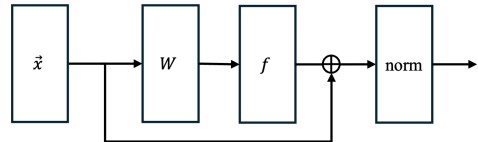

Figure 2: **Generic Residual Architectural Block.** This diagram illustrates the structure of a typical residual block used in deep neural networks. The input vector $x$ is transformed through a sequence of operations: a learnable linear transformation $W$, a non-linear activation function $f$, and a residual (skip) connection that adds the original input to the transformed signal. The output is then passed through a normalization layer (`norm`).

excellent candidates for QRAM-free implementation, since the number of parameters they contain are usually much smaller than the dimension of the vectorized tensors which they act upon. Indeed, we essentially obtain Lemma 5 by efficiently constructing a block-encoding of the matrix-form of the highly-structured object corresponding to each parameter in the convolutional kernel, and then taking a linear combination of the result. This explains why the complexity of our procedure is polylogarithmic in the dimension, whilst being polynomial in the number of parameters. This is in contrast to exact classical algorithms which have polynomial dimension-dependence. Moreover, our result can be substantially optimized further, potentially by exploiting the fact that circulant convolutions are diagonalized by the Fourier transform.

## 3 Architectural Blocks

In this section we will derive two key architectural blocks, a residual block, and a multi-layer residual block, which allow our subsequent complexity claims. We present an additional architectural block building on these in Appendix C, but do not include it in the main text as it is not essential for understanding the key complexity details of such quantum implementations.

**Lemma 6** (General Skip Norm Block). *Let $\epsilon_1 \in (0,1]$. Let $\kappa \in [1,2]$. Consider the architecture shown in Figure 2. Let $N = 2^n$. We are given the unitary $U_\psi$ a $(1, a, \epsilon_0)$-VE for $|\psi\rangle_n$ with circuit complexity $O(T_1)$, and are given the unitary $U_W$ a $(1, b, 0)$-block-encoding for the $n$-qubit operator $W/\kappa$ with circuit complexity $O(T_2)$ such that $\|W\|_2 \leq 1$. Define $f(x) := \mathrm{erf}(4x/5)$, $|\psi_f\rangle_n := |\psi\rangle_n + f(W|\psi\rangle_n)$, and $\mathcal{N} := \||\psi_f\rangle_n\|_2$. Then, we can obtain a $(1, 2(a+b)+n+9, 712(\epsilon_0 + \epsilon_1))$-VE for $|\psi_f\rangle_n/\mathcal{N}$ with circuit complexity $O(\log(\frac{\sqrt{N}}{\epsilon_1}) \log(\frac{1}{\epsilon_1})(a + b + n + T_1 + T_2))$.[3]*

The rigorous proof of this result is provided in Appendix C, but it essentially follows from using our preceding results on matrix-vector multiplication, vector sums, and the extant results on layer normalization and applications of the error-function. The key insight enabling this proof is that in a residual block such as the one we have described, the forward norm of the vector is efficiently lower-bounded prior to every normalization layer. Without such skip connection, and the techniques we developed for working with vector-encodings (which enable effective tracking of the norm of a vector propagating through a network), the norm at the end of such a block could be arbitrarily small, leading to complexities which could be on the order of $\approx N^k$ (or even unbounded) for $k$-layer architectures – completely intractable even for constant depth networks. As a consequence, we are able to prove the following result for multi-layer residual blocks.

**Lemma 7** (Sequence of $k$ Residual Blocks). *Let $N = 2^n$. Suppose we are given a unitary $U_\psi$ with circuit complexity $O(T_1)$ such that it is a $(1, a, 0)$-VE for $|\psi\rangle_n$. Let $k$ be an asymptotic constant. Suppose we have a sequence of $k$ residual blocks (as per Lemma 6), with weights implemented by $k$ unitaries $\{U_{W_i}\}_i$ such that $U_{W_i}$ (with circuit complexity $O(T_2)$) is a $(1, b, 0)$-block-encoding for the $n$-qubit operator $W_i/2$, and $\forall i, \|W_i\|_2 \leq 1$. Then, we can prepare a $(1, 2^k(a + 2b + n + 9), \epsilon)$-VE for the output of the $k$ residual blocks with $O(\log(\sqrt{N}/\epsilon)^{2k}(a + 2b + n + T_1 + T_2))$ circuit depth.*

This result is proven in Appendix C, and follows by repeatedly invoking Lemma 6 with its output as the next input. It appears that the complexity of this result as a function of the number of layers $k$ is

---

[3]We implicitly assume that $\||W|\psi\rangle_n\| > 0$, which is a reasonable assumption for any input which comes from the same distribution as the training data.

a fundamental limitation of any quantum algorithm. As described in greater detail in Appendix C, for a unitary matrix (a linear operator) to enact a non-linear transformation on a vector, its definition must in general be input-dependent. Consequently, unless Lemma 6 can be implemented with only a single copy of its input, it seems unlikely that this complexity can be avoided. This suggests that quantum computers are best suited for accelerating the wide and shallow regime, which is a popular regime for classical inference accelerators (since wide networks can be parallelized on classical hardware, but depth cannot be parallelized). Classically, with the aim of accelerating both inference and training, there are a range of techniques for compressing neural networks (Cheng et al., 2018). Moreover, classically, deep neural networks are much harder to accelerate than their shallow and wide counterparts (you can parallelize matrix-multiplications, but not consecutive layers). Consequently, there are a number of classical architectures striving for shallow networks (e.g. Zagoruyko & Komodakis (2016)) which can serve as sources of inspiration for designing architectures best suited for quantum acceleration. We discuss this in greater detail in Appendix C.

## 4    ARCHITECTURES

We will now use the architectural blocks derived above to prove the quantum complexity in inference for the architectures shown in Figure 1 (a), which is then used to prove the complexity of the architecture in panel (b). A corollary is used to prove the complexity of the architecture in panel (c).

In all 3 regimes, the key architectural block shared in common is the sequence of $k$ residual convolutional blocks, which is enacted by combining Lemma 5 and Lemma 7. The architectures then only differ in how the input tensor is transformed, and in how the output of the $k$ residual convolutional blocks is processed. Consequently, we will now provide high-level intuition for the important sequence of $k$ residual convolution blocks. First, Lemma 7 is simply obtained by chaining the result for a single residual block (given by Lemma 6) $k$ times, using the output of each invocation as the input for the next. Lemma 6 itself is implemented by enacting each of the vector-encoding operations corresponding to the operations shown in Figure 2: matrix-vector multiplication via Lemma 2, non-linear activation via Lemma B.19, vector sum via Lemma 1, and vector normalization via Lemma B.8. Noting that Lemma B.19 and Lemma B.8 are straight-forward improvements over the results from prior work, we delegate them to the appendix. It is also worth noting that our selection of the erf activation function is not restrictive, and was selected for analytical convenience. This could easily be swapped with other activation functions compatible with Lemma B.18, e.g., GELU or tanh. Finally, the last key piece of intuition regards the dimension of the specific vectorized tensor which is input to the sequence of $k$ residual blocks. In Regime 1, this tensor is simply a fixed concatenation of the input tensor, and consequently for an input with vectorized dimension $O(N)$ has dimension $O(N)$. In Regimes 2 and 3, the input tensor is mapped through a tensor product $d$ times, resulting in an input to the residual block sequence of dimension $O(N^2)$ (when $d = 2$).

Thus, our results in all 3 data-access regimes all follow from the general result, formally stated below:

**Theorem 2** (General Multilayer Convolutional Network with Skip Connections). *Let $M = 2^m$, $N = 2^n = M^2$. Consider the neural network architecture shown in Figure 1 (a). Let the input $X$ be a rank$-3$ tensor of dimension $4 \times M \times M$ (with an R, G, B and null channel, where the null channel has all 0s). Assume that $\|vec(X)\|_2 = 1$, and that we have access to a unitary $U_X$ that is a $(1, 0, 0)$-VE for the input in column-major layout $|X\rangle_{2+2m} = \sum_{i=0}^{4} \sum_{j=0}^{M-1} \sum_{k=0}^{M-1} X_{i,k,j} |i\rangle_2 |j\rangle_m |k\rangle_m$. Assume that $U_X$ has $O(T_X)$ circuit complexity. As shown in the figure, we have a sequence of $k$ residual convolutional layers, where each convolutional layer has $16$ input channels, $16$ output channels (i.e., $16$ filters) with filter width and height $3$. I.e., each convolutional layer has $16 \times 16 \times 3 \times 3 = 2304$ parameters. Assume that there is $0$ padding so the input and outputs always have the same dimension, and that there is a stride of $1$. Suppose each convolutional layer has been regularized, so that its spectral norm is at most $1$. Let $W$ represent the $N \times N$ full-rank linear layer applied in the final output block of the network, and assume that $\|W\|_2 \le 1$. Let $C$ represent the number of output classes, and assume that $C = 2^c$ (padding can be added otherwise). Let the overall network be represented by the function $h : \mathbb{R}^{4 \times M \times M} \mapsto \mathbb{R}^C$. Let $\boldsymbol{y} = h(X)$ (and note that $\|\boldsymbol{y}\|_1 = 1$, and $\boldsymbol{y} \in \mathbb{R}^C$). Then, with $O(\log(\sqrt{N}/\epsilon)^{2k+1}(T_X + n^2))$ total circuit depth, and with $O(2^k n)$ ancillary qubits, we can draw a sample from an $\ell_1$-normalized $C$-dimensional vector $\tilde{\boldsymbol{y}}$ such that $\|\boldsymbol{y} - \tilde{\boldsymbol{y}}\|_2 \le \epsilon$.*

*Proof.* We have a $4$ channel input and we want to map this to a $16$ channel input (by concatenating $|X\rangle_{2+2m}$ vector with itself $4$ times). Let $|X\rangle_{4+2m} := \frac{1}{\sqrt{4}} (\langle X|_{2+2m} \quad \langle X|_{2+2m} \quad \langle X|_{2+2m} \quad \langle X|_{2+2m})^T$. We can invoke Lemma 4 with $U_X$ four times, obtaining a $(1,0,0)$-VE for $|X\rangle_{4+2m}$ with $O(T_X)$ circuit complexity. Using Lemma 5, for each of the $i = 0, ..., k-1$ convolutions, we can obtain a $(1, 27, 0)$-block-encoding for $\mathcal{C}_i/2\|\mathcal{C}_i\|_2$ (the matrix form of the corresponding convolution) with $O(m^2)$ circuit depth. Consequently, we can invoke Lemma 7 to obtain $U_{\text{conv}}$ a $(1, 2^k(63+n), \epsilon)$-VE for the $\ell_2$-normalized output of the sequence of $k$ residual blocks. Moreover, $U_{\text{conv}}$ has $O(\log(\sqrt{N}/\epsilon)^{2k}(n + T_X + m^2))$ circuit depth. Then, we can invoke Lemma C.2 with $U_{\text{conv}}$ to draw a sample from some probability vector $\tilde{y} \in \mathbb{R}^C$ such that $\|\tilde{y} - y\|_2 \le \epsilon$ with $O(\log(\sqrt{N}/\epsilon)^{2k+1}(T_X + n^2))$ circuit depth and with $O(2^k n)$ ancilla qubits. $\square$

An important point to consider is that in order for a unitary matrix (or more generally, any linear operator) to enact a non-linear transformation, its definition must depend on the vector it is being applied to. For instance, consider the simple example where we are given a vector $x$, and we define $A := \text{diag}(x)$. Then, $Ax = (x)^2$ (with the square applied element-wise) which is clearly a non-linear transformation. Consequently, our algorithm for Theorem 2 adaptively (and efficiently) constructs a new circuit on the fly for each new input – this is accounted for in the result statement.

## 4.1 KEY RESULTS UNDER DIFFERING QUANTUM DATA ACCESS ASSUMPTIONS

The feasibility of quantum random access memory, the primary method assumed in the literature for accessing classical data in quantum algorithms, is widely debated in the literature (Jaques & Rattew, 2025). However, recent work (Dalzell et al., 2025a) provides a promising path forward, addressing many of the limitations raised in Jaques & Rattew (2025). Regardless, algorithms papers often fail to meaningfully address the memory assumptions they make, and so we include a comprehensive discussion of it in Appendix D highlighting the feasibility of the technology, and that importantly our QRAM assumptions are no stronger than the usual made in such algorithms papers. The key concept discussed in Appendix D is that any algorithm utilizing a QRAM device must consider the classical opportunity cost of using that device, which dictates the constraints placed on realizing a useful QRAM (e.g., for such purposes the physical QRAM device cannot simply be implemented in the circuit model).

**Regime 1: Input and Network Use QRAM.** The primary purpose of the architecture we presented in Regime 1 is to show that quantum computers can implement multi-layer neural networks based on real architectures coherently, with reasonable input assumptions, and with cost polylogarithmic in the dimension of the network. As per the main-text, in this regime we assume that the matrix weights (in particular for the final full-rank linear layer) and vectorized input are provided via QRAM. The architecture for this regime is shown in Figure 1 (a). Let the dimension of the vectorized input be $O(N)$. Since the input is provided via QRAM, $T_X$ as defined in Theorem 2 is $T_X \in O(\text{polylog}(N))$ (see, Section D.2). **Thus, for a constant number of layers $k$, the cost to perform inference (in accordance with Definition 1) becomes** $O(\text{polylog}(\sqrt{N}/\epsilon)^k)$. Please see Section E.1 for a detailed discussion outlining important application areas where such input assumptions are practical (namely, where the input can be constructed in an amortized fashion online). Moreover, in Section E.1 we also discuss considerations relating the receptive field of such architectures, and argue that existing techniques are insufficient to dequantize this result.

**Regime 2: Network Stored in QRAM, Input Loaded Without QRAM.** The architecture in this regime is shown in Figure 1 (b). The architecture contains $d$ paths of purely classical neural networks, which each operate on $O(N)$ dimensional (vectorized) inputs. These classical architectures are assumed to have $\tilde{O}(N)$ time complexity in terms of the input. These separate paths are then normalized, converted to quantum states, and then the Kronecker product of the result is taken. The result is fed into exactly the same architecture as in Regime 1. This architecture is inspired by bilinear neural networks (Lin et al., 2015). Consequently, to determine the cost of this architecture, we can again invoke Theorem 2. Here, we need to pay an $\tilde{O}(N)$ cost to load each of the input paths in as a quantum state (via brute-force (Plesch & Brukner, 2011)),$T_X \in O(N)$. Consequently, we obtain an overall algorithmic complexity of $O(N \log(\frac{N^{d/2}}{\epsilon})^{2k})$, which for constant $k$ and $d$, simplifies to

$\tilde{O}(N \log(1/\epsilon)^{2k})$. When $d = 2$, the dimension after the tensor product is $N^2$. Consequently, the final linear layer contains a matrix multiplication of an $N^2 \times N^2$ matrix with an $N^2$ dimensional vector, which takes $\Omega(N^4)$ time. **Consequently, for a constant $k$, this architecture produces a quartic speedup for the inference problem defined in Definition 1 over exact classical computation**. When $d = 1$, the speedup due to the final layer is instead quadratic. This speedup can be increased by setting $d$ to larger values.

**Regime 3: No QRAM.** This architecture is identical to the one presented in Regime 2, only dropping the final full-rank linear block. In Section E.3 we show that the architecture in Figure 1 (c) can perform inference with a total $O(N \log(1/\epsilon)^{2k})$ circuit complexity. Since the dimension of the vector acted on by the 2D convolution is $O(N^2)$ (when d=2), the classical cost to compute this is $\Omega(N^2)$: showing **a quadratic speedup over an exact classical implementation**. The speedup can be made asymptotically larger by increasing $d$. We have a more detailed discussion of this regime in Section E.3.

## 5 CONCLUSION

This work proposes a modular framework for accelerating classical deep learning inference using fault-tolerant quantum subroutines. Our approach offers direct quantum implementations of important neural network architectural blocks (such as convolutions, activation functions, normalization layers, and residual connections), and uses structured primitives such as quantum block-encodings.

In summary, we provide a number of novel theoretical contributions. We further develop the VE framework for quantum vector encodings. We derive a novel quantum algorithm for the multiplication of an arbitrary dense and full-rank matrix with the element-wise square of a given vector, which to the best of our knowledge, is the first such result which does not incur a Frobenius norm (and thus rank) complexity dependence. We provide a novel QRAM-free block-encoding of multi-filter 2D convolutions. We then prove the first end-to-end complexity guarantees for the coherent quantum acceleration of multi-layer neural network inference, under three QRAM regimes. In the first regime, we give complexity which is polylogarithmic in both the dimension of the input, and the number of parameters in the network. In the second, we show a quartic speedup over exact classical computation. In the third, we show a quadratic speedup.

## 6 FUTURE WORK

To the best of our knowledge, this is the first paper to implement multi-layer neural networks coherently on a quantum computer, and as such, many important open directions of research remain. Moreover, progress towards achieving a practically passive QRAM is important for realizing the speedups in the first two regimes. Moreover, exploring the connection between this work and the techniques utilized in scientific computing (e.g., quantum differential equation solvers, finite difference methods, etc (Cao et al., 2013; Montanaro, 2016; Childs et al., 2021; Berry & Costa, 2024; Jennings et al., 2024; An et al., 2026; Shang et al., 2025; Liu et al., 2021a; 2023; Krovi, 2023; Costa et al., 2025; Wu et al., 2025)) would be interesting. Most importantly, we wonder if it is possible to coherently enact sequences of non-linear transformations without an exponentially increasing circuit depth (and with polylogarithmic error-dependence), thereby allowing very deep multi-layer architectures to be quantized, but we suspect that this may be provably impossible (at least in general). Furthermore, it is conceivable that an approach enacting the non-linear transformations coherently with techniques based on QPE (Mitarai et al., 2019) might be able to enact a sequence of non-linearities without exponentially increasing circuit depth (albeit at the cost of an exponentially worse and exponentially decaying error-dependency). Combining such approaches may let quantum computers coherently accelerate architectures with depths of e.g., up to 25. Alternatively, one could combine sequences of coherent multi-layer architectural blocks with intermittent tomography to reset the depth cost, in essence fusing the techniques presented in our paper with those used in the prior work. It would also be worthwhile to explore accelerating UNet based architectures, as many of our techniques directly apply, and a distilled UNet-based diffusion model could potentially be quite shallow. Finally, while this work assumes our networks are trained classically, it would be interesting to explore how the techniques we develop could also be used to help accelerate training.

ACKNOWLEDGMENTS

AGR would like to thank Simon Benjamin and Sam Jaques for helpful discussions about QRAM. AGR acknowledges support from the Engineering and Physical Sciences Research Council (EPSRC) project Software Enabling Early Quantum Advantage (SEEQA) under grant EP/Y004655/1, and additionally acknowledges support from Quantum Motion. PWH acknowledges support from the EPSRC Doctoral Training Partnership (DTP) under grant EP/W524311/1, with a CASE Conversion Studentship in collaboration with Quantum Motion. PWH further acknowledges support from the Ministry of Education, Taiwan, for a Government Scholarship to Study Abroad (GSSA) and St. Catherine's College, University of Oxford, for an Alan Tayler Scholarship. NG, LP, and PR are supported by the National Research Foundation, Singapore, and A*STAR under its CQT Bridging Grant and its Quantum Engineering Programme under grant NRF2021-QEP2-02-P05. NG also acknowledges support through the Research Excellence Scholarship from SandboxAQ. LP additionally acknowledges support from the Alice Postdoctoral Fellowship, awarded by the Centre for Quantum Technologies, National University of Singapore.

AUTHOR CONTRIBUTIONS

AGR conceived of the project, led it, and developed the theory. PWH contributed to the design of the architectures considered, and produced the key figures. AGR and PWH drafted the original manuscript. PR supervised the project, and provided input on the theory. NG verified the proofs. LP helped position the paper over prior work. All authors contributed to the final manuscript.

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

TECHNICAL APPENDICES AND SUPPLEMENTARY MATERIAL

In Appendix A we present a summary of Quantum Random Access Memory (QRAM), which we subsequently use. In Appendix B we present a number of existing techniques which we require to manipulate vectors and matrices with quantum computers, and then use them to develop a number of new useful results for quantum matrix-vector arithmetic. In Appendix C, we use the techniques developed in Appendix B to construct quantum-implementations of key architectural blocks. In Appendix D, we discuss the feasibility of QRAM. In Appendix E we use the architectural blocks obtained in Appendix C to derive end-to-end complexities for a number of architectures under different QRAM assumptions.

## A  QUANTUM RANDOM ACCESS MEMORY (QRAM)

Quantum Random Access Memory (Giovannetti et al., 2008b) is a widely assumed mechanism in the quantum computing literature for accessing data in a quantum computer. In this paper, we make a range of QRAM assumptions under different regimes of assumed feasibility. With the aim of enabling practical end-to-end speed-ups, it is important to explicitly state the different assumptions and consider the feasibility of each of these regimes.

In this section, we will formally define QRAM, and state the assumed complexities. In Appendix D, we dive into a deeper discussion of the feasibility of our various QRAM assumptions, with the aim of providing a clear understanding of what sorts of end-to-end speed-ups our results can offer in practice.

**Definition A.1** (QRAM for Classical Data). *Let $N = 2^n$ and $D = 2^d$. Let $|i\rangle_n$ be any $n$-qubit standard basis vector, and let $x_i \in [D]$. Then, a QRAM with $O(dN \log N)$ total qubits can implement the mapping,*

$$U|i\rangle_n|0\rangle_d = |i\rangle_n|x_i\rangle_d \tag{A.1}$$

*with $O(d \log N)$ circuit depth.*

As mentioned in a number of sources, e.g., Hann et al. (2021); Giovannetti et al. (2008a) an $N$ qubit QRAM can be implemented with $O(\log N)$ depth complexity. Consequently, performing a sequence of $d$ of these (to implement each of the $d$-bits in each memory register), a circuit depth complexity of $O(d \log N)$ trivially follows.

**Definition A.2** (QRAM for Quantum Data  (Prakash, 2014; Kerenidis & Prakash, 2017; Kerenidis et al., 2020)). *Let $N = 2^n$, $M = 2^m$. Let $|i\rangle_n$ be any $n$-qubit standard basis vector. Allow $|\psi_i\rangle_m$ to be an arbitrary $m$-qubit normalized quantum states. Then, a QRAM with $\tilde{O}(MN)$ total qubits, and $\tilde{O}(MN)$ classical pre-processing to construct the data-structure, can implement the mapping,*

$$U|i\rangle_n|0\rangle_m = |i\rangle_n|\psi_i\rangle_m \tag{A.2}$$

*with $O(\log^2(NM))$ circuit depth.*

Importantly, as per Prakash (2014); Kerenidis & Prakash (2017); Kerenidis et al. (2020) QRAM for quantum data can be implemented by a circuit (based on Grover & Rudolph (2002)) with depth and width $O(\text{polylog}(MN))$ with access to a QRAM data structure (as per Definition A.1) containing all the entries of each state in the quantum data (along with $O(\log M)$ copies for each of the sets of partial norms). Thus, if QRAM for classical data is feasible (as discussed in Appendix D), QRAM for quantum data is as well (with pre-processing to construct the appropriate data-structures).

In this work, we will use QRAM to describe QRAM for both quantum and classical data, and will make the distinction clear when it is relevant.

## B  QUANTUM MATRIX-VECTOR ARITHMETIC

In this section, we formally derive a number of tools for quantum matrix-vector arithmetic.

**Lemma B.1** (Product of block encodings (Gilyén et al., 2019)). *If $U$ is an $(\alpha, a, \delta)$-block-encoding of an $s$-qubit operator $A$, and $V$ is an $(\beta, b, \epsilon)$-block-encoding of an $s$-qubit operator $B$ then $(I_b \otimes U)(I_a \otimes V)$ is an $(\alpha\beta, a + b, \alpha\epsilon + \beta\delta)$-block-encoding of $AB$.*

In Lemma B.1 we adopt the tensor product notation used in Gilyén et al. (2019); the tensor product in this lemma is used differently than it is used anywhere else in this paper.

We now present a standard result (see Lemma 1 of Camps & Van Beeumen (2020) or Lemma 21 of Chakraborty et al. (2023)), and we include the proof for completeness, as it is the basis of a subsequent proof Lemma 3. In particular, our derivation closely follows that of Lemma 1 of Camps & Van Beeumen (2020).

**Lemma B.2** (Tensor Product of Block-Encoded Operators). *Given a unitary $U_A$ which is an $(\alpha, a, \epsilon_0)$-block-encoding for $n$-qubit operator $A$ with $O(T_A)$ circuit complexity, and a unitary $U_B$ which is a $(\beta, b, \epsilon_1)$-block-encoding for $m$-qubit operator $B$ with $O(T_B)$ circuit complexity, we can obtain an $(\alpha\beta, a + b, \epsilon_0\beta + \epsilon_1\alpha + \epsilon_0\epsilon_1)$-block-encoding for $A \otimes B$ with $O(\max(T_A, T_B) + \max(n, b))$ circuit complexity.*

*Proof.* The main idea is that $U_A \otimes U_B$ almost directly implements a block-encoding of $A \otimes B$, but the ancillas and the main computation registers are in the wrong order. To correct this, we need to swap the ancilla register of $U_B$ with the main register of $U_A$.

Consequently, define the operator $\Pi$ such that it swaps the $n$-qubit register with the $b$-qubit register (and leaves the other registers unchanged), so that all the ancilla registers precede the main registers. If $n \geq b$, $\Pi$ can be implemented by a sequence of $O(n/b)$ swaps, with each swap swapping $O(b)$ qubits in parallel. If $n < b$, then it can be implemented with $O(b/n)$ swaps. Thus, $\Pi$ has a circuit depth bounded by $O(\max(n/b, b/n)) \in O(\max(n, b))$. Then, $\Pi(|0\rangle_{a+b} \otimes I_{n+m}) = (|0\rangle_a \otimes I_n) \otimes (|0\rangle_b \otimes I_m)$, and $(\langle 0|_{a+b} \otimes I_{n+m})\Pi^\dagger = (\langle 0|_a \otimes I_n) \otimes (\langle 0|_b \otimes I_m)$.

Following Camps & Van Beeumen (2020), define $\tilde{A} := (\langle 0|_a \otimes I_n)U_A(|0\rangle_a \otimes I_n)$, and $\tilde{B} := (\langle 0|_b \otimes I_m)U_B(|0\rangle_b \otimes I_m)$. Let $E_A := A - \alpha\tilde{A}$, and let $E_B := B - \beta\tilde{B}$. Define $V := \Pi^\dagger(U_A \otimes U_B)\Pi$. Then, $A \otimes B = (\alpha\tilde{A} + E_A) \otimes (\beta\tilde{B} \otimes E_B)$, and $(\langle 0|_{a+b} \otimes I_{n+m})V(|0\rangle_{a+b} \otimes I_{n+m}) = \tilde{A} \otimes \tilde{B}$, so

$$\|A \otimes B - \alpha\beta(\langle 0|_{a+b} \otimes I_{n+m})V(|0\rangle_{a+b} \otimes I_{n+m})\|_2 \tag{B.1}$$

$$= \left\|(\alpha\tilde{A} + E_A) \otimes (\beta\tilde{B} \otimes E_B) - \alpha\beta\tilde{A} \otimes \tilde{B}\right\|_2 \tag{B.2}$$

$$\leq \epsilon_0\beta + \epsilon_1\alpha + \epsilon_0\epsilon_1. \tag{B.3}$$

$\square$

We now present a result from the literature allowing a block-encoding to have all of its singular values scaled by a constant value. We present the result nearly verbatim from Lemma 5 of Wada et al. (2025) (with trivial modifications to make it easier to invoke in our context), which presents the results of Low & Chuang (2017); Gilyén et al. (2019) cleanly in the language of block-encodings.

**Lemma B.3** (Uniform Singular Value Amplification (Wada et al., 2025; Low & Chuang, 2017; Gilyén et al., 2019)). *Let $\epsilon, \delta \in (0, 1/2)$, and let $\gamma > 1$. Let $U_A$ be an $(1, a, 0)$-block-encoding of the $n$-qubit operator $A$ with $O(T)$ circuit depth. Suppose $\|A\|_2 \leq (1 - \delta)/\gamma$. Then, we can obtain a quantum circuit $V$ which is a $(1, a + 1, \epsilon)$-block-encoding for $\gamma A$ with $O(\frac{\gamma}{\delta} \log(\gamma/\epsilon)(T + a))$ circuit depth, and with $O(poly(\frac{\gamma}{\delta} \log(\gamma/\epsilon)))$ classical computation to determine the QSVT rotation angles.*

*Proof.* This is taken directly from Wada et al. (2025); Low & Chuang (2017); Gilyén et al. (2019), simply noting that an $a$-controlled $X$ gate can be implemented by a sequence of $O(a)$ single and two-qubit gates. $\square$

We now present a simple result which is just a special case of uniform singular value amplification (Wada et al., 2025; Low & Chuang, 2017; Gilyén et al., 2019) in the case where all the singular values of an encoded operator are either 0 or 1/2. This is done following the ideas of oblivious amplitude amplification (see Gilyén et al. (2019)).

**Lemma B.4** ($\frac{1}{2}$ Oblivious Amplitude Amplification). *We are given a matrix $A \in \mathbb{C}^{N \times N}$, with singular values either 1 or 0. Assume we have access to $U_A$ a $(2, a, 0)$-BE of $A$ with $O(T)$ circuit depth. One can construct $(1, a + 1, 0)$-BE of $A$ with $O(T)$ circuit depth, and with 3 calls to a controlled-$U$ circuit.*

*Proof.* Note that $T_3(x) = 4x^3 - 3x$ satisfies the condition that $|T_3(x)| \leq 1$ for $x \in [-1, 1]$ and $T_3(\frac{1}{2}) = -1$. Therefore, one can achieve the task by implementing the function $-T_3(x)$ via QSVT and the block encoding. The first kind of the Chebyshev polynomial can be directly achieved without any classical processing to determine angle rotations, so one can construct the block encoding with no error. $\qquad \square$

For completeness, we now re-derive an existing result on the linear combination of block-encoded matrices, directly following Gilyén et al. (2019) (which presents the result of Childs & Wiebe (2012) in the context of block-encodings).

**Lemma B.5** (Linear Combination of Block-Encodings (Childs & Wiebe, 2012; Gilyén et al., 2019)). *Suppose we are given a set of $D = 2^d$ unitaries $\{U_i\}_i$ such that each $U_i$ is an $(\alpha, a, \epsilon)$-block-encoding for $n$ qubit operator $A_i$, and each $U_i$ has a total of $O(T_0)$ single and two qubit gates. Define the vector $\boldsymbol{b} \in \mathbb{C}^D$ such that $\boldsymbol{b} = \begin{pmatrix} b_0 & b_1 & \ldots & b_{D-1} \end{pmatrix}^T$. Define $|b\rangle_d = \sum_{j=0}^{D} \sqrt{b_j} |j\rangle_d$ and $\beta := \||b\rangle_d\|_2^2 = \|\boldsymbol{b}\|_1$. We are given the $d$-qubit unitary $U_b$, with $O(T_1)$ single and two qubit gates, such that $U_b|0\rangle_d = |b\rangle_d / \||b\rangle_d\|_2$. Define $A := \sum_{j=0}^{D-1} b_j A_j$. Then, we can obtain a unitary $V$ with $O(dDT_0 + T_1)$ circuit depth which is an $(\alpha\beta, a + d, \alpha\beta\epsilon)$-block-encoding for $A$.*

*Proof.* For each $j \in [D]$, let $\tilde{A}_j := (\langle 0|_a \otimes I_n) U_j (|0\rangle_a \otimes I_n)$, and let $E_j := A_j - \alpha \tilde{A}_j$. Define $S := \sum_{j=0}^{D-1} |j\rangle\langle j|_d \otimes U_j$. Note that $S$ can be implemented by a sequence of $D$ multi-controlled $U_j$ operators. Note that by using Saeedi & Pedram (2013), a $d$ controlled gate targeting 1 or 2 qubits can be decomposed into a sequence of $O(d)$ single and two qubit gates. Consequently, each $d$-controlled $U_j$ has $O(dT_0)$ circuit depth in terms of single and two qubit gates. Thus, $S$ consists of a total of $O(dDT_0)$ single and two qubit gates. Then, define $V := (U_b^\dagger \otimes I_{a+n}) S (U_b \otimes I_{a+n})$.

Noting that $(\langle 0|_d \otimes I_{a+n}) V (|0\rangle_d \otimes I_{a+n}) = \frac{1}{\beta} \sum_{j=0}^{D-1} b_j U_j$. Using the fact that $|0\rangle_{a+d} \otimes I_n = (|0\rangle_d \otimes I_{a+n})(|0\rangle_a \otimes I_n)$, we then obtain

$$(\langle 0|_{a+d} \otimes I_n) V (|0\rangle_{a+d} \otimes I_n) = \frac{1}{\beta} (\langle 0|_a \otimes I_n)(\sum_{j=0}^{D-1} b_j U_j)(|0\rangle_a \otimes I_n) = \frac{1}{\beta} \sum_{j=0}^{D-1} b_j \tilde{A}_j. \quad \text{(B.4)}$$

Consequently,

$$\|A - \alpha\beta(\langle 0|_{a+d} \otimes I_n) V (|0\rangle_{a+d} \otimes I_n)\|_2 = \left\| \sum_{j=0}^{D-1} b_j(\alpha \tilde{A}_j + E_j) - \sum_{j=0}^{D-1} \alpha b_j \tilde{A}_j \right\|_2 \quad \text{(B.5)}$$

$$= \left\| \sum_{j=0}^{D-1} b_j \alpha E_j \right\|_2 \leq \alpha \sum_{j=0}^{D-1} |b_j| \|E_j\|_2 \quad \text{(B.6)}$$

$$\leq \alpha\beta\epsilon. \quad \text{(B.7)}$$

Thus, $V$ gives a $(\alpha\beta, a, \alpha\beta\epsilon)$-block-encoding for $A$, and has $O(dDT_0 + T_1)$ circuit depth. $\qquad \square$

The following is a standard result which has been used in various contexts, and is included for completeness.

**Lemma B.6** (Block Encoding of Rank 1 Projector of Basis Vectors). *Let $n \in \mathbb{N}_{\geq 0}$, and let $N = 2^n$. Define $i \in [N]$ and $j \in [N]$. Then, we can get a unitary $U$ which is a $(1, 2, 0)$-block-encoding of the $n$ qubit operator $|i\rangle\langle j|$. Moreover, $U$ has $O(n)$ circuit depth.*

*Proof.* Following Jaques & Rattew (2025), a $(1, 2, 0)$ block-encoding of the matrix $|0\rangle\langle 0|$, call it $V$, can be obtained with $O(n)$ circuit complexity. This follows by constructing a $(1, 0, 0)$ block-encoding of the Grover reflection operator, $I - 2|0\rangle\langle 0|$, and taking a linear combination with $I$ via the sum of block-encoding result of Gilyén et al. (2019). The circuit complexity is dominated by reflection operator, which can be implemented by applying a $n - 1$ controlled $XZX$ gate on the most significant qubit, controlled on the 0 state of the other $n - 1$ qubits. Using Saeedi & Pedram (2013) this can be decomposed into a sequence of $O(n)$ two-qubit gates. Decompose

$i$ and $j$ into bits as, $i = i_0 i_1 \ldots i_{n-1}$, and $j = j_0 j_1 \ldots j_{n-1}$. We now define two operators, $M_i := X^{i_0} \otimes X^{i_1} \otimes \ldots \otimes X^{i_{n-1}}$ and $M_j := X^{j_0} \otimes X^{j_1} \otimes \ldots \otimes X^{j_{n-1}}$. Clearly, $M_i |0\rangle\langle 0| M_j = |i\rangle\langle j|$. Then, since $(I_2 \otimes M_i) V (I_2 \otimes M_j) = \begin{pmatrix} |i\rangle\langle j| & \cdot \\ \cdot & \cdot \end{pmatrix}$. Thus, $(I_2 \otimes M_i) V (I_2 \otimes M_j)$ is a $(1, 2, 0)$ block-encoding for $|i\rangle\langle j|$. $\qquad\qquad\square$

We now present a simple result which helps intuitively visualize VEs as encoding vectors in a subspace.

**Lemma B.7** (Intuitive Picture of VE as a Vector Subspace Encoding). *Let $U_\psi$ be an $(\alpha, a, \epsilon)$-VE for $|\psi\rangle_n$. Define $|E_\psi\rangle_n := |\psi\rangle_n - \alpha(\langle 0|_a \otimes I_n) U_\psi |0\rangle_{a+n}$. Define the $a$-qubit operator $p_j^a := |j\rangle\langle j|$. Then,*

$$U_\psi |0\rangle_{a+n} = \frac{|0\rangle_a |\psi\rangle_n - |0\rangle_a |E_\psi\rangle_n}{\alpha} + \sum_{j=1}^{2^a - 1} (p_j^a \otimes I_n) U_\psi |0\rangle_{a+n} = \begin{pmatrix} \frac{|\psi\rangle_n - |E_\psi\rangle_n}{\alpha} \\ \vdots \end{pmatrix}. \qquad (\text{B.8})$$

*Proof.* $|E_\psi\rangle_n = |\psi\rangle_n - \alpha(\langle 0|_a \otimes I_n) U_\psi |0\rangle_{a+n}$ implies that $|0\rangle_a |E_\psi\rangle_n = |0\rangle_a |\psi\rangle_n - \alpha(p_0^j \otimes I_n) U_\psi |0\rangle_{a+n}$. The result follows trivially by algebraic maniuplation of $U_\psi |0\rangle_{a+n} = (\sum_{j=0}^{2^a - 1} p_j^a \otimes I_n) U_\psi |0\rangle_{a+n}$. $\qquad\square$

Intuitively, in the absence of error, the first $2^n$ entries of $U_\psi |0\rangle_{a+n}$ will contain the sub-normalized vector $|\psi\rangle_n / \alpha$.

We now state the following result from Rattew & Rebentrost (2023) nearly verbatim, slightly improving the complexity. The following result is a tool essentially implementing $\ell_2$ layer normalization, follows directly from oblivious amplitude amplification (see e.g., Gilyén et al. (2019)), and is taken nearly verbatim from Rattew & Rebentrost (2023).

**Lemma B.8** (Vector Normalization, Lemma 18 of Rattew & Rebentrost (2023)). *Let $\epsilon_0 \in [0, 1/2]$, $\alpha \geq 1$, $a \in \mathbb{N}$, $\epsilon_1 > 0$. Let $\alpha'$ be a known bound such that $\alpha' \geq \alpha$. Given a unitary $U_\psi$, a $(\alpha, a, \epsilon_0)$-VE for the $\ell_2$-normalized quantum state $|\psi\rangle_n$ with circuit complexity $O(T_\psi)$, we can construct a $(1, a + 4, 2(\epsilon_0 + \epsilon_1))$-VE for $|\psi\rangle_n$ with circuit complexity $O((T_\psi + a + n)\alpha' \log(1/\epsilon_1))$ and with $O(\alpha' \log(1/\epsilon_1))$ queries to a $U_\psi$ and $U_\psi^\dagger$ circuit.*

This implements vector normalization by boosting the scaling factor so the norm of the encoded vector is 1, and all the padding entries are 0 (up to logarithmic error).

*Proof.* Define $|\phi\rangle_n := (\langle 0|_a \otimes I_n) U_\psi |0\rangle_{a+n}$, $\mathcal{N}_\phi := \||\phi\rangle_n\|_2$, $|\Phi\rangle_n := |\phi\rangle_n / \mathcal{N}_\phi$. Then, $U_\psi$ is equivalently a $(\mathcal{N}_\phi, a, 0)$-VE for $|\phi\rangle_n / \mathcal{N}_\phi$. Using Lemma B.6, we can get $U_0$ a $(1, 2, 0)$-block-encoding of the $n + a$ qubit projector $|0\rangle\langle 0|$ with $O(n + a)$ circuit depth. Then, $V = (I_2 \otimes U_\psi) U_0$ is a $(1, 2, 0)$-block-encoding for $U_\psi |0\rangle\langle 0|$, with $O(T_\psi + a + n)$ circuit complexity. Noting that $(\langle 0|_2 \otimes I_{a+n}) V (|0\rangle_2 \otimes I_{a+n}) = U_\psi |0\rangle\langle 0|$, then $(\langle 0|_{2+a} \otimes I_n) V (|0\rangle_{2+a} \otimes I_n) = (\langle 0|_a \otimes I_n) U_\psi |0\rangle\langle 0| (|0\rangle_a \otimes I_n) = |\phi\rangle\langle 0|_a$, so

$$\||\phi\rangle\langle 0|_a - \langle 0|_{2+a} \otimes I_n) V (|0\rangle_{2+a} \otimes I_n)\|_2 = 0. \qquad (\text{B.9})$$

Thus, we have a $(1, a + 2, 0)$-block-encoding of $|\phi\rangle\langle 0|_a = \mathcal{N}_\phi |\Phi\rangle\langle 0|$. This object has singular value $\mathcal{N}_\phi$. Thus, we want to apply a polynomial approximation to this block-encoding, such that the error of the polynomial approximation is at most $\epsilon_1$ on the interval $[\mathcal{N}_\phi, 1]$. From Corollary 6 of Low & Chuang (2017), we know that there exists an odd polynomial $P_k(x)$ with degree $k \in O(\frac{1}{\tau} \log(1/\epsilon_1))$ such that

$$\max_{x \in [-1, -\frac{\tau}{2}] \cup [\tau/2, 1]} |P_k(x) - \text{sign}(x)| \leq \epsilon_1 \qquad (\text{B.10})$$

and $\max_{x \in [-1,1]} |P_k(x)| \leq 1$. Since $\mathcal{N}_\phi \geq \frac{1}{2\alpha} \geq \frac{1}{2\alpha'}$, we can set $\tau = \frac{1}{2\alpha'}$, guaranteeing that $P(\mathcal{N}_\phi) \geq 1 - \epsilon_1$. Consequently, we can invoke quantum singular value transformation (QSVT) (Gilyén et al., 2019) with $P_k$, yielding $V_f$ a $(1, a+4, \epsilon_1)$-block-encoding for $P(\mathcal{N}_\phi |\Phi\rangle\langle 0|) = c |\Phi\rangle\langle 0|$,

where $1 \geq c \geq 1 - \epsilon_1$. Moreover, $V_f$ has $O(\frac{1}{\alpha'} \log(1/\epsilon_1)(T_\psi + a + n))$ circuit complexity. Noting that

$$\||\psi\rangle\langle 0| - P(\mathcal{N}_\phi |\Phi\rangle\langle 0|)\|_2 = \||\psi\rangle\langle 0| - |\Phi\rangle\langle 0| + |\Phi\rangle\langle 0| - c|\Phi\rangle\langle 0|\|_2 \tag{B.11}$$

$$\leq \||\psi\rangle\langle 0| - |\Phi\rangle\langle 0|\|_2 + \||\Phi\rangle\langle 0| - c|\Phi\rangle\langle 0|\|_2 \tag{B.12}$$

$$\leq \||\psi\rangle_n - |\Phi\rangle_n\|_2 + \epsilon_1. \tag{B.13}$$

Moreover,

$$\||\psi\rangle_n - |\Phi\rangle_n\|_2 \leq \||\psi\rangle_n - \alpha|\phi\rangle_n\|_2 + \|\alpha|\phi\rangle_n - \frac{1}{\mathcal{N}_\phi}|\phi\rangle_n\|_2 \tag{B.14}$$

$$\leq \epsilon_0 + \frac{1}{\mathcal{N}_\phi}\|\alpha\mathcal{N}_\phi|\phi\rangle_n - |\phi\rangle_n\| = \epsilon_0 + \frac{|\alpha\mathcal{N}_\phi - 1|}{\mathcal{N}_\phi}\||\phi\rangle_n\|_2 \tag{B.15}$$

$$\leq \epsilon_0 + |\alpha\mathcal{N}_\phi - 1|. \tag{B.16}$$

Moreover, using the reverse triangle inequality with $\||\psi\rangle_n - \alpha|\phi\rangle_n\|_2 \leq \epsilon_0$, we get $|1 - \alpha\||\phi\rangle_n\|_2| = |1 - \alpha\mathcal{N}_\phi| \leq \epsilon_1$, which implies that $1 - \epsilon_0 \leq \alpha\mathcal{N}_\phi \leq 1 + \epsilon_0$. Consequently, $|\alpha\mathcal{N}_\phi - 1| \leq \epsilon_0$, and so

$$\||\psi\rangle_n - |\Phi\rangle_n\|_2 \leq 2\epsilon_0. \tag{B.17}$$

Thus,

$$\||\psi\rangle\langle 0| - P(\mathcal{N}_\phi |\Phi\rangle\langle 0|)\|_2 \leq 2\epsilon_0 + \epsilon_1. \tag{B.18}$$

Moreover, since $V_f$ is a $(1, a + 4, \epsilon_1)$-block-encoding for $P(\mathcal{N}_\phi |\Phi\rangle\langle 0|)$,

$$\|P(\mathcal{N}_\phi |\Phi\rangle\langle 0|) - (\langle 0|_{a+4} \otimes I_n)V_f(|0\rangle_{a+4} \otimes I_n)\|_2 \leq \epsilon_1. \tag{B.19}$$

Thus,

$$\||\psi\rangle\langle 0| - (\langle 0|_{a+4} \otimes I_n)V_f(|0\rangle_{a+4} \otimes I_n)\|_2 \leq 2(\epsilon_0 + \epsilon_1). \tag{B.20}$$

$\square$

Sometimes it is necessary to increase the norm of the vector encoded in the subspace of a VE. This is equivalent to multiplying all of the entries in the encoded vector by a constant with value greater than or equal to one. The following lemma achieves the opposite: it allows the norm of the encoded vector to be shrunk by an arbitrarily large amount. This is equivalent to dividing all the entries in the encoded vector by a constant greater than or equal to one. It is worth noting that the following result is trivial and can almost certainly be further optimized, e.g., by removing the additional ancillary qubits added.

**Lemma B.9** (Vector De-Amplification). *Let $\tau \geq 1$, $\alpha \geq 1$, $\epsilon \geq 0$. Given $U_\psi$ an $(\alpha, a, \epsilon)$-VE for $|\psi\rangle_n$, with circuit complexity $O(T)$, we can obtain $U'_\psi$ an $(\alpha\tau, a + 2, \epsilon)$-VE for $|\psi\rangle_n$ with circuit complexity $O(T + a)$.*

*Proof.* Let $|\phi_j\rangle_n := (\langle 0|_a \otimes I_n)U_\psi|0\rangle_{a+n}$. Then, note that $U_\psi|0\rangle_{a+n} = \sum_{j=0}^{2^a-1} |j\rangle_a \otimes |\phi_j\rangle_n$. By Definition 3, we know that $\||\psi\rangle_n - \alpha|\phi_0\rangle_n\| \leq \epsilon$.

We introduce two single-qubit ancillas as the most significant bits, and then apply a multiple-controlled $X$ gate (with $a$ controls each activated by the 0 state of each of the previous $a$ ancilla qubits) targeting the first newly added ancilla qubit. Using Saeedi & Pedram (2013) this can be implemented with $O(a)$ two-qubit gates. We then apply a controlled $R_{1/\tau^2}$ (as per Definition B.1) gate targeting the second new ancilla qubit, controlled on the first new ancilla. This yields the state,

$$|1\rangle_1\left(\frac{1}{\tau}|0\rangle_1 + \sqrt{1 - \frac{1}{\tau^2}}|1\rangle_1\right)|0\rangle_a|\phi_0\rangle_n + |0\rangle_1|0\rangle_1 \sum_{j=1}^{2^a-1} |j\rangle_a|\phi_j\rangle_n. \tag{B.21}$$

We then apply a $X$ gate to the first ancilla qubit, and we call the $2 + a$-qubit unitary containing all the preceding operations $V$. Then, $U'_\psi := (V \otimes I_n)(I_2 \otimes U_\psi)$. Simple analysis thus shows that $(\langle 0|_{2+a} \otimes I_n)U'_\psi|0\rangle_{2+a+n} = |\phi_0\rangle_n/\tau$. Then,

$$\||\psi\rangle_n - \alpha\tau(\langle 0|_{2+a} \otimes I_n)U'_\psi|0\rangle_{2+a+n}\|_2 = \||\psi\rangle_n - \alpha|\phi_0\rangle_n\| \leq \epsilon. \tag{B.22}$$

$\square$

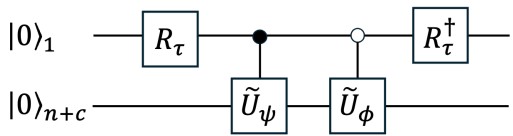

Figure 3: **Circuit for addition of VE encoded vectors.** Given two unitary matrices, $U_\psi$ which is a $(\alpha, a, \epsilon_0)$-VE for the $n$-qubit state $|\psi\rangle$, and $U_\phi$ which is a $(\beta, b, \epsilon_1)$-VE for the $n$-qubit state $|\phi\rangle$, define $c := \max(a, b)$. We define $\tilde{U}_\psi$ by appropriately tensoring $U_\psi$ with $I_{c-a}$ and we define $\tilde{U}_\phi$ by appropriately tensoring $U_\phi$ with $I_{c-b}$, such that $\tilde{U}_\psi$ and $\tilde{U}_\phi$ both act on $n + c$ qubits. Then, the given circuit yields a VE of the sum of the encoded vectors, as shown in Lemma 1.

**Definition B.1** (Real Rotation Single Qubit Gate). *Let* $0 \le \tau \le 1$. *Then, define the following single-qubit gate:*

$$R_\tau := \begin{pmatrix} \sqrt{\tau} & -\sqrt{1-\tau} \\ \sqrt{1-\tau} & \sqrt{\tau,} \end{pmatrix}. \tag{B.23}$$

***Proof of Lemma 1*** *(Vector Sum)*. This result follows using a common techniques, see e.g., LCU (Childs & Wiebe, 2012), or the sum of block-encodings result (Gilyén et al., 2019). As per Figure 3, we will augment $U_\psi$ and $U_\phi$ so that they both act on $c = \max(a + b)$ ancilla qubits. Then, define the $n + c$ qubit states, $|\tilde{\psi}\rangle_{n+c} := U_\psi |0\rangle_{n+c}$. We will drop the subscripts on these states for the rest of the proof, as their dimension is clear from the context. This block-encoding will be obtained with the circuit shown in Figure 3, and so we will now analyze the action of that circuit. First, we start with the state $|0\rangle_{1+n+c}$, which we will write as $|0\rangle|0\rangle$, where the first register has one qubit, and the second register has the remaining $n+c$ qubits. We then apply $R_\tau$ (as defined in Definition B.1) to the first qubit, yielding the state $(\sqrt{\tau}|0\rangle + \sqrt{1-\tau}|1\rangle)|0\rangle$. Next, we apply the controlled $U_\psi$ and $U_\phi$ gates, yielding, $\sqrt{\tau}|0\rangle|\tilde{\psi}\rangle + \sqrt{1-\tau}|1\rangle|\tilde{\phi}\rangle$. Next, we apply $R_\tau^\dagger = \begin{pmatrix} \sqrt{\tau} & \sqrt{1-\tau} \\ -\sqrt{1-\tau} & \sqrt{\tau} \end{pmatrix}$ on the first qubit, yielding the output of the new VE, $V|0\rangle = |0\rangle(\tau|\tilde{\psi}\rangle + (1-\tau)|\tilde{\phi}\rangle) + \sqrt{\tau(1-\tau)}|1\rangle(|\tilde{\phi}\rangle - |\tilde{\psi}\rangle)$. Define $|E_\psi\rangle := |\psi\rangle - \alpha(\langle 0|^{\otimes(c)} \otimes I_n)|\tilde{\psi}\rangle$ and note that $\||E_\psi\rangle\|_2 \le \epsilon_0$. Similarly define $|E_\phi\rangle$, and note that $\||E_\phi\rangle\|_2 \le \epsilon_1$. As a result, we can determine the properties of this VE by bounding the following,

$$\left\| |\bar{\Gamma}\rangle - \frac{1}{\mathcal{N}}(\langle 0|^{\otimes(1+c)} \otimes I_n)V|0\rangle_{1+c+n} \right\|_2 \tag{B.24}$$

$$= \left\| |\bar{\Gamma}\rangle - \frac{1}{\mathcal{N}}(\langle 0|^{\otimes c} \otimes I_n)(\tau|\tilde{\psi}\rangle + (1-\tau)|\tilde{\phi}\rangle) \right\|_2 \tag{B.25}$$

$$= \left\| |\bar{\Gamma}\rangle - \frac{1}{\mathcal{N}}\left(\frac{\tau}{\alpha}(|\psi\rangle - |E_\psi\rangle) + \frac{1-\tau}{\beta}(|\phi\rangle - |E_\phi\rangle)\right) \right\|_2 \tag{B.26}$$

$$= \left\| \frac{1}{\mathcal{N}}\left(\frac{\tau}{\alpha}|E_\psi\rangle + \frac{1-\tau}{\beta}|E_\phi\rangle\right) \right\|_2 \le \frac{1}{\mathcal{N}}\left(\frac{\tau\epsilon_0}{\alpha} + \frac{(1-\tau)\epsilon_1}{\beta}\right) \tag{B.27}$$

$$\le \frac{1}{\mathcal{N}}\left(\frac{\epsilon_0}{\alpha} + \frac{\epsilon_1}{\beta}\right) \le \frac{\epsilon_0 + \epsilon_1}{\mathcal{N}}. \tag{B.28}$$

where the final inequality comes from the definition of a VE imposing that $\alpha \ge 1$ and $\beta \ge 1$. Thus, the unitary circuit $V$ is a $(\mathcal{N}^{-1}, 1 + a + b, \mathcal{N}^{-1}(\epsilon_0 + \epsilon_1))$-VE for $|\bar{\Gamma}\rangle$. $\square$

***Proof of Lemma 2*** *(Matrix Vector Product)*. We now require a result allowing for matrix-vector products with our vector-encodings. This result is essentially a special case of the product of the standard product of block-encodings result (Lemma 53 of Gilyén et al. (2019)). As a result, the following proof closely follows that in Gilyén et al. (2019).

In this lemma, again following the notation of Gilyén et al. (2019) for tensor products, it is assumed that $U_A$ and $U_\psi$ act trivially on the other's ancillas. To be explicit, the tensor products in $(I_b \otimes$

$U_A)(I_a \otimes U_\psi)$ use a special definition only in this lemma. Let $\mathcal{N} := \||A|\psi\rangle_n\|_2$. We wish to upper-bound,

$$\xi := \left\| \frac{A|\psi\rangle_n}{\mathcal{N}} - \frac{\alpha\beta}{\mathcal{N}}(\langle 0|_{a+b} \otimes I_n)(I_b \otimes U_A)(I_a \otimes U_\psi)|0\rangle_{a+b+n} \right\|_2 \tag{B.29}$$

$$= \frac{1}{\mathcal{N}} \left\| A|\psi\rangle_n - \alpha\beta(\langle 0|_{a+b} \otimes I_n)(I_b \otimes U_A)(I_a \otimes U_\psi)(|0\rangle_{a+b} \otimes I_n)|0\rangle_n \right\|_2 \tag{B.30}$$

Then, directly from the proof of Lemma 53 in Gilyén et al. (2019),

$$\xi = \frac{1}{\mathcal{N}} \left\| A|\psi\rangle_n - \alpha\beta \left[ (\langle 0|_a \otimes I_n)U_A(|0\rangle_a \otimes I_n) \right] \left[ (\langle 0|_b \otimes I_n)U_\psi(|0\rangle_b \otimes I_n) \right] |0\rangle_n \right\|_2 \tag{B.31}$$

Let $\tilde{A} := \alpha(\langle 0|_a \otimes I_n)U_A(|0\rangle_a \otimes I_n)$ and let $|\tilde{\psi}\rangle := \beta(\langle 0|_b \otimes I_n)U_\psi(|0\rangle_{b+n})$. Then,

$$\xi = \frac{1}{\mathcal{N}} \left\| A|\psi\rangle_n - \tilde{A}|\tilde{\psi}\rangle_n \right\|_2 = \frac{1}{\mathcal{N}} \left\| A|\psi\rangle_n - \tilde{A}|\psi\rangle_n + \tilde{A}|\psi\rangle_n - \tilde{A}|\tilde{\psi}\rangle_n \right\|_2 \tag{B.32}$$

$$\leq \frac{1}{\mathcal{N}} \left( \left\| A - \tilde{A} \right\|_2 + \left\| \tilde{A} \right\|_2 \left\| |\psi\rangle_n - |\tilde{\psi}\rangle_n \right\|_2 \right) \tag{B.33}$$

Noting that $\left\| \tilde{A} \right\|_2 \leq \alpha$, we then get

$$\xi \leq (\epsilon_0 + \alpha\epsilon_1)/\mathcal{N}. \tag{B.34}$$

Consequently, $(I_b \otimes U_A)(I_a \otimes U_\psi)$ gives a $(\alpha\beta/\mathcal{N}, a+b, (\epsilon_0 + \alpha\epsilon_1)/\mathcal{N})$-VE for $A|\psi\rangle_n/\mathcal{N}$.

$\square$

In the following lemma we derive a technical result handling the case where you have a vector encoding for some vector $|\psi\rangle$, and another vector of interest $|\phi\rangle$ is sub-encoded as $|\psi\rangle = \begin{pmatrix} |\phi\rangle/\beta \\ \cdot \end{pmatrix}$. Our result also handles the case where each vector is imperfectly encoded (i.e., encoded with error).

**Lemma B.10** (Vector Sub-Encodings). *Let $m, n$ be integers such that $m > n$. Let $U_\psi$ be an $(\alpha, a, \epsilon)$-VE for $|\psi\rangle_m$, and let $|\psi\rangle_m \approx V_\phi|0\rangle_m$ (precisely, $\||\psi\rangle_m - V_\phi|0\rangle_m\|_2 \leq \gamma$), where $V_\phi$ is a $(\beta, m-n, \delta)$-VE for $|\phi\rangle_n$. Then, $U_\psi$ is an $(\alpha\beta, a+m-n, \delta + \beta(\epsilon+\gamma))$-VE for $|\phi\rangle_n$.*

*Proof.* Let $b = m - n$. First, define $|E_\psi\rangle_m := |\psi\rangle_m - \alpha\left(\langle 0|_a \otimes I_m\right)U_\psi|0\rangle_{a+m}$, and $|E_\phi\rangle_n := |\phi\rangle_n - \alpha\left(\langle 0|_b \otimes I_n\right)U_\psi|0\rangle_{b+n}$. By Definition 3, $\||E_\psi\rangle_m\|_2 \leq \epsilon$ and $\||E_\phi\rangle_n\|_2 \leq \delta$. Let $|E_v\rangle_m := |\psi\rangle_m - V_\phi|0\rangle_m$. Now observe,

$$(\langle 0|_b \otimes I_n)\left(\langle 0|_a \otimes I_m\right)U_\psi|0\rangle_{a+m} = (\langle 0|_b \otimes I_n)\left(|\psi\rangle_m - |E_\psi\rangle_m\right)/\alpha \tag{B.35}$$

$$= (\langle 0|_b \otimes I_n)\left(V_\phi|0\rangle_m + |E_v\rangle_m - |E_\psi\rangle_m\right)/\alpha \tag{B.36}$$

$$= \left((|\phi\rangle_n - |E_\phi\rangle_n)/\beta + (\langle 0|_b \otimes I_n)(|E_v\rangle_m - |E_\psi\rangle)\right)/\alpha. \tag{B.37}$$

Consequently, since $(\langle 0|_b \otimes I_n)(\langle 0|_a \otimes I_m) = \langle 0|_{a+b} \otimes I_n$,

$$\||\phi\rangle_n - \alpha\beta(|0\rangle_{a+b} \otimes I_n)U_\psi|0\rangle_{a+b+n}\|_2 \tag{B.38}$$

$$\leq \||E_\phi\rangle_n\|_2 + \beta \||E_\psi\rangle_m\|_2 + \beta \||E_v\rangle_m\|_2 \leq \delta + \beta(\epsilon + \gamma). \tag{B.39}$$

$\square$

**Lemma B.11** (Tracing Out Qubits in Vector Sub-Encodings). *Let $U$ be an $(\alpha, a, \epsilon)$-VE for $|0\rangle_b|\psi\rangle_n$. Then, $U$ is an $(\alpha, a+b, \epsilon)$-VE for $|\psi\rangle_n$.*

*Proof.* Let $|E\rangle_{b+n} := |0\rangle_b|\psi\rangle_n - \alpha(\langle 0|_a \otimes I_{n+b})U|0\rangle_{a+b+n}$. Since $\langle 0|_{a+b} \otimes I_n = (\langle 0|_b \otimes I_n)(\langle 0|_a \otimes I_{b+n})$, $(\langle 0|_{a+b} \otimes I_n)U|0\rangle_{a+b+n} = \frac{1}{\alpha}(|\psi\rangle_n - (\langle 0|_b \otimes I_n)|E\rangle_{b+n})$. Thus,

$$\||\psi\rangle_n - \alpha(\langle 0|_{a+b} \otimes I_n)U|0\rangle_{a+b+n}\|_2 = \|(\langle 0|_b \otimes I_n)|E\rangle_{b+n}\|_2 \leq \epsilon. \tag{B.40}$$

$\square$

***Proof of Lemma 3 (Vector Tensor Product).*** This result closely follows the derivation of the tensor product of block-encodings (Lemma B.2), which was a rederivation of Lemma 1 of Camps & Van Beeumen (2020).

$U_\psi$ acts on an $a$-qubit ancilla register and a $n$-qubit main register, while $U_\phi$ acts on an $b$-qubit ancilla register and a $m$-qubit main register.

As per Lemma B.2, define $\Pi$ to swap the $n$-qubit register with the $b$-qubit register acting trivially on the other two registers. Again, $\Pi$ has a circuit depth bounded by $O(\max(n/b, b/n)) \in O(\max(n, b))$. Then, $(\langle 0|_{a+b} \otimes I_{n+m})\Pi^\dagger = (\langle 0|_a \otimes I_n) \otimes (\langle 0|_b \otimes I_m)$. Let $V = \Pi^\dagger(U_\psi \otimes U_\phi)$. Let $|E_\psi\rangle_n = |\psi\rangle_n - \alpha(\langle 0|_a \otimes I_n)U_\psi|0\rangle_{a+n}$ and $|E_\phi\rangle_m = |\phi\rangle_m - \beta(\langle 0|_b \otimes I_m)U_\phi|0\rangle_{b+m}$. Then,

$$(\langle 0|_{a+b} \otimes I_{n+m})\Pi^\dagger(U_\psi \otimes U_\phi)|0\rangle_{a+b+n+m} = \frac{1}{\alpha\beta}(|\psi\rangle_n - |E_\psi\rangle_n) \otimes (|\phi\rangle_m - |E_\phi\rangle_m), \quad \text{(B.41)}$$

and so,

$$\||\psi\rangle_n|\phi\rangle_m - \alpha\beta(\langle 0|_{a+b} \otimes I_{n+m})\Pi(U_\psi \otimes U_\phi)|0\rangle_{a+b+n+m}\|_2 \leq \epsilon + \delta + \epsilon\delta. \quad \text{(B.42)}$$

$\square$

***Proof of Lemma 4 (Vector Concatenation).*** We now present the proof of a simple result on the concatenation of vectors stored in VEs. This result follows from a simple modification of LCU (Childs & Wiebe, 2012). In essence, given a set of $D = 2^d$ vectors $\{|\psi_j\rangle_n\}_j$, we first create vector encodings of $\{|j\rangle_d|\psi_j\rangle_n\}_j$ and then take the resulting sum of the encoded vectors following LCU, yielding an encoding of $(\langle\psi_0|_n \quad \cdots \quad \langle\psi_{D-1}|_n)^\dagger$.

For all $j$, define $|E_{\psi_j}\rangle_n := |\psi_j\rangle_n - \alpha(\langle 0|_a \otimes I_n)U_i|0\rangle_{a+n}$.

First, let $j$ be $d$ bits, and let $j = j_0 j_1 \ldots j_{d-1}$. Define $X_j := X^{j_0} \otimes X^{j_1} \otimes \ldots \otimes X^{j^{d-1}}$. Note that $|j\rangle_d = X_j|0\rangle_d$, and thus that $X_j$ is a $(1, 0, 0)$-VE for $|j\rangle_d$. Then, we can invoke Lemma 3 with $U_j$ and $X_j$ to obtain $V_j$, an $(\alpha_j, a, \epsilon)$-VE for $|j\rangle_d|\psi_j\rangle_n$ with $O(T + n)$ circuit complexity. Moreover, by inspecting Lemma 3, we find that $(\langle 0|_a \otimes I_{n+d})V_j|0\rangle_{a+d+n} = \frac{1}{\alpha_j}(|j\rangle_d|\psi_j\rangle_n - |j\rangle_d|E_{\psi_j}\rangle_n)$.

Additionally, define $S := \sum_{j=0}^{D-1} |j\rangle\langle j|_d \otimes V_j$. This can be implemented by a sequence of $O(D)$ multi-controlled gates, each enacting $V_j$ when the control register is $|j\rangle_d$ (in the standard fashion of LCU (Childs & Wiebe, 2012)). First, note that by using Saeedi & Pedram (2013) a multiple-controlled gate with $O(d)$ controls can be split into a sequence of $O(d)$ single and two-qubit gates. By splitting each of the $d$ control qubits into $a + d + n$ copies (with $O(\log(a + d + n))$ depth), we can control each gate in each layer of $U_j$ in parallel with $O(d)$ circuit depth. Since these ancillas can be uncomputed and traced out, we ignore them in the complexity analysis. Thus, each multi-controlled $V_j$ gate can be decomposed into a sequence of $O(dT)$ single and two-qubit gates. Thus, $S$ has a total circuit depth of $O(dDT)$. Let $\hat{H} := H^{\otimes d} \otimes I_{d+n+a}$. Using $\langle 0|_{a+d} \otimes I_{n+d} = (\langle 0|_d \otimes I_{n+d})(I_d \otimes \langle 0|_a \otimes I_{d+a})$,

$$(\langle 0|_{d+a} \otimes I_{n+d})\hat{H}S\hat{H}|0\rangle_{2d+a+n} \quad \text{(B.43)}$$

$$= (\langle +|_d \otimes I_{n+d})(I_d \otimes \langle 0|_a \otimes I_{n+d}) \sum_{j=0}^{D-1} (|j\rangle\langle j|_d \otimes V_j)|+\rangle_d|0\rangle_{a+d+n} \quad \text{(B.44)}$$

$$= \frac{1}{D} \sum_{j=0}^{D-1} \frac{|j\rangle_d|\psi_j\rangle_d - |j\rangle_d|E_{\psi_j}\rangle_n}{\alpha_j}. \quad \text{(B.45)}$$

Then, noting that $\mathcal{N}^2 = \sum_{j=0}^{D-1} \frac{1}{\alpha_j^2}$, and that $\left\|\sum_{j=0}^{D-1} |j\rangle_d|E_{\psi_j}\rangle_n/\alpha_j\right\|_2 \leq \mathcal{N}\epsilon$,

$$\left\|\frac{|\Psi\rangle_{d+n}}{\mathcal{N}} - \frac{D}{\mathcal{N}}(\langle 0|_{d+a} \otimes I_{n+d})\hat{H}S\hat{H}|0\rangle_{2d+a+n}\right\|_2 = \frac{1}{\mathcal{N}}\left\|\sum_{j=0}^{D-1} |j\rangle_d|E_{\psi_j}\rangle_n/\alpha_j\right\|_2 \leq \epsilon. \quad \text{(B.46)}$$

Thus, $\hat{H}S\hat{H}$ is a $(D/\mathcal{N}, d + a, \epsilon)$ for $\frac{|\Psi\rangle_{d+n}}{\mathcal{N}}$ with $O(dDT)$ circuit complexity. $\square$

## B.1 GENERAL MATRIX-VECTOR-SQUARED PRODUCT

In this subsection, we will derive a procedure which given an *arbitrary* matrix $W$ and quantum state $|\psi\rangle$, allows for a state proportional to the product of $W(|\psi\rangle)^2$ to be obtained with complexity *independent* of the Frobenius norm (and thus rank), and sparsity, of $W$. To the best of our knowledge, this is the first result which allows such a product without either a rank or sparsity condition on $W$. The key insight is to avoid ever constructing a block-encoding of the operator $W$, and directly query its columns weighted by the entries of the vector it is being applied to. In particular, at a high-level we construct two objects. Define the columns of $W = (\boldsymbol{w}_0 \quad \ldots \quad \boldsymbol{w}_{N-1})$, define the column norms $a_j := \|\boldsymbol{w}_j\|_2$, and the normalized versions of the columns $|w_j\rangle_n = \boldsymbol{w}_j/a_j$. Additionally, define the state we are applying it to as $|\psi\rangle_n = \sum_j \psi_j |j\rangle_n$. First, we construct the normalized state $\sum_j \psi_j |j\rangle_n |w_j\rangle_n$. Clearly, this object has no Frobenius norm dependence. We would like to map all the vectors in the first register to the $|0\rangle$ state so that we have something resembling the matrix-vector product, and to do this we construct another operator. Note that the matrix $Q = \begin{pmatrix} a_0 I_n & \ldots & a_{N-1} I_n \\ & \mathbf{0} & \end{pmatrix}$ (i.e., the first $N$ rows are non-zero, and the rest are all zero) when applied to $|\phi\rangle_{2n} = \sum_j \psi_j |j\rangle_n |w_j\rangle_n$ yields $Q|\phi\rangle_{2n} = |0\rangle_n \otimes (W|\psi\rangle_n)$. However, this object has a spectral norm $\Omega(\|W\|_F)$. Instead, we define $M := \begin{pmatrix} a_0 \psi_0 I_n & \ldots & a_{N-1} \psi_{N-1} I_n \\ & \mathbf{0} & \end{pmatrix}$ and note that $M$ can be shown to have $\|M\|_2 \le 1$, and moreover, we subsequently show how a block-encoding of this operator can be efficiently obtained. Consequently, since $M|\phi\rangle_{2n} = |0\rangle_n \otimes (W(|\psi\rangle_n)^2)$, the result follows. The rest of this section simply derives the ingredients necessary to rigorously prove this intuition.

**Definition B.2** ($R_Y(t)$ Gate). *Let $t \in \mathbb{R}$, and let $Y$ be the standard single-qubit Pauli-Y gate. Then, define*

$$R_Y(t) := e^{-itY} = \cos(t)I - i\sin(t)Y = \begin{pmatrix} \cos(t) & -\sin(t) \\ \sin(t) & \cos(t) \end{pmatrix}. \tag{B.47}$$

For completeness, we will now present a standard result allowing one to transfer digitally represented information to the amplitudes of a quantum state.

**Lemma B.12** ($CR_Y(t)$ Gate). *Let $t \in \mathbb{R}$. Let $Y$ be a standard Pauli-Y gate. Let $|a\rangle_d$ be a d-bit standard basis vector, and let $|\psi\rangle_1$ be an arbitrary single-qubit quantum state. Then, we can define the gate $CR_Y(t)$ by the following action,*

$$CR_Y(t)|\psi\rangle_1 |a\rangle_d = (e^{-iatY}|\psi\rangle_1)|a\rangle_d. \tag{B.48}$$

*In the event that $|\psi\rangle_1 = |0\rangle_1$, this action can be simplified to*

$$CR_Y(t)|0\rangle_1 |a\rangle_d = (\cos(at)|0\rangle_1 + \sin(at)|1\rangle_1)|a\rangle_d. \tag{B.49}$$

*Moreover, the $CR_Y(t)$ gate is implemented with $O(d)$ circuit depth.*

*Proof.* This is a standard result. This proof is included for completeness, and follows the one in Rattew & Koczor (2022). Let $D = 2^d$. First, note that $CR_Y(t) = \sum_{a=0}^{D-1} e^{-iatY} \otimes |a\rangle\langle a|$. Additionally, let $a = a_{d-1}a_{d-2}\ldots a_1 a_0 = a_{d-1}2^{d-1} + \ldots + a_1 2 + a_0$. Then,

$$e^{-iatY} = e^{-i(a_{d-1}2^{d-1}+\ldots+a_1 2+a_0)tY} = e^{-ia_{d-1}2^{d-1}tY} \cdot \ldots \cdot e^{-ia_1 tY} e^{-ia_0 tY}. \tag{B.50}$$

Then, $CR_Y(t)$ can be implemented by applying a sequence of $d$ controlled $e^{-i2^j tY}$ gates (Definition B.2), targeting the first register, controlled on the $j^{th}$ bit of the second register. $\square$

We now present a result on obtaining a block-encoding of an arbitrary diagonal matrix whose entries are stored in QRAM. This is essentially a special case of Lemma 48 of Gilyén et al. (2019), but by considering this special case moderate improvements in complexity can be obtained.

**Lemma B.13** (Quantum Block-Encoding of Diagonal Matrices from QRAM). *Let $N = 2^n$. We are given a set of $N$ real coefficients, $\{a_j\}_j$ such that $\forall j, |a_j| \le 1$. Assume that each $a_j$ can be represented exactly in a binary encoding with d-bits of precision, and define $D = 2^d$. Define*

$b_j := \arccos(a_j)D/\pi$, *and for simplicity assume that each $b_j$ can also be implemented with exactly $d$-bits of precision[4], and note that $b_j \in [D]$. Assume that we are given an oracle, implemented via QRAM, such that $U|0\rangle_d|j\rangle_n = |b_j\rangle_d|j\rangle_n$. Then, we can obtain $U_A$, a $(1, d+1, 0)$-block-encoding for $A = diag(a_0, \ldots, a_{N-1})$, with $O(dn)$ circuit depth.*

*Proof.* Define the circuit $V := (I_1 \otimes U^\dagger)(CR_Y(\frac{\pi}{D}) \otimes I_n)(I_1 \otimes U)$, with $CR_Y(\frac{\pi}{D})$ defined as per Lemma B.12. First, since for any $|\phi\rangle$ and basis vector $|j\rangle$, $|\phi\rangle \otimes |j\rangle\langle j| = (|\phi\rangle|j\rangle)\langle j|$, observe that

$$(I_1 \otimes U)(|0\rangle_{d+1} \otimes I_n) = \sum_{j=0}^{N-1} [(I_1 \otimes U)|0\rangle_1|0\rangle_d|j\rangle_n] \langle j|_n = \sum_{j=0}^{N-1} (|0\rangle_1|b_j\rangle_d|j\rangle_n)\langle j|_n. \quad \text{(B.51)}$$

Then, since $\cos(b_j\frac{\pi}{D}) = \arccos(a_j)$,

$$(CR_Y(\frac{\pi}{D}) \otimes I_n)(I_1 \otimes U)(|0\rangle_{d+1} \otimes I_n) = \sum_{j=0}^{N-1} \left( (a_j|0\rangle_1 + \sqrt{1-a_j^2}|1\rangle_1)|b_j\rangle_d|j\rangle_n \right) \langle j|_n. \quad \text{(B.52)}$$

Then, since $(\langle 0|_{d+1} \otimes I_n)(I_1 \otimes U^\dagger) = [(I_1 \otimes U)(|0\rangle_{d+1} \otimes I_n)]^\dagger = \sum_{j=0}^{N-1} |j\rangle_n(\langle 0|_1\langle b_j|_d\langle j|_n)$, we readily find that

$$(\langle 0|_{d+1} \otimes I_n)V(|0\rangle_{d+1} \otimes I_n) = \sum_{j=0}^{N-1} a_j|j\rangle\langle j| = diag(a_0, \ldots, a_{N-1}) = A. \quad \text{(B.53)}$$

Thus, $V$ is a $(1, d+1, 0)$-block-encoding for $A$. The circuit depth of implementing $U$ is the depth of making a QRAM query, and is thus $O(d \log N) = O(nd)$ (see Definition A.1). The cost of implementing the $CR_Y$ gate is simply $O(d)$ as per Lemma B.12, and thus the overall circuit complexity of this block-encoding is $O(nd)$. □

In the case where each $a_j \in \mathbb{C}$, the complex and real parts need to be specified separately. A diagonal block-encoding of the real and imaginary parts can then be obtained using Lemma B.13, and can then be summed by adding an ancilla to obtain a $(2, d+2, 0)$-block-encoding with the same overall circuit complexity. One might wonder why, given a QRAM assumption, a state-preparation unitary yielding a state proportional to $\sum_j a_j|j\rangle$ can't be used instead, in combination with the diagonal block-encoding of state amplitudes result of Rattew & Rebentrost (2023). If each $a_j$ represent the column norm of some matrix $W$, doing so would result in a normalization factor of $\left\| \sum_j a_j|j\rangle \right\|_2 = \sqrt{\sum_j |a_j|^2} = \|W\|_F$, yielding a Frobenius norm-dependence which this approach avoids.

The following data-structure is useful in situations where you are willing to pay a pre-processing cost linear (up to polylogarithmic factors) in the number of non-zero matrix elements, but want a fast algorithm at runtime. This is the case with accelerating neural network inference. The following data structure is very similar to the one given in Kerenidis & Prakash (2017).

**Definition B.3** (Preprocessed Matrix QRAM Data Structure). *Let $N = 2^n$, and let $D = 2^d$.*

*Let $W \in \mathbb{C}^{N \times N}$ and let $\|W\|_2 \leq 1$. Let the columns of $W$ be represented as $W = (\boldsymbol{w}_0 \ \ldots \ \boldsymbol{w}_{N-1})$. Additionally, define $|w_j\rangle = \boldsymbol{w}_j/\|\boldsymbol{w}_j\|_2$, and $a_j = \|\boldsymbol{w}_j\|$. Let $b_j := \arccos(a_j)D/\pi$. For simplicity, we assume that $b_j$ can be exactly written with $d$-bits, and thus that $b_j$ will be an integer between $[0, D-1]$. We say we have access to a Preprocessed QRAM Data Structure for $W$ if we have a QRAM oracle $U_W$ (as per Definition A.2) such that*

$$U_W|j\rangle_n|0\rangle_n = |j\rangle_n|w_j\rangle_n, \quad \text{(B.54)}$$

*and we also have access to a QRAM yielding the mapping,*

$$U_A|0\rangle_d|j\rangle_n = |b_j\rangle_d|j\rangle_n. \quad \text{(B.55)}$$

*$U_W$ can be implemented with $O(\log^2 N)$ circuit depth, and with $\tilde{O}(N^2)$ total qubits (as per Definition A.2). $U_A$ can be implemented with $O(d \log N)$ circuit depth, and with $\tilde{O}(dN)$ total qubits (as per Definition A.1).*

---

[4]In practice this will result in an additional logarithmic source of error, which we are neglecting, as it is akin to finite-precision arithmetic error which is usually neglected in classical algorithm analysis.

We are now ready to present a somewhat surprising result on matrix-vector multiplication with arbitrary (potentially full-rank and dense) matrices and the element-wise square of a given vector. The following uses ideas similar to importance-sampling.

**Theorem B.1** (Product of Arbitrary Matrix with a Vector Element-wise Squared). *Let $N = 2^n$. We are given a matrix $W \in \mathbb{C}^{N \times N}$ through the data-structure in Definition B.3. Let $d$ be the number of bits required to represent the function of the column norms of $W$, $b_j$, as per Definition B.3. Additionally, we are given the unitary $U_\psi$ with circuit complexity $O(T_\psi)$, a $(\alpha, a, \epsilon)$-VE for the quantum state $|\psi\rangle_n$. Define the function $g : \mathbb{C} \mapsto \mathbb{R}$ as $g(x) = |x|^2$, and $\mathcal{N} := \|W g(|\psi\rangle_n)\|_2$. Then we can construct the unitary $U_f$ which is a $(\frac{\alpha^2}{\mathcal{N}}, 2a + d + 3 + n, \frac{2\alpha\epsilon}{\mathcal{N}})$-VE for $W g(|\psi\rangle_n)/\mathcal{N}$, and has a circuit depth of $O(T_\psi + dn + n^2)$.*

*Proof.* Noting that $a_j = \|W|j\rangle\|_2$, it is easy to show $\|W\|_2 \leq 1 \implies \forall j, a_j \leq 1$; $a_j = \|W|j\rangle\|_2 \leq \max_{\boldsymbol{x}:\|\boldsymbol{x}\|_2=1} \|W\boldsymbol{x}\|_2 = \|W\|_2 \leq 1$. Consequently, by Lemma B.13 we can immediately get $U_A$, a $(1, d + 1, 0)$-block-encoding for $A = \text{diag}(a_0, \ldots, a_{N-1})$ with $O(dn)$ circuit depth.

Let $|\psi_1\rangle_n := A|\psi\rangle_n = \sum_{j=0}^{N-1} a_j \psi_j |j\rangle_n$, $\mathcal{N}_1 := \||\psi_1\rangle_n\|_2$. By Lemma 2, we can combine $U_A$ and $U_\psi$ to obtain $V_1$, a $(\alpha/\mathcal{N}_1, a + d + 1, \epsilon/\mathcal{N}_1)$-VE for $|\psi_1\rangle_n/\mathcal{N}_1$. This has circuit complexity $O(T_\psi + dn)$.

By Lemma B.6, we can get $U_0$, a $(1, 2, 0)$-block-encoding for the $n + a + d + 1$-qubit projector $|0\rangle\langle 0|$. Let $|E_{\psi_1}\rangle_n := \frac{|\psi_1\rangle}{\mathcal{N}_1} - \frac{\alpha}{\mathcal{N}_1}(\langle 0|_{a+d+1} \otimes I_n)V_1(|0\rangle_{n+a+d+1})$. Then, by Definition 3, $\||E_{\psi_1}\rangle_n\|_2 \leq \epsilon/\mathcal{N}_1$. Moreover, $\langle 0|_{a+d+1} \otimes I_n)V_1(|0\rangle_{n+a+d+1}) = \frac{1}{\alpha}(|\psi_1\rangle_n - \mathcal{N}_1|E_{\psi_1}\rangle_n)$. Then, observe that $V_2 := U_0(I_2 \otimes V_1^\dagger)$ is a $(1, 2, 0)$-block-encoding for $|0\rangle\langle 0|V_1^\dagger$. Let $c = a + d + 1$. Noting that $(|0\rangle_{c+2} \otimes I_n) = (|0\rangle_2 \otimes I_{c+n})(|0\rangle_c \otimes I_n)$, then,

$$(\langle 0|_{c+2} \otimes I_n)V_2(|0\rangle_{c+2} \otimes I_n) = (\langle 0|_c \otimes I_n)(\langle 0|_2 \otimes I_{c+n})V_2(|0\rangle_2 \otimes I_{c+n})(|0\rangle_c \otimes I_n) \quad \text{(B.56)}$$

$$= (\langle 0|_c \otimes I_n)|0\rangle\langle 0|V_1^\dagger(|0\rangle_c \otimes I_n) \quad \text{(B.57)}$$

$$= \frac{1}{\alpha}(|0\rangle_n(\langle \psi_1|_n - \mathcal{N}_1\langle E_{\psi_1}|_n)). \quad \text{(B.58)}$$

The third inequality follows by noting that $(\langle 0|_c \otimes I_n)|0\rangle_{n+c} = |0\rangle_n$, and that by Definition 2, $(\langle 0|_2 \otimes I_{c+n})V_2|0\rangle_2 \otimes I_{c+n}) = |0\rangle\langle 0|V_1^\dagger$. Then, letting $|0\rangle\langle \psi_1|$ be a $2^n \times 2^n$ projector,

$$\||0\rangle\langle \psi_1| - \alpha(\langle 0|_{c+2} \otimes I_n)V_2(|0\rangle_{c+2} \otimes I_n)\|_2 = \mathcal{N}_1 \||0\rangle\langle E_{\psi_1}|\|_2 \leq \epsilon. \quad \text{(B.59)}$$

Consequently, $V_2$ is a $(\alpha, a + d + 3, \epsilon)$-block-encoding for the $2^n \times 2^n$ projector $|0\rangle\langle \psi_1|$. Moreover, the circuit complexity of $V_2$ is dominated by the circuit complexity of $V_1$, and thus is $O(T_\psi + dn)$. Then, $V_3 := V_2 \otimes I_n$ is a $(\alpha, a + d + 3, \epsilon)$-block-encoding for $(|0\rangle\langle \psi_1|) \otimes I_n$.

Let $U_W$ be defined as in Definition B.3, i.e., it enacts $U_W|j\rangle_n|0\rangle_n = |j\rangle_n|w_j\rangle_n$.

Define $|\phi\rangle_{2n} := \sum_{j=0}^{N-1} \psi_j|j\rangle_n|w_j\rangle_n$.

Then, let $S := (I_a \otimes U_W)(U_\psi \otimes I_n)$. We will now show that $S$ is an $(\alpha, a, \epsilon)$-VE for $|\phi\rangle_{2n}$.

Let $|E_\psi\rangle_n := |\psi\rangle_n - \alpha(\langle 0|_a \otimes I_n)U_\psi|0\rangle_{a+n}$, thus, $(\langle 0|_a \otimes I_n)U_\psi|0\rangle_{a+n} = \frac{1}{\alpha}(|\psi\rangle_n - |E_\psi\rangle_n)$

Moreover, define the $a$-qubit projector, $p_j^a := |j\rangle\langle j|$. Then, $I_{a+n} = \sum_{j=0}^{2^a-1} p_j^a \otimes I_n$. Finally, define $|\gamma_j\rangle_n := (\langle j|_a \otimes I_n)U_\psi|0\rangle_{a+n}$. Of course,

$$U_\psi|0\rangle_{a+n} = \left(\sum_{j=0}^{2^a-1} p_j^a \otimes I_n\right)U_\psi|0\rangle_{a+n} = \frac{1}{\alpha}(|0\rangle_a(|\psi\rangle_n - |E_\psi\rangle_n)) + \sum_{j=1}^{2^a-1} |j\rangle_a|\gamma_j\rangle_n. \quad \text{(B.60)}$$

Consequently,

$$(\langle 0|_a \otimes I_{2n})S|0\rangle_{a+2n} = (\langle 0|_a \otimes I_{2n})(I_a \otimes U_W)(U_\psi \otimes I_n)|0\rangle_{a+2n} \quad \text{(B.61)}$$

$$= (\langle 0|_a \otimes U_W)\left[\frac{1}{\alpha}(|0\rangle_a(|\psi\rangle_n - |E_\psi\rangle_n)) + \sum_{j=1}^{2^a-1} |j\rangle_a|\gamma_j\rangle_n\right]|0\rangle_n \quad \text{(B.62)}$$

$$= \frac{1}{\alpha}(|\phi\rangle_{2n} - U_W|E_\psi\rangle_n|0\rangle_n). \quad \text{(B.63)}$$

Thus,

$$\| |\phi\rangle_{2n} - \alpha((\langle 0|_a \otimes I_{2n})S|0\rangle_{a+2n}\|_2 = \|U_W|E_\psi\rangle_n|0\rangle_n\|_2 \leq \epsilon. \tag{B.64}$$

Thus, $S$ is an $(\alpha, a, \epsilon)$-VE for $|\phi\rangle_{2n}$. Moreover, the circuit complexity of $S$ comes from summing the circuit complexity of $U_\psi$ and $U_W$. As per Definition B.3, the circuit complexity of $U_W$ is $O(n^2)$, giving an overall circuit complexity for $S$ of $O(T_\psi + n^2)$.

Define $|\Gamma\rangle_n := Wg(|\psi\rangle_n)$, and note that

$$[(|0\rangle\langle\psi_1|) \otimes I_n]|\phi\rangle_{2n} = |0\rangle_n \sum_{j=0}^{N-1} |\psi_j|^2 a_j |w_j\rangle_n = |0\rangle_n|\Gamma\rangle_n. \tag{B.65}$$

We now have $V_3$, a $(\alpha, a + d + 3, \epsilon)$-block-encoding for $(|0\rangle\langle\psi_1|) \otimes I_n$, and $S$ an $(\alpha, a, \epsilon)$-VE for $|\phi\rangle_{2n}$. We will now invoke Lemma 2 to take the product of the matrix encoded in $V_3$ with the vector encoded in $S$, and then will invoke Lemma B.11 to remove the $|0\rangle_n$ tensored register. This yields $U_f$, an $(\frac{\alpha^2}{\mathcal{N}}, 2a + d + 3 + n, \frac{2\alpha\epsilon}{\mathcal{N}})$-VE for $|\Gamma\rangle_n/\mathcal{N}$ with circuit complexity $O(T_\psi + dn + n^2)$. □

### B.2 CONVOLUTION BLOCK-ENCODING

In this section, we will first provide a matrix-form of a 2D multi-filter convolution (with stride 1 and 0 padding to ensure the input and outputs have the same dimension). We then derive a quantum block-encoding of the matrix form of the convolution.

As a note, some popular deep learning frameworks such as PyTorch (Paszke et al., 2019) actually implement cross-correlation rather than convolution. However, in the pre-processing stage, our convolutional block-encoding immediately gives a cross-correlation block-encoding by simply switching the $Q$ operator (Definition B.6) with a $Q^T$ operator. Finally, in this section, we assume that all addition on basis vectors is mod the dimension of the vector. I.e., for integers $i, j$, $|i+j\rangle_n = |(i+j) \bmod N\rangle_n$ (with $N = 2^n$).

**Definition B.4** (Permutation Matrix). *Define the following $N$ dimensional unitary permutation matrix that maps an input basis vector $i$ to the basis vector $(i + 1) \bmod N$.*

$$P := \sum_{i=0}^{N} |i+1\rangle\langle i| = \begin{pmatrix} 0 & 0 & 0 & \dots & 1 \\ 1 & 0 & 0 & & 0 \\ 0 & 1 & 0 & & 0 \\ \vdots & & & \ddots & \vdots \\ 0 & 0 & 0 & \dots & 0 \end{pmatrix}. \tag{B.66}$$

**Definition B.5** ($R_Z$ Phase Gate). *Define the single-qubit phase gate, $R_Z(t) := e^{itZ} = \begin{pmatrix} e^{it} & 0 \\ 0 & e^{-it} \end{pmatrix}$.*

We now derive a block-encoding of the permutation matrix $P$ acting on $m$ qubits. We include this result for completeness, and similar results may be found in the literature (see e.g., Motlagh & Wiebe (2024), where they derive a 1D circulant convolution via QSP, or Camps et al. (2024)). Our implementation of $P^m$ is identical to a $+m$ adder implemented with QFT, see e.g., Draper (2000).

**Lemma B.14** (Permutation Matrix Block-Encoding). *Let $m \in \mathbb{N}_{>0}$. Let $N = 2^n$. The $m^{th}$ power of the permutation matrix $P$ is given by $P^m = \sum_{j=0}^{N-1} |j+m\rangle\langle j|$. Then, we can get a $(1, 1, 0)$-block-encoding with $O(n^2)$ circuit complexity for $P^m$.*

*Proof.* Drawing inspiration from Motlagh & Wiebe (2024); Sedghi et al. (2019), let $F := QFT$ represent the Quantum Fourier Transform on $n$ qubits. Define $\omega_N^j := e^{2\pi ij/N}$. Noting that $F = \frac{1}{\sqrt{N}} \sum_{i=0}^{N-1} \sum_{j=0}^{N-1} \omega_N^{ij}|i\rangle\langle j|$, it is easy to show that $P^m F|j\rangle = \omega_N^{-mj} F|j\rangle$. Consequently, we can write $P^m = FDF^{-1}$, where $D = \text{diag}(\omega_N^0, \omega_N^{-m}, \dots, \omega_N^{-m(N-1)})$. Thus, by getting a block-encoding of $D$, we can implement $P^m$ by taking a product of $FDF^{-1}$. Let $|j\rangle_n$ be a basis vector, and let $j = 2^{n-1}j_{n-1} + \dots + 2^1 j_1 + 2^0 j_0$. We will now give a unitary $V_m$ which implements the mapping $V_m|0\rangle_1|j\rangle_n = \omega_N^{-jm}|0\rangle_1|j\rangle_n$. Noting that $\omega_N^{-jm} = e^{-2\pi ijm/N} = \prod_{l=0}^{n-1} e^{-2\pi im(2^{jl} j_l)/N}$,

we can apply a sequence of $n$ controlled $R_Z(t)$ gates, where the $l^{th}$ gate is controlled on bit $j_l$ and applies $R_Z(m2^{j_l}/N)$ on the ancilla qubit. This implements the desired mapping, and can be easily shown to be a $(1, 1, 0)$-block-encoding for $D$. The Quantum Fourier Transform (Coppersmith, 2002) can be implemented with $O(n^2)$ circuit complexity (Nielsen & Chuang, 2010), and so we can get a trivial $(1, 0, 0)$-block-encoding for both $F$ and $F^\dagger$. Thus, $(I_1 \otimes F)V_m(I_1 \otimes F^\dagger)$ is a $(1, 1, 0)$-block-encoding for $P^m$, with $O(n^2)$ circuit depth. Its worth noting that since the ancilla qubit in $V_m$ is separable after the computation, this could be equivalently considered a $(1, 0, 0)$-block-encoding. $\square$

**Definition B.6** (Discrete Unilateral Shift Operator). *Define $Q$ to be the $N$-dimensional discrete unilateral shift operator,*

$$Q := \sum_{j=0}^{N-2} |j+1\rangle\langle j| = \begin{pmatrix} 0 & 0 & \dots & 0 & 0 \\ 1 & 0 & & 0 & 0 \\ 0 & 1 & & 0 & 0 \\ \vdots & & \ddots & & \vdots \\ 0 & 0 & \dots & 1 & 0 \end{pmatrix}. \tag{B.67}$$

*This is just the permutation matrix $P$ without wrap-around.*

**Lemma B.15** (Block-Encoding of $Q$). *Let $N = 2^n$. Define $Q$ as per Definition B.6. Then, we can obtain a $(1, 4, 0)$-block-encoding for $Q$ with $O(n^2)$ circuit complexity.*

*Proof.* By Lemma B.14, we can obtain a $U_P$ a $(1, 1, 0)$-block-encoding of $P = \sum_{j=0}^{N-1} |j+1\rangle\langle 1|$ with $O(n^2)$ circuit complexity. By Lemma B.6, we can obtain $V$ a $(1, 2, 0)$-block-encoding of the $n$-qubit projector $|0\rangle\langle N-1|$ with $O(n)$ circuit depth. Following LCU (Childs & Wiebe, 2012; Gilyén et al., 2019), we can get the sum of these two block-encodings, introducing an additional ancilla, with the circuit $U_f := (H \otimes I_{2+n})(|0\rangle\langle 0|_1 \otimes I_1 \otimes U_P - |1\rangle\langle 1|_1 \otimes V)(H \otimes I_{2+n})$. Then, $U_f$ is a $(1, 3, 0)$-block-encoding for $\frac{1}{2}(P - |0\rangle\langle N-1|) = \frac{1}{2}Q$, with $O(n^2)$ circuit complexity. Noting that $Q^\dagger Q = I_n - |N-1\rangle\langle N-1|$, it is clear that $\|Q\|_2 \leq 1$. Moreover, since all the singular values of $Q/2$ are either 0 or $1/2$, we can invoke Lemma B.4, a special case of oblivious amplitude amplification (Gilyén et al., 2019), to immediately convert this to a $(1, 4, 0)$-block-encoding for $Q$ with only 3 calls to $U_f$. $\square$

**Lemma B.16** (Block-Encoding of $Q^m$). *Let $m \in \mathbb{N}_{>0}$ and let $N = 2^n$. Define the $N$-dimensional operator $Q$ as per Definition B.6. Then, we can obtain a $(1, 4m, 0)$-block-encoding of $Q^m$ with $O(mn^2)$ circuit complexity.*

*Proof.* As per Lemma B.15, we can obtain $U_Q$ a $(1, 4, 0)$-block-encoding for $Q$ with $O(n^2)$ circuit complexity. Invoking Lemma 53 (Product of Block-Encoded Matrices) of Gilyén et al. (2019) with $U_Q$ $m$ times directly yields a $(1, 4m, 0)$-block-encoding of $Q^m$ with $O(mn^2)$ circuit complexity.[5] $\square$

Now, we present a standard well-known result giving the matrix form of a 2D multi-filter convolution (see e.g., Sedghi et al. (2019); Kerenidis et al. (2020)).

**Lemma B.17** (Matrix Form of $2D$ Multi-Filter Convolution). *Let $M = 2^m$, let $n = 2m$, let $N = 2^n$, and let $D = 2^d$. Let $C = 2^c$ represent the number of input and output channels. Let $X$ represent the rank$-3$ input tensor, which in vectorized form (stored in column-major order for each input channel) is given by, $|X\rangle_{n+c} = \sum_{i=0}^{C-1} \sum_{j=0}^{M-1} \sum_{k=0}^{M-1} X_{i,k,j}|i\rangle_c|j\rangle_m|k\rangle_m$. I.e., $|X\rangle_{n+c}$ is of dimension $M^2 C = NC$. Define $\tilde{X}_{i,j,k} = X_{i,j,k}$ if $j \geq 0$ and $k \geq 0$, and $\tilde{X}_{i,j,k} = 0$ otherwise. We can define the convolutional kernel $K$ to be a rank-4 tensor containing each of the $C$, $C \times D \times D$ filters[6], where the first index represents the output channel, the second index represent the input channel, the third index represents the row index, and the fourth index represents the column index. Then, entry $y, z$ of the $x^{th}$ output channel after convolution with $K$ is given by,*

$$[X * K]_{x,y,z} := \sum_{j=0}^{C-1} \sum_{k=0}^{D-1} \sum_{l=0}^{D-1} K_{x,j,k,l} \tilde{X}_{j,z-k,y-l}. \tag{B.68}$$

---

[5]This can likely be optimizing by using QSVT (Gilyén et al., 2019).

[6]If the number of channels is 1 (i.e., $C = 1$), then the kernel is $D \times D$ dimensional.

*Defining $Q$ as per Definition B.6, we can give the matrix form of the convolution,*

$$\mathcal{C} := \sum_{i=0}^{C-1} \sum_{j=0}^{C-1} \sum_{k=0}^{D-1} \sum_{l=0}^{D-1} K_{i,j,k,l}(|i\rangle\langle j|_c \otimes Q^l \otimes Q^k). \tag{B.69}$$

*I.e., $\mathcal{C}|X\rangle_{n+c} = vec(X * K)$.*

*Proof.* We will verify that $\mathcal{C}$ indeed implements the mapping specified in Equation (B.68) by computing the following, $\langle x|_c\langle y|_m\langle z|_m\mathcal{C}|X\rangle_{n+c}$. Note that for all $i < l$, $\langle i|Q^l = 0$, and that for all $i \geq l$, $\langle i|Q^l = \langle i - l|$. Consequently, if $y - l \geq 0$, $z - k \geq 0$, then $\langle j|_c \otimes (\langle y|_m\langle z|_m Q^l \otimes Q^k)|X\rangle_{n+c} = X_{j,z-k,y-l}$, and if $y - l < 0$ or $z - k < 0$ then $\langle j|_c \otimes (\langle y|_m\langle z|_m Q^l \otimes Q^k)|X\rangle_{n+c} = 0$. Thus, $\langle j|_c \otimes (\langle y|_m\langle z|_m Q^l \otimes Q^k)|X\rangle_{n+c} = \tilde{X}_{j,z-k,y-l}$. Therefore,

$$\langle x|_c\langle y|_m\langle z|_m \sum_{j=0}^{C-1} K_{i,j,k,l}(|i\rangle\langle j|_c \otimes Q^l \otimes Q^k)|X\rangle_{n+c} = \sum_{j=0}^{C-1} K_{x,j,k,l}\tilde{X}_{j,z-k,y-l}. \tag{B.70}$$

As a result,

$$\langle x|_c\langle y|_m\langle z|_m\mathcal{C}|X\rangle_{n+c} = \sum_{j=0}^{C-1} \sum_{k=0}^{D-1} \sum_{l=0}^{D-1} K_{x,j,k,l}\tilde{X}_{j,z-k,y-l} = [X * K]_{x,y,z}. \tag{B.71}$$

$\square$

*Proof of Lemma 5.* Define $|X\rangle_{n+c}$, $K$, and $\mathcal{C}$ as per Lemma B.17. As a result, obtaining a block-encoding of $\mathcal{C}$ allows us to implement the desired $2D$ convolution in the vectorized setting.

First, for a given $i, j, k, l$, we will show how to obtain a block-encoding of $K_{i,j,k,l}(|i\rangle\langle j|_c \otimes Q^l \otimes Q^k)$.

Using Lemma B.6, we can obtain $U_{i,j}$ a $(1, 2, 0)$-block-encoding of the $c$-qubit projector $|i\rangle\langle j|_c$, with $O(c)$ circuit depth. Then, using Lemma B.16, we can obtain $U_{Q^l}$ a $(1, 4l, 0)$-block-encoding of $m$ qubit $Q^l$ with $O(lm^2)$ circuit complexity. We similarly obtain $U_{Q^k}$ a $(1, 4k, 0)$-block-encoding of $m$ qubit $Q^k$ with $O(km^2)$ circuit complexity. We can then invoke Lemma B.2 with $U_{i,j}$ and $U_{Q^l}$, and again with $U_{Q^k}$, to obtain $U_{i,j,l,k}$, a $(1, 2 + 4l + 4k, 0)$-block-encoding of $|i\rangle\langle j|_c \otimes Q^l \otimes Q^k$ with $O(c + Dm^2)$ circuit complexity. To make each operator act on the same number of qubits, we will augment each with the appropriate number of tensored identities to yield a $(1, 2 + 8D, 0)$-block-encoding for the corresponding operator.

Define $|K\rangle_{2c+2d} := \sum_{i=0}^{C-1} \sum_{j=0}^{C-1} \sum_{k=0}^{D-1} \sum_{k=0}^{D-1} K_{i,j,k,l}|i\rangle_c|j\rangle_c|k\rangle_d|l\rangle_d$, and define $|\sqrt{K}\rangle_{2c+2d} = \sqrt{|K\rangle_{2c+2d}}$ (with the square-root applied element-wise). Then, define $\mathcal{N}_K := \left\||\sqrt{K}\rangle_{2c+2d}\right\|_2 = \||K\rangle_{2c+2d}\|_1^{1/2}$, and $|\overline{K}\rangle_{2c+2d} := |K\rangle_{2c+2d}/\mathcal{N}_K$. Noting that this vector is $C^2D^2$ dimensional, we can brute-force construct a unitary $U_K$, with a total of $O(C^2D^2)$ single and two qubit gates, such that $U_K|0\rangle_{2c+2d} = |\sqrt{K}\rangle_{2c+2d}/\mathcal{N}_K$ (Plesch & Brukner, 2011). We can then invoke Lemma B.5, obtaining a $(\mathcal{N}_K^2, 2 + 8D + 2\log(CD), 0)$-block-encoding for $\mathcal{C}$ with $O(cdC^2D^3m^2)$ circuit complexity. This is equivalent to a $(1, 2 + 8D + 2\log(CD), 0)$-block-encoding for $\mathcal{C}/\||K\rangle_{2c+2d}\|_1$. Since we are concerned with accelerating inference, we will ignore classical pre-computation costs that must only be paid one time to construct this datastructure. We can then invoke Lemma B.3, setting $\gamma = \||K\rangle_{2c+2d}\|_1/2\|\mathcal{C}\|_2$ and $\delta = 1/2$, since $\|\mathcal{C}/\||K\rangle_{2c+2d}\|_1\|_2 \leq \frac{1}{2}\frac{2\|\mathcal{C}\|_2}{\||K\rangle_{2c+2d}\|_1}$. Neglecting the logarithmic error-terms incurred by Lemma B.3 (as these will not dominate complexity), this then yields a $(1, 3 + 8D + 2\log(CD), 0)$-block-encoding for $\frac{\mathcal{C}}{2\|\mathcal{C}\|_2}$ with $O(\frac{\||K\rangle_{2c+2d}\|_1}{\|\mathcal{C}\|_2}cdC^2D^3m^2)$ circuit depth. We will now show that $\frac{\||K\rangle_{2c+2d}\|_1}{\|\mathcal{C}\|_2} \leq DC^{3/2}$, and thus that the overall circuit depth is bounded by $O(cdm^2C^3D^4)$.

We will now upper-bound $\||K\rangle_{2c+2d}\|_1$. Define the basis vector $|x\rangle_{c+2m} = |x_1\rangle_c|x_2\rangle_m|x_3\rangle_m$. Then, the $x^{th}$ row of $\mathcal{C}$ is given by $\langle x|_{c+2m}\mathcal{C}$. Simple analysis shows that $\langle x|_{c+2m}\mathcal{C} = \sum_{j=0}^{C-1} \sum_{k=0}^{D-1} \sum_{l=0}^{D-1} K_{x_1,j,k,l}\langle j|_c \otimes \langle x_2 - l|_m \otimes \langle x_3 - k|_m$, where $\langle x_2 - l|_m = 0$ if $x_2 - l < 0$

and $\langle x_3 - k|_m = 0$ if $x_3 - k = 0$. Then it can be readily shown that $\||\langle x|_{c+2m}\mathcal{C}\||_2^2 = \sum_{j=0}^{C-1} \sum_{k=0}^{D-1} \sum_{l=0}^{D-1} |K_{x_1,j,k,l}|^2$. For any operator $A$ with $\|A\|_2 \leq 1$, the maximum column norm $\max_{|i\rangle} \|A|i\rangle\|_2 \leq \max_{|\psi\rangle : \||\psi\rangle\|_2 = 1} \|A|i\rangle\|_2 \leq 1$. Similarly, since $\|A\|_2 = \|A^\dagger\|_2$, the maximum row norm cannot exceed the spectral norm of the matrix. Therefore, any row of $\mathcal{C}$ must have $\ell_2$-norm bounded by $\|\mathcal{C}\|_2$, thus, $\sum_{j=0}^{C-1} \sum_{k=0}^{D-1} \sum_{l=0}^{D-1} |K_{x_1,j,k,l}|^2 \leq \|\mathcal{C}\|_2^2$. Consequently, $\||K\rangle_{2c+2d}\|_2^2 = \sum_{i=0}^{C-1} \sum_{j=0}^{C-1} \sum_{k=0}^{D-1} \sum_{l=0}^{D-1} |K_{i,j,k,l}|^2 \leq C \|\mathcal{C}\|_2^2$. Moreover, for an $n$-dimensional vector $\boldsymbol{x}$, $\|\boldsymbol{x}\|_1 \leq \sqrt{n} \|\boldsymbol{x}\|_2$, and thus, $\||K\rangle_{2c+2d}\|_1 \leq \sqrt{C^2 D^2}\sqrt{C} \|\mathcal{C}\|_2 = DC^{3/2} \|\mathcal{C}\|_2$. Consequently, $\||K\rangle_{2c+2d}\|_1 / \|\mathcal{C}\|_2 \leq DC^{3/2}$. $\qquad\square$

To see a set of related block-encoding circuits, see Camps et al. (2024).

It is also worth noting that the preceding result can be made substantially more efficient by utilizing a circulant convolution to implement the non-circulant convolution. We will now quickly sketch this idea for future optimization. For simplicity, we assume that the convolution has one input channel and one output channel, and that the input is a rank-2 tensor (e.g., a black and white image). Let $M = 2^m$. Then, if the input image is $X \in \mathbb{R}^{M \times M}$, we can add 0 padding with the $M \times M$ projector, $|0\rangle\langle 0|_m \otimes X$. Then, enacting a circulant convolution on this augmented operator and projecting onto the zero-state of the first register yields the desired non-circulant convolution. Moreover, we can define a circulant 2D convolution as $[X * K]_{i,j} = \sum_{k=0}^{l-1} \sum_{l=0}^{d-1} K_{k,l} X_{i-k,j-l}$. The following sketch generalizes the 1D circulant convolution given in Motlagh & Wiebe (2024), and also follows the ideas discussed in Sedghi et al. (2019). Consequently, the operator $C := \sum_{i=0}^{d-1} \sum_{j=0}^{d} K_{i,j} P^j \otimes P^i$ implements $X * K$ in the vectorized setting (using a column-major vectorization for $X$). Let $\omega_M := \exp(2\pi i/M)$ be the $M^{th}$ root of unity. Let $F := QFT$ represent the Quantum Fourier Transform on $m$ qubits. It is easy to show that $P^k F|j\rangle = \omega_M^{-kj} F|j\rangle$. Thus, let $D := F^{-1}PF = \mathrm{diag}(\omega_M^0, \omega_M^{-1}, \ldots, \omega_M^{-(M-1)})$. Consequently, $P = FDF^{-1}$, and so $C = (F \otimes F)\left(\sum_{i=0}^{d-1} \sum_{j=0}^{d-1} K_{i,j} D^j \otimes D^i\right)(F^{-1} \otimes F^{-1})$. Clearly, since implementing the QFT is efficient on a quantum computer, the key to implementing $C$ is in implementing a block-encoding of the diagonal matrix $\Gamma := \sum_{i=0}^{d-1} \sum_{j=0}^{d-1} K_{i,j} D^j \otimes D^i$. Noting that this is a 1-sparse matrix with efficiently computable entries, a technique such as Gilyén et al. (2019) can be immediately used to obtain the desired block-encoding (replacing QRAM assumptions with arithmetic oracles computing the locations and values of the non-zero elements). This can be further optimized by replacing the arithmetic with QRAM. In the multi-filter case, the diagonal matrix becomes a block-diagonal matrix (with blocks of height and width given by the number of input and output channels), and the sparse block-encoding techniques can still be used.

## B.3 Non-Linear Transformation of Vector-Encodings

We now present an essential result on transforming the amplitudes of a state encoded as a VE. This result is a direct translation of the ideas in the result given in Rattew & Rebentrost (2023) (which in turn builds on Guo et al. (2024a); Mitarai et al. (2019)) to the setting of VEs. While Rattew & Rebentrost (2023) also give a similar result in the setting of a VE (called an SPBE in that paper), they obtain it by treating the whole unitary VE as a state-preparation unitary, and then invoke their non-linear amplitude transformation (NLAT) result on that, which gives slightly worse complexity than just directly re-deriving the whole transformation result in the framework of VEs. We include the following for completeness and simplicity, and do not claim novelty on this result.

**Lemma B.18** (NLAT of VE (Rattew & Rebentrost, 2023)). *Let $N = 2^n$. Let $0 \leq \epsilon_0 \leq 1$, and $\alpha \geq 1$. We are given a unitary matrix $U_\psi$ which is an $(\alpha, a, \epsilon_0)$-VE for the $n$-qubit real quantum state $|\psi\rangle_n$ with circuit complexity $O(T)$, and a function $f : \mathbb{R} \mapsto \mathbb{R}$ with Lipschitz constant $L$ such that $f(0) = 0$. Define $\epsilon_1$ such that $0 < \epsilon_1 \leq L$. Define $\mathcal{N} := \|f(|\psi\rangle_n/\alpha)\|_2$. Define the interval of approximation $[-\tau, \tau]$, where $0 < \tau \leq 1$ which can be set to either $\tau = 1$ or any value such that $\tau \geq \frac{1+\epsilon_0}{\alpha}$ if a smaller region of approximation yields a better complexity. Define the polynomial $P : \mathbb{R} \mapsto \mathbb{R}$, such that with degree $k$, $\max_{x \in [-\tau, \tau]} |P(x) - f(x)| \leq \frac{L\epsilon_1}{2\sqrt{N}}$. Suppose we are given a bound $\tilde{\gamma}$ satisfying $\tilde{\gamma} \geq \max_{x \in [-1,1]} |P(x)/x|$, and require that $P(0) = 0$. Then, we can obtain a unitary circuit $U_f$ that is a $\left(\frac{4\tilde{\gamma}}{\mathcal{N}}, n + 2a + 4, \frac{L}{\mathcal{N}}(\epsilon_0 + \epsilon_1)\right)$-VE for $f(|\psi\rangle_n/\alpha)/\mathcal{N}$,*

*and which requires $O(k)$ calls to a controlled $U_\psi$ and $U_\psi^\dagger$ circuit, and has a total circuit depth of $O(k(n + a + T))$. This circuit can be obtained with $O(poly(k, \log(\frac{\tilde{\gamma}}{\mathcal{N}\epsilon_1})))$ classical time complexity.*

*Proof.* We will begin by considering the domain we require for the polynomial approximation. Essentially, by noting that if $\alpha > 1$, the function is being applied to a sub-component of an $\ell_2$-normalized vector, and thus the maximum value of its input will be strictly less than $\frac{1+\epsilon_0}{\alpha}$. In some cases, this could yield a more efficient polynomial approximation, and so we will write our result both in the setting where the interval of approximation is $[-1, 1]$ and $[-\frac{1+\epsilon_0}{\alpha}, \frac{1+\epsilon_0}{\alpha}]$. In particular, the function will be applied to $(\langle 0|_a \otimes I_n)U_\psi|0\rangle_{a+n}$, and so we must upper-bound the maximum amplitude in this quantity. Define $c \in \mathbb{R}$ such that $0 < c \le 1$. Define the un-normalized vector $|\phi\rangle_n := (\langle 0|_a \otimes I_n)U_\psi|0\rangle_{a+n}$. Define $|E_\psi\rangle_n := |\psi\rangle_n - \alpha|\phi\rangle_n$, and note that $\||\phi\rangle_n\|_2 \le 1$, and thus that $\frac{1}{\alpha}\||\psi\rangle_n - |E_\psi\rangle_n\|_2 \le 1$. Additionally, by Definition 3, $\||E_\psi\rangle_n\|_2 \le \epsilon_0$. Define $\{\phi_j\}_j$ such that $|\phi\rangle_n = \sum_{j=1}^N \phi_j|j\rangle_n$. Thus, $|\phi_j| \le \||\phi\rangle_n\|_2 \le \frac{1}{\alpha}(1 + \epsilon_0)$. Define $c := \min(\frac{1}{\alpha}(1 + \epsilon_0), 1)$.

Let $\mathcal{N}_\psi := \mathcal{N}$. Let $\mathcal{N}_P := \|P(|\psi\rangle/\alpha)\|_2$. Define the degree $k - 1$ polynomial $Q(x) := P(x)/x$, and define $\epsilon_2$ such that $\max_{x \in [-c,c]} |P(x) - f(x)| \le \epsilon_2$.

Using Lemma 6 of Rattew & Rebentrost (2023), we can get a $(1, a + n + 2, 0)$-block-encoding $U_A$ of $A := \text{diag}(U_\psi|0\rangle_{a+n})$ with $O(a + n)$ circuit depth, and 6 additional calls to a controlled $U_\psi$ circuit. Invoking Theorem 56 of Gilyén et al. (2019) with $Q(x)/4\tilde{\gamma}$, we get the unitary $U_Q$, a $(1, a + n + 4, \delta)$-block-encoding for $Q(A)/4\tilde{\gamma}$, requiring $O((a + n)k)$ single and two-qubit gates, $O(k)$ calls to a controlled $U_A$ circuit, and $O(poly(k, \log(1/\epsilon)))$ classical computation to determine the QSVT rotation angles to implement the degree $k$ polynomial. We can equivalently call $U_Q$ a $(1, a + n + 4, 0)$-block-encoding for some matrix $V$, such that $\|V - Q(A)/4\tilde{\gamma}\|_2 \le \delta$. Additionally, define $E_Q := V - Q(A)/4\tilde{\gamma}$. Since for any vector $\boldsymbol{x}$, $Q(\text{diag}(\boldsymbol{x}))\boldsymbol{x} = P(\boldsymbol{x})$, we get, $VU_\psi|0\rangle_{a+n} = \frac{P(U_\psi|0\rangle_{a+n})}{4\tilde{\gamma}} + E_V U_\psi|0\rangle_{a+n}$. Additionally, noting that $(\langle 0|_a \otimes I_n)P(\boldsymbol{x}) = P((\langle 0|_a \otimes I_n)\boldsymbol{x})$, and that $(\langle 0|_a \otimes I_n)U_\psi|0\rangle_{a+n} = |\phi\rangle_n$, we get $(\langle 0|_a \otimes I_n)VU_\psi|0\rangle_{a+n} = \frac{P(|\phi\rangle_n)}{4\tilde{\gamma}} + (\langle 0|_a \otimes I_n)E_V U_\psi|0\rangle_{a+n}$. Define $\tilde{U}_\psi := I_{a+n+4} \otimes U_\psi$.

First, note that $\tilde{U}_\psi|0\rangle_{2a+2n+4} = (|0\rangle_{n+a+4} \otimes I_{a+n})U_\psi|0\rangle_{a+n}$. Then, note that $(\langle 0|_{n+2a+4} \otimes I_n) = (\langle 0|_a \otimes I_n)(\langle 0|_{n+a+4} \otimes I_{a+n})$. Consequently, by Definition 2, since $(\langle 0|_{n+a+4} \otimes I_{a+n})U_Q(|0\rangle_{n+a+4} \otimes I_{a+n}) = V$,

$$(\langle 0|_{n+2a+4} \otimes I_n)U_Q\tilde{U}_\psi|0\rangle_{2n+2a+4} = (\langle 0|_a \otimes I_n)VU_\psi|0\rangle_{a+n} \tag{B.72}$$

$$= \frac{P(|\phi\rangle_n)}{4\tilde{\gamma}} + (\langle 0|_a \otimes I_n)E_V U_\psi|0\rangle_{a+n}. \tag{B.73}$$

We will now show that $U_Q\tilde{U}_\psi$ is a VE for $\frac{1}{\mathcal{N}_\psi}f(|\psi\rangle_n/\alpha)$. Precisely, we must upper-bound,

$$\xi_1 := \left\|\frac{1}{\mathcal{N}_\psi}f(|\psi\rangle_n/\alpha) - \frac{4\tilde{\gamma}}{\mathcal{N}_\psi}(\langle 0|_{n+2a+4} \otimes I_n)U_Q\tilde{U}_\psi|0\rangle_{2n+2a+4}\right\|_2 \tag{B.74}$$

$$\le \frac{1}{\mathcal{N}_\psi}\left(\|f(|\psi\rangle_n/\alpha) - P(|\phi\rangle_n)\|_2 + 4\tilde{\gamma}\|(\langle 0|_a \otimes I_n)E_V U_\psi|0\rangle_{a+n}\|_2\right). \tag{B.75}$$

Let $\langle j|E_\psi\rangle := e_j$. We will now prove a sequence of simple facts. Since $|f(x) - f(x + b)| \le L|b|$, and using $|\phi\rangle_n = \frac{|\psi\rangle_n - |E_\psi\rangle_n}{\alpha}$, we have that $\|f(|\psi\rangle_n/\alpha) - f(|\phi\rangle_n)\|_2^2 = \sum_{i=j}^N |f(\psi_j) - f((\psi_j - e_j)/\alpha)|^2 \le \frac{L^2}{\alpha^2}\sum_{j=1}^N |e_j|^2 = \frac{L^2}{\alpha^2}\||E_\psi\rangle_n\|_2^2 \le \frac{L^2\epsilon_0}{\alpha^2}$. Then, $\|f(|\phi\rangle_n) - P(|\phi\rangle_n)\|_2^2 = \sum_{j=1}^N |f(\phi_j) - P(\phi_j)|^2 \le \max_{x \in [-c,c]} |f(x) - P(x)|^2 N \le \epsilon_2^2 N$. Then,

$$\|f(|\psi\rangle_n/\alpha) - P(|\phi\rangle_n)\|_2 = \|f(|\psi\rangle_n/\alpha) - f(|\phi\rangle_n) + f(|\phi\rangle_n) - P(|\phi\rangle_n)\|_2 \tag{B.76}$$

$$\le \frac{L\epsilon_0}{\alpha} + \epsilon_2\sqrt{N}. \tag{B.77}$$

At this point, the proof branches into two cases. The first case is where we simply use the uniform approximation to the function on the entire interval $[-1, 1]$. The second case, which should only be used when approximating the function on $[-\tau, \tau]$ yields a better asymptotic approximation, will be proven after.

Noting that $\|(\langle 0|_a \otimes I_n)E_V U_\psi |0\rangle_{a+n}\|_2 \leq \delta$, we can now get the overall bound of

$$\xi_1 \leq \frac{1}{\mathcal{N}_\psi}\left(\frac{L\epsilon_0}{\alpha} + \epsilon_2\sqrt{N} + 4\tilde{\gamma}\delta\right) \leq \frac{1}{\mathcal{N}_\psi}\left(L\epsilon_0 + \epsilon_2\sqrt{N} + 4\tilde{\gamma}\delta\right). \tag{B.78}$$

Thus, we have shown that $U_Q\tilde{U}_\psi$ is a $(\frac{4\tilde{\gamma}}{\mathcal{N}_\psi}, 2a+n+4, \frac{1}{\mathcal{N}_\psi}(L\epsilon_0+\epsilon_2\sqrt{N}+4\tilde{\gamma}\delta))$-VE for $\frac{1}{\mathcal{N}_\psi}f(|\psi\rangle_n/\alpha)$. To get the overall error-bound, we will set $\epsilon_2\sqrt{N} = L\epsilon_1/2$, and $4\tilde{\gamma}\delta = L\epsilon_1/2$, yielding $\epsilon_2 = \frac{L\epsilon_1}{2\sqrt{N}}$, and $\delta = \frac{L\epsilon_1}{8\tilde{\gamma}}$. This gives a $(\frac{4\tilde{\gamma}}{\mathcal{N}_\psi}, 2a+n+4, \frac{L}{\mathcal{N}_\psi}(\epsilon_0+\epsilon_1))$-VE for $\frac{1}{\mathcal{N}_\psi}f(|\psi\rangle_n/\alpha)$, and requires $O(k)$ calls to a controlled $U_\psi$ and $U_\psi^\dagger$ circuit, and has a total circuit depth of $O(k(n+a+T))$. This circuit can be obtained with $O(\text{poly}(k, \log(\frac{\tilde{\gamma}}{L\epsilon_1})))$ classical time complexity. $\qquad\square$

To make the preceding result easier to use, we provide a special case for transformation by the error function, and again do not claim novelty.

**Lemma B.19** (Application of $\text{erf}(\nu x)$ to a Vector Encoding). *Let $N = 2^n$, let $\nu \geq 1/2$, let $1 \geq \epsilon_0 \geq 0$ and let $0 < \epsilon_1 \leq 2$. We are given a unitary matrix $U_\psi$ with circuit complexity $O(T)$ which is an $(\alpha, a, \epsilon_0)$-VE for the $n$-qubit quantum state $|\psi\rangle_n$, and we are also given the error function $f_\nu(x) = \text{erf}(\nu x)$. Let $\mathcal{N} := \|f_\nu(|\psi\rangle_n/\alpha)\|_2$. Then, we can obtain a $\left(\frac{16\nu}{\sqrt{\pi}\mathcal{N}}, 2a + n + 4, 2\nu\alpha(\epsilon_0 + \epsilon_1)\right)$-VE for $f_\nu(|\psi\rangle_n/\alpha)/\mathcal{N}$, with $O(\nu\log(\frac{\sqrt{N}}{\epsilon_1}))$ queries to a controlled $U_\psi$ and $U_\psi^\dagger$ circuit, and with a total circuit depth of $O(\nu\log(\frac{\sqrt{N}}{\epsilon_1})(a + n + T))$. Moreover, $\mathcal{N} \geq \frac{1}{2\alpha}$.*

*Proof.* From Lemma F.1, we know that the Lipschitz constant $L$ of $\text{erf}(\nu x)$ is $L = \frac{2\nu}{\sqrt{\pi}}$.

Define $c = O(1/\alpha)$. Using Lemma F.1, we can obtain a degree $k \in O(\nu\log(\nu/\alpha\epsilon'))$ polynomial $P_{k,\nu}$ such that $P_{k,\nu}(0) = 0$ and $\max_{x\in[-c,c]}|P_{k,\nu}(x) - f_\nu(x)| \leq \epsilon'$. Since we need $\epsilon' \leq \frac{L\epsilon_1}{2\sqrt{N}}$, we can set $\epsilon' = \frac{\nu\epsilon_1}{10\sqrt{N}}$ in accordance with Lemma B.18, we have a degree $k \in O(\nu\log(\frac{\sqrt{N}}{\epsilon_1}))$ polynomial approximation.

From Lemma F.1, for $\nu \geq 1/2$, we know that $\forall x \in [-1,0)\cup(0,1], |\text{erf}(\nu x)| \geq |x/2|$. Consequently, $\mathcal{N}^2 = \sum_{j=1}^N |f(\psi_j/\alpha)|^2 \geq (\frac{1}{2\alpha})^2$. Additionally, we know that $\tilde{\gamma} = \max_{x\in[-1,1]}|P_{k,\nu}(x)/x| \leq \frac{4\nu}{\sqrt{\pi}}$. Invoking Lemma B.18, setting with all of the above facts and setting $\tilde{\gamma} = \frac{4\nu}{\sqrt{\pi}}$ then gives the complexity. $\qquad\square$

## C  GENERAL ARCHITECTURAL BLOCKS

The architectural blocks we present in this paper are intended to demonstrate how the different operations on encoded matrices and vectors can be combined to coherently implement various architectures on quantum computers. There is a rich set of possibilities, and we are only exploring a small but elucidating set.

Two of the most important concepts governing the complexity of the quantum implementation of any classical architecture are: (1) the number of non-linear activation layers, and (2) the $\ell_2$ norm of the vectorized input tensor as it propagates through the network.

In order for a unitary matrix (a linear operator) to enact a non-linear transformation on a vector, its definition must depend on the vector it is being applied to. Consequently, techniques which enact non-linear transformations on state-amplitudes (e.g., Rattew & Rebentrost (2023); Guo et al. (2024a)) must have circuit definitions which depend on the vector-encoding circuit they are being applied to. Thus, if the unitary circuit implementing the transformation requires *even two* calls to the input vector encoding, then the circuit complexity will grow exponentially with the number of non-linear activations. Consequently, wide but shallow multi-layer architectures are ideal for quantum acceleration. Finally, an alternative to fully coherent quantum acceleration is to periodically read-out the vector in intermediate layers of the network. As discussed in the introduction, several quantum computing papers have proposed this approach. However, in general, since reading out a quantum state incurs a dimension-dependent cost (Cramer et al., 2010; van Apeldoorn et al., 2023) (and incurs

polynomial error-dependence) this either imposes significant constraints on the types of architectures that can be accelerated (requiring frequent mappings to very low-dimensional spaces where readout is cheaper), or incur asymptotically dominating error accumulation. Nevertheless, there are certain settings where periodic state readout may be desirable, and our techniques are fully compatible with these ideas.

The second key concept governing the complexity of a quantum implementation of an architecture relates to the norm of the encoded vector as it propagates through the network. Whenever a sample is drawn from an encoded vector, a cost inversely proportional to the norm of the encoded vector must be paid. Similarly, whenever an encoded vector is normalized, an inverse norm-cost must be paid. Consequently, to obtain provable end-to-end complexity results, we need to be able to lower-bound the norm of the encoded vector whenever we apply a layer norm (or draw a sample from the output of the network). A key tool in doing this is the skip connection, as it allows the norm from the previous layer to be preserved in the output of the next layer. Additionally, if the weight layers are normalized (i.e., if $W$ represents the matrix form of any parameter layer, then $\|W\|_2 \leq 1$), and the activation function is scaled so that its Lipschitz constant on the interval $[-1, 1]$ is at most 1, this results in provable norm-preservation bounds. Requiring weight-layers to be sub-normalized has been extensively explored in the classical deep learning literature (Miyato et al., 2018; Yoshida & Miyato, 2017; Gouk et al., 2020), as sub-normalization can help prevent network norm explosion as deeper networks are trained.

It is worth briefly noting that, in certain cases, the sub-normalization condition on the weight layers can be removed (i.e., for matrix $W$, $0 \leq \|W\|_2 \leq c$ where $c \geq 1$). This is done by implementing $W/\|W\|_2$, and then scaling the input of the subsequent activation function by $\|W\|_2$. If using the error function activation, this increases the cost of the polynomial approximation by an amount proportional to $c$. We do not consider this regime as it makes it more challenging to prove norm preservation properties after the skip connection, but stress that quantum computers can actually implement such regimes. Numerical studies examining norm preservation for such networks could shed light into their efficiency.

We will now formally define our $\ell_2$-norm squared pooling; this is essentially just an $\ell_2$-norm pooling operation followed by an element-wise square. Throughout we will assume that dimensions neatly line-up, noting that if they don't padding can be used to easily and efficiently ensure alignment.

**Definition C.1** (Squared $\ell_2$ Norm Pooling). *Given an $N$-dimensional vector $|\phi\rangle = \sum_{k=1}^{N} \phi_k |k\rangle$, and a positive integer $C$ such that $N$ is divisible by $C$, define $f_j := (j-1)\frac{N}{C} + 1$. Then, we define $\ell_2$-norm squared pooling by $\mathrm{pool}_C(|\phi\rangle) := \sum_{j=1}^{C} \sum_{l=f_j}^{j\frac{N}{C}} \phi_l^2 |j\rangle$, where $\{|j\rangle\}$ is the set of $C$-dimensional basis vectors.*

**Lemma C.1** (Error Propagated Through $\ell_2$ Norm Squared Pooling). *Define the $N$-dimensional vectors $|\phi\rangle$ and $|\tilde{\phi}\rangle$, such that $\left\| |\phi\rangle - |\tilde{\phi}\rangle \right\|_2 \leq \epsilon$. Then, defining a positive integer $C$ such that $N$ is divisible by $C$, and defining $\mathrm{pool}_C$ as per Definition C.1, we have that $\left\| \mathrm{pool}_C(|\phi\rangle) - \mathrm{pool}_C(|\tilde{\phi}\rangle) \right\|_2 \leq \frac{2N\epsilon}{\sqrt{C}}$.*

*Proof.* Let $|\phi\rangle = \sum_{j=1}^{N} \phi_j |j\rangle$, and let $|\tilde{\phi}\rangle = \sum_{j=1}^{N} \tilde{\phi}_j |j\rangle$. Then, $\left\| |\phi\rangle - |\tilde{\phi}\rangle \right\|_2 \leq \epsilon$ implies that $\forall j, |\phi_j - \tilde{\phi}_j| \leq \epsilon$. Then, additionally using that $|\phi_j + \tilde{\phi}_j| \leq 2$,

$$\left\| \mathrm{pool}_C(|\phi\rangle) - \mathrm{pool}_C(|\tilde{\phi}\rangle) \right\|_2^2 = \sum_{j=1}^{C} \left( \sum_{l=f_j}^{jN/C} (\phi_l - \tilde{\phi}_l)(\phi_l + \tilde{\phi}_l) \right)^2 \leq 4 \sum_{j=1}^{C} \left( \frac{N\epsilon}{C} \right)^2 = \frac{4N^2\epsilon^2}{C}.$$

(C.1)

$\square$

***Proof of Lemma 6.*** The parameter $\kappa$ in the lemma is designed for situations where we don't have a perfect block-encoding of the matrix we would like. For instance, in cases where we want to apply some matrix $A$, but we are only able to get a block-encoding of $A/2$. We can fix this when applying the activation function by scaling its input to remove the $1/2$ factor.

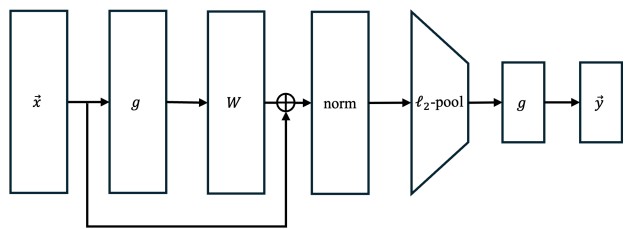

Figure 4: **Full-rank linear-pooling output block.**

Figure 5: This figure shows the final output architectural block used in our neural networks for Regimes 1 and 2. Here, $g(x) = x^2$ and $W$ is a sub-normalized (potentially full-rank and dense) matrix.

Let $\nu := 4\kappa/5$. Let $|\phi_1\rangle_n := W|\psi\rangle_n/\kappa$, $\mathcal{N}_1 := \||\phi_1\rangle_n\|_2$, and $|\Phi_1\rangle_n := |\phi_1\rangle_n/\mathcal{N}_1$. Using Lemma 2 we get $U_1$ a $(\mathcal{N}_1^{-1}, a+b, \epsilon_0\mathcal{N}_1^{-1})$-VE for $|\Phi_1\rangle_n$ with $O(T_1 + T_2)$ circuit complexity.

Let $|\phi_2\rangle_n := f(W|\psi\rangle_n))$, $\mathcal{N}_2 := \||\phi_2\rangle_n\|_2$, and $|\Phi_2\rangle_n := |\phi_2\rangle_n/\mathcal{N}_2$. Define $0 < \epsilon_1 \leq 1$. Invoking Lemma B.19 with $U_1$ and $f(\kappa x) = \operatorname{erf}(4\kappa x/5) = \operatorname{erf}(\nu x)$, we obtain $U_2$ a $\left(\frac{16\nu}{\sqrt{\pi}\mathcal{N}_2}, 2(a+b) + n + 4, 2\nu\mathcal{N}_1^{-1}(\epsilon_0 + \epsilon_1)\right)$-VE for $f(|\Phi_1\rangle_n\mathcal{N}_1)/\|f(|\Phi_1\rangle_n\mathcal{N}_1)\|_2 = |\Phi_2\rangle_n$. $U_2$ has circuit complexity $O(\nu \log(\frac{\sqrt{N}}{\epsilon_1})(a+b+n+T_1+T_2))$.

So as to invoke Lemma 1 to implement the skip connection and obtain a state proportional to $|\psi_f\rangle_n$, we will need to factor out a common factor of $\frac{\sqrt{\pi}}{16\nu}$. Consequently, we invoke Lemma B.9 on $U_\psi$ to obtain $U_\psi'$ a $(\frac{16\nu}{\sqrt{\pi}}, a+2, \epsilon_0)$-VE for $|\psi\rangle_n$ with $O(T_1 + a)$ circuit complexity.

Define $|\gamma\rangle_n := \frac{\sqrt{\pi}}{32\nu}(|\psi\rangle_n + |\Phi_2\rangle_n\mathcal{N}_2) = \frac{\sqrt{\pi}}{32\nu}(|\psi\rangle_n + f(W|\psi\rangle_n))$, $\mathcal{N}_\gamma := \||\gamma\rangle_n\|_2$ and $|\Gamma\rangle_n := |\gamma\rangle_n/\mathcal{N}_\gamma$. Then, we can invoke Lemma 1 (setting $\tau = 1/2$) with $U_\psi'$ and $U_2$, yielding $U_3$ a $\left(\mathcal{N}_\gamma^{-1}, 2(a+b) + n + 5, \mathcal{N}_\gamma^{-1}[\frac{\epsilon_0\sqrt{\pi}}{16\nu} + \frac{\mathcal{N}_2\sqrt{\pi}}{16\nu}(2\nu\mathcal{N}_1^{-1}(\epsilon_0 + \epsilon_1))]\right)$-VE for $|\Gamma\rangle_n$, with circuit complexity $O(\nu \log(\frac{\sqrt{N}}{\epsilon_1})(a+b+n+T_1+T_2))$. We will now simplify the error component of this VE statement.

First, define $|x\rangle_n = \sum_i x_i|i\rangle_n = W|\psi\rangle_n$. Then, using the fact that $f(x) = \operatorname{erf}(4x/5)$ has a Lipschitz-constant of $\frac{8}{5\sqrt{\pi}}$ (as per Lemma B.19), $\mathcal{N}_2^2 = \|f(W|\psi\rangle_n)\|_2^2 \leq \sum_i |f(x_i)|^2 \leq (\frac{8}{5\sqrt{\pi}})^2 \sum_i |x_i|^2 = (\frac{8}{5\sqrt{\pi}})^2 \|W|\psi\rangle_n\|_2^2$. Since $\|W\|_2 \leq 1$, $\|W|\psi\rangle_n\|_2 \leq 1$, and thus $\mathcal{N}_2 \leq \frac{8}{5\sqrt{\pi}} \leq 0.91$. Next, we must lower-bound $\mathcal{N}_\gamma$. Thus, $\mathcal{N}_\gamma^2 = (\frac{\sqrt{\pi}}{32\nu})^2 (1 + \mathcal{N}_2^2 + 2\mathcal{N}_2\langle\Phi_2|\psi\rangle) \geq (\frac{\sqrt{\pi}}{32\nu})^2 (1 - \mathcal{N}_2)^2 \geq (\frac{\sqrt{\pi}}{32\nu})^2(0.09)^2$. Consequently, using that $\nu \leq 8/5$ (since $\kappa \leq 2$) we get that $\mathcal{N}_\gamma \geq 1/400$. Additionally, it is straight-forward to show that $\mathcal{N}_2/\mathcal{N}_1 \leq 0.91\kappa \leq 2$. Inserting all of these values and performing simple algebra, we find that $U_3$ is equivalently a $\left(\mathcal{N}_\gamma^{-1}, 2(a+b) + n + 5, 355(\epsilon_0 + \epsilon_1)\right)$-VE for $|\Gamma\rangle_n$.

Let $0 < \epsilon_2 \leq 1$. Then, invoking Lemma B.8, we get $U_f$, a $(1, 2(a+b) + n + 9, 2(355(\epsilon_0 + \epsilon_1) + \epsilon_2))$-VE for $|\Gamma\rangle_n$, with circuit complexity $O(\log(\frac{\sqrt{N}}{\epsilon_1})\log(\frac{1}{\epsilon_2})(a+b+n+T_1+T_2))$. If we let $\epsilon_2 = \epsilon_1$, then we can simplify this to a $(1, 2(a+b) + n + 9, 712(\epsilon_0 + \epsilon_1))$-VE with circuit complexity $O(\log(\frac{\sqrt{N}}{\epsilon_1})\log(\frac{1}{\epsilon_1})(a+b+n+T_1+T_2))$. $\square$

***Proof of Lemma 7***. This result comes from repeatedly invoking Lemma 6, with the output of each application becoming the input of the next.

We will first give a bound on the total number of ancilla qubits of the block-encoding giving the final output after $k$ residual block layers. Let $a_0 = a$. $c = 2b + n + 9$. After one application of the residual block, the number of ancillas is given by $a_1 = 2a_0 + c$. Then, the general form for the number of ancillas is given by the recurrence $a_i = 2a_{i-1} + c$. We can obtain an upper-bound by

instead setting $a_i = 2(a_{i-1} + c)$. Clearly, $a_i = 2^i(a + c) = 2^i(a + 2b + n + 9)$. Thus, we have a $(1, 2^k(a + 2b + n + 9), \epsilon)$-block-encoding.

We will now determine a bound on the resulting error, $\epsilon$. Note that the $i^{th}$ residual block introduces a new error-parameter $\epsilon_i$ which controls the error in the activation function and the normalization of that block. After the first iteration, the error $\delta_1$ is given by $\delta_1 = 712(\epsilon_0 + \epsilon_1)$. After the second iteration, the error from the previous iteration becomes the new $\epsilon_0$, and so the error after the second iteration is given by $\delta_2 = 712(\delta_1 + \epsilon_2)$. We can set $\epsilon_i = \delta_{i-1}$, giving the general form of the error after the $i^{th}$ residual layer of $\delta_i = 1424\delta_{i-1} = 724 \cdot 1424^{i-1}\epsilon_1 = 1424^i\epsilon_1/2$. Noting that we want a final error of at most $\epsilon$, we must set $\delta_k \le \epsilon$. I.e., we can set $\epsilon = 1424^k\epsilon_1/2 \implies \epsilon_1 = 2\epsilon/1424^k$. Thus, for $i > 1$, each $\epsilon_i = \delta_{i-1} = 1424^i\epsilon_1/2 = \frac{1424^i}{1424^k}\epsilon = \epsilon/1424^{k-i}$.

Define $h(\epsilon_i) := \log(\sqrt{N}/\epsilon_i)\log(1/\epsilon_i)$. Let the circuit complexity of the block-encoding after applying $i$ residual blocks be $O(R_i)$. Noting that $R_1 \in O(h(\epsilon_1)(a_0 + b + n + T_1 + T_2))$, $R_i$ asymptomatically dominates $T_1, T_2, a_{i-1}, n$ and $b$. Then, the circuit complexity after block $i + 1$ will be $O(h(\epsilon_{i+1})(a_i + b + n + R_i + T_1)) \in O(h(\epsilon_{i+1})(a_i + R_i))$. Then, we can simplify to find that $R_k \in O((a_k + R_1)\prod_{i=1}^k h(\epsilon_i)) \in O((2^k(a + 2b + n) + T_1 + T_2)\prod_{i=1}^k h(\epsilon/1424^{k-i}))$. Noting that $\prod_{i=1}^k h(\epsilon_i) \in O((\prod_{i=1}^k \log(\sqrt{N}/\epsilon_i))^2)$, $\prod_{i=1}^k h(\epsilon/1424^{k-i}) \in O((\prod_{i=1}^k (k - i + \log(\sqrt{N}/\epsilon_i)))^2) \in O((k + \log(\sqrt{N}/\epsilon))^{2k})$. Since $k$ is an asymptotic constant, $O(k + \log(\sqrt{N}/\epsilon)) \in O(\log(\sqrt{N}/\epsilon))$, and so $\prod_{i=1}^k h(\epsilon/1424^{k-i}) \in O(\log(\sqrt{N}/\epsilon)^{2k})$. Thus, the overall circuit complexity is given by $O(\log(\sqrt{N}/\epsilon)^{2k}(a + 2b + n + T_1 + T_2))$.

$\square$

**Lemma C.2** (Full-Rank Linear Pooling Output Block). *Consider the architecture block shown in Figure 5. Let the dimension of the input vector be $N = 2^n$, and let the dimension of the output of the network block be $C = 2^c$ (i.e., the number of classes). Let the output of the network be given by the vector $|y\rangle_c$. Suppose we have $U_\psi$ an $(1, a, \epsilon_0)$-VE for the $N$-dimensional input vector $|\psi\rangle_n = \boldsymbol{x}$ with $O(T_{\epsilon_0})$ circuit complexity. Here, $T_{\epsilon_0}$ makes explicit that the complexity of the input circuit will be dependent on the desired error of the vector encoding of the layer input to this architectural block. Suppose we are given access to an arbitrary matrix $W$ such that $\|W\|_2 \le 1$ as per Theorem B.1. Then, if the weight on the skip-path is $\tau = 0.51$, we can draw a sample from a vector $|\tilde{\phi}\rangle_c$ such that $\left\| |\tilde{\phi}\rangle_c - |y\rangle_c \right\|_2 \le \epsilon$ with $O(\log(\frac{N}{\sqrt{C}\epsilon})(T_{\epsilon_0} + a + n^2))$ circuit complexity and with $O(a + n)$ total ancilla qubits.*

*Proof.* Let $d$ represent the number of bits in part of the QRAM encoding of $W$, as per Theorem B.1. Note that $d$ is assumed to be an asymptotic constant. Let $|\phi_1\rangle_n := Wg(|\psi\rangle_n)$, $\mathcal{N}_1 := \||\phi_1\rangle_n\|$ and $|\Phi_1\rangle_n := |\phi_1\rangle_n/\mathcal{N}_1$. Using Theorem B.1, we can get a $(\mathcal{N}_1^{-1}, 2a + d + 3 + n, 2\epsilon_0\mathcal{N}_1^{-1})$-VE for $|\Phi_1\rangle_n$ with $O(T_{\epsilon_0} + dn + n^2)$ circuit complexity. Here $d$ is a constant specifying the precision in the representation of the elements of the matrix stored as per Definition B.3.

Let $|\gamma\rangle_n := \tau|\psi\rangle_n + (1 - \tau)|\Phi_1\rangle_n\mathcal{N}_1 = \tau|\psi\rangle_n + (1 - \tau)Wg(|\psi\rangle_n)$, and let $\mathcal{N}_\gamma := \||\gamma\rangle_n\|_2$.

Then, Lemma 1 yields $V_2$ a $(\mathcal{N}_\gamma^{-1}, 2a + d + 4 + n, 3\epsilon_0\mathcal{N}_\gamma^{-1})$-VE for $|\gamma\rangle_n/\mathcal{N}_\gamma$ with $O(T_{\epsilon_0} + dn + n^2)$ circuit complexity.

We will now lower-bound $\mathcal{N}_\gamma$. The main idea is that if you are summing two vectors, one with norm 1, and the other with norm at most 1, if you put arbitrarily more mass on the constant-norm vector ($\delta$), you are guaranteed that the vectors cannot fully cancel out, and thus that some norm is preserved in the sum. Note that $\mathcal{N}_1 = \|Wg(|\psi\rangle_n)\|_2 \le \|W\|_2 \left\|\sum_j \psi_j^2|j\rangle_n\right\|_2 \le \left\|\sum_j \psi_j|j\rangle_n\right\|_2 = 1$. Consequently, $|\langle\psi|\Phi_1\rangle| \le 1$, and so

$$\mathcal{N}_\gamma^2 = \|\tau|\psi\rangle_n + (1 - \tau)|\Phi_1\rangle_n\mathcal{N}_1\|_2^2 = \tau^2 + (1 - \tau)^2\mathcal{N}_1^2 + 2\tau(1 - \tau)\mathcal{N}_1\langle\psi|\Phi_1\rangle \quad (\text{C.2})$$

$$\ge \tau^2 + (1 - \tau)^2\mathcal{N}_1^2 + 2\tau(1 - \tau) = (\tau - (1 - \tau)\mathcal{N}_1)^2. \quad (\text{C.3})$$

For some parameter $\delta \in [0, 1]$, assuming that $\tau = (1 + \delta)/2$, we then get that $\mathcal{N}_\gamma \ge \delta$.

Then, define $\epsilon_1 \in (0, 1]$. We can then invoke Lemma B.8 yielding $V_3$ a $(1, 2a + d + 8 + n, \frac{6\epsilon_0}{\delta} + 2\epsilon_1)$-VE for $|\gamma\rangle_n/\mathcal{N}_\gamma$ with $O(\frac{1}{\delta}\log(1/\epsilon_1)(T_{\epsilon_0} + a + dn + n^2))$ circuit complexity.

Define $\text{pool}_C$ as per Definition C.1. Noting that $\text{pool}_C(|\gamma\rangle_n/\mathcal{N}_\gamma) = |y\rangle_c$.

We can equivalently define some $\ell_2$-normalized state $|\tilde{\Gamma}\rangle_n$ such that $V_3$ is a $(1, 2a + d + 8 + n, 0)$-VE for $|\tilde{\Gamma}\rangle_n$. Then, since $\left\| |\tilde{\Gamma}\rangle_n - \frac{|\gamma\rangle_n}{\mathcal{N}_\gamma} \right\|_2 \leq \frac{6\epsilon_0}{\delta} + 2\epsilon_1$, we can invoke Lemma C.1 which shows that $\left\| \text{pool}_C(|\tilde{\Gamma}\rangle_n) - |y\rangle_c \right\|_2 \leq \frac{2N}{\sqrt{C}}(\frac{6\epsilon_0}{\delta} + 2\epsilon_1)$.

Consequently, to get an error of at most $\epsilon$, we set $\frac{2N}{\sqrt{C}}(\frac{6\epsilon_0}{\delta} + 2\epsilon_1) = \epsilon$, by setting $\epsilon_1 = \frac{\sqrt{C}\epsilon}{8N}$ and $\epsilon_0 = \frac{\epsilon\sqrt{C}\delta}{24N}$. Then, we can simply draw a sample $\epsilon$-close to $|y\rangle_c$ in $\ell_2$-norm distance by sampling the state prepared by $V_3$ and then assigning it to the appropriate bin.

Setting $\delta = 0.02$ gives $\tau = 0.51$. Then, $V_3$ is a $(1, 2a + d + 8 + n, \epsilon)$-VE for $|\gamma\rangle_n/\mathcal{N}_\gamma$ with $O(\log(\frac{N}{\sqrt{C}\epsilon})(T_{\epsilon_0} + a + dn + n^2))$ circuit complexity. Consequently, we can draw a sample from some vector $|\tilde{\phi}\rangle_c$ such that $\left\| |\tilde{\phi}\rangle_c - |y\rangle_c \right\|_2 \leq \epsilon$ with $O(\log(\frac{N}{\sqrt{C}\epsilon})(T_{\epsilon_0} + a + dn + n^2)) \in O(\log(\frac{N}{\sqrt{C}\epsilon})(T_{\epsilon_0} + a + n^2))$ circuit complexity, and with $O(a + n)$ ancilla qubits, noting that $d$ is an asymptotic constant. $\qquad\square$

# D  FEASIBILITY OF QRAM ASSUMPTIONS

In this section, we consider the feasibility of different QRAM assumptions to help motivate our discussion in Appendix E. In Section D.1 we consider the feasibility of our QRAM assumptions. In Section D.2 we summarize how arbitrary quantum states can be prepared by using a QRAM data-structure, in service of our subsequent discussion of the different architectural regimes.

## D.1  PASSIVE AND ACTIVE QRAM

It is clear that, if a fault-tolerant quantum computer can be constructed, that a QRAM based on the various quantum circuit constructions (see Jaques & Rattew (2025); Giovannetti et al. (2008a); Hann (2021)) can be directly implemented. Moreover, these circuit constructions have log-depth access costs. However, as laid out in Jaques & Rattew (2025), the fundamental issue regarding the practicality of QRAM comes down to the opportunity cost of the total energy required to implement a query to the QRAM. Precisely, given a QRAM with $N$ bits of memory, a QRAM is considered *passive* if and only if each query to the QRAM requires $o(N)$ *total* energy input. If the query instead requires $\Omega(N)$ energy input (even if the time complexity is $O(\text{polylog}(N))$) then the QRAM is *active*. Importantly, this means that any QRAM implemented in the error-corrected circuit-model must be active, as each qubit requires $O(1)$ classical resources to run the error-correction, resulting in an $\Omega(N)$ total energy cost per QRAM query. Even if error-correction is not used, if enacting the gates in the system requires constant energy input (e.g., by enacting the gates as laser pulses) then the QRAM will be active. If the QRAM is active, then Jaques & Rattew (2025) show that a wide-range of quantum linear algebra applications lose quantum speedup. Moreover, there are additional challenges such as how a noisy (non-error corrected) quantum memory could be interfaced with an error-corrected quantum processor.

However, as noted in Jaques & Rattew (2025) there is some hope in practice, and we will now outline their arguments. As an example, consider classical Dynamic Random Access Memory (DRAM). DRAM requires a constant power draw for each bit in memory, and thus an $N$-bit memory requires $\Omega(N)$ energy input. This makes DRAM active. Nevertheless, because the energy expenditure of DRAM is often dwarfed by the energy expenditure of the CPU accessing it, it is usually treated as being a passive component in classical algorithm design. For instance, Carroll & Heiser (2010) demonstrates that for mobile phones, "RAM power is insignificant in real workloads", and Mahesri & Vardhan (2005) draws a similar conclusion for laptops. At larger server-scales, the asymptotics of active memory become more noticeable, but memory still usually draws less power than the controlling CPU (Ahmed et al., 2021; Fan et al., 2007). Analogously, consider a regime where a QRAM is active, but its constant energy costs are extremely small relative to the energy costs of the error-corrected quantum computer it is being interfaced with. Given the *substantial* expected overheads of quantum error-correction (Babbush et al., 2021), the ratio of energy consumption for an error-corrected QPU to an active QRAM could be even more favourable than in the classical setting.

Then, if there is some way to interface this noisy device with the error-corrected QPU, for moderate scales (e.g., terabytes of memory), it is conceivable that the QRAM could be practically treated as passive. We will call this a "practically passive QRAM". Nevertheless, even though practically passive QRAMs are asymptotically active, they are unlikely to allow full error-correction without losing their constant advantages (unless, for some reason, the structure of QRAM allows for extremely efficient custom-made error-correcting codes). Consequently, it is important that the QRAM implementation is resilient to errors. Indeed QRAMs based on the bucket-brigade architecture (Giovannetti et al., 2008b), are intrinsically *exponentially* (in terms of the number of memory registers) robust to errors (Hann et al., 2021; Hann, 2021; Hong et al., 2012).

In this paper, for simplicity, when making a QRAM assumption we treat the QRAM as passive. We stress that substantially more work is needed to fully understand the feasibility of QRAM, but that it is plausible that the QRAM assumptions made in this paper could be physically realized in practice. In particular, assuming that truly passive QRAM is impossible, we outline the following questions (building on Jaques & Rattew (2025)) which could result in our results being practically useful. How can a noisy QRAM system be interfaced with an error-corrected quantum computer? If such an interface is possible, how do errors in the QRAM propagate through the error-correction in the QPU? Recent promising work (Dalzell et al., 2025a) provides answers to these two preceding questions, and offers a path forward for research aiming to construct practically passive and useful QRAM. Additional questions which need to be investigated to help realize a practically passive QRAM include some of the following. What is the ratio in energy consumption for plausible practically passive QRAM systems to the energy consumption of the controlling fault-tolerant QPUs for different error-correcting codes? Given potential active (practically passive) QRAM architectures, what is the total expected energy consumption for different sized memories?

### D.2 INPUT PREPARATION VIA QRAM

The data-structure due to Kerenidis & Prakash (2017) can allow for an arbitrary quantum state to be prepared, so long as the state amplitudes are made available through a specific QRAM data-structure.

**Lemma D.1** (Input Data QRAM Data-Structure (Kerenidis & Prakash, 2017)). *Let $N = 2^n$. Given a vector $\boldsymbol{x} \in \mathbb{R}^N$, we can define a data-structure utilizing a QRAM with $\tilde{O}(N)$ total qubits [7] storing $\boldsymbol{x}$. Then: (1) the cost to update (insert, delete, or modify) an entry $x_j$ is $O(n^2)$, (2) using the QRAM data-structure, the state $|x\rangle = \boldsymbol{x} / \|\boldsymbol{x}\|_2$ can be prepared by a circuit with depth $O(n^2)$, acting on $O(n)$ qubits.*

This is just a special case of the more general result in Kerenidis & Prakash (2017) giving a similar data-structure for arbitrary matrices (which we presented as QRAM for quantum data in Appendix A). Intuitively, the state can be prepared by following Grover-Rudolph (Grover & Rudolph, 2002), using the QRAM data structure containing the tree of binary partial norms of the vector to compute the controlled rotation angles for each additional qubit.

## E    ARCHITECTURES IN DIFFERENT REGIMES

As summarized in the main text, the results presented thus far can be used to construct a range of architectures in a number of different settings. In particular, we consider three regimes characterized by the QRAM assumptions they make. In the first regime, we assume that both the input to the network and the weights in the network are made available via QRAM. In the second regime, we assume that the network may use QRAM (since its QRAM data-structure may be pre-computed prior to inference-time), but that the input to the network is received classically and entirely on-the-fly, and thus that the input cannot be provided with QRAM (so a cost linear in the dimension of the input must be paid to load it into the quantum computer). In the third regime, we assume no QRAM. We will now expand on the arguments presented in the main text in greater detail.

### E.1    REGIME 1: INPUT AND NETWORK USE QRAM

Here we expand on the argument presented in Section 4.1.

---

[7]Neglecting the finite precision error due to storing vector elements (and their partial squared sums) in binary representations

**Online Input Construction**    Noting that as per Section D.2 QRAM data-structures can be efficiently updated, we note that there are a number of settings where it might be realistic for the input vector to be provided via QRAM. For example, in any setting where inference needs to be repeatedly performed on a slowly-changing input (e.g., in an interactive chat with an autoregressive LLM, where each output token becomes part of the new input), or where the input is the result of some other quantum algorithm. For example, in the context of auto-regressive interactive LLM (where the output would be a probability distribution over tokens instead of classes), the initial vector $x$ might be an encoding of the hidden prompt to the network (and so the associated data-structure can be pre-computed). As a user queries the LLM, a small number of tokens are added to $x$, and these updates can be efficiently performed to the data structure. Then, the network is run, and the new output token is added to $x$, again efficiently. This process can then continue to repeat, and so the cost of loading the data is either entirely precomputed, or amortized on-the-fly. We can envision similar applications in the classification of video, where a very large, but slowly-changing, video needs to be analysed one frame at a time. Here, a cost would need to be paid proportional to the number of changing pixels between each frame, and so the input data-structure could be efficiently updated. Additional settings where it might be reasonable for the input to be provided efficiently could be if the input corresponds to some combination of continuous function (via Rattew & Koczor (2022)), or if it was prepared as the output of some other quantum algorithm.

**Receptive Field**    To understand the importance of the final linear layer in the architecture for this regime, we must first summarize the receptive field problem of multi-layer convolutional architectures.

For simplicity, consider a 2D convolution with one input channel and one output channel, and consider a sequence of $k$ such convolutional layers. Let the kernel be $D \times D$. Since a convolutional layer can map the information in location $i, j$ to, at the furthest, the location $i + D, j + D$, after $k$ layers the information in any given entry will come from local information in the input at most $\approx kD$ pixels away. Consequently, the final layer which is input to the output linear-layer-residual block will contain features with $kD$ local information, which the linear layer then combines in a global fashion. We conjecture that having a full-rank layer at this stage is more effective for merging the local information than a similar dimension, but low-rank, linear layer. Since the cost of the quantum algorithm grows exponentially with depth, without the final linear layer, with such an architecture no learning could occur which requires global information from the input image.

Moreover, there are other approaches which could be taken to make the local information globally accessible to the earlier convolutional layers, potentially improving the power of such quantum-amenable architectures in practice. For instance, after a set number of convolutional layers, a linear layer could be added to make local information global (however, this damages the nice algebraic properties of convolutional layers). Alternatively, a sequence of convolutions can be implemented in each residual block (without activation functions between them) as this would not increase the complexity exponentially, potentially allowing for many more convolutions in sequence. Most appealingly, a solution can be found in the popular classical architecture of bilinear neural networks (Lin et al., 2015) (which forms the basis of the architecture presented for Regime 2). Here, paths of convolutional-based residual blocks are passed into a Kronecker product, which is followed by more layers. Via Lemma 3, we can efficiently do this in a quantum computer. Since the Kronecker product makes all local information globally available, it immediately solves the receptive field problem. However, while a Kronecker product makes local information globally accessible, it loses positional information. This can be resolved by enacting a positional encoding along one of the paths of the network prior to the product, e.g., as is done when Tokenizing the inputs to transformer architectures (Vaswani et al., 2017).

**Dequantization**    A number of quantum algorithms which were believed to have exponential speedups over their classical counterparts lost their exponential speedup after new classical randomized algorithms were developed which mirrored the quantum input assumptions. For example, see the works of Kerenidis & Prakash (2017) and Tang (2019). Indeed, it seems likely that, as was the case with the quantum CNN implementation in Kerenidis et al. (2020), that the convolutional residual blocks in our architectures could be dequantized (even though they make no QRAM assumptions). However, our new techniques enables the final linear-residual block to contain an arbitrary full-rank and dense matrix. Since known dequantization techniques require the matrix to be either low-rank (Chia et al., 2022; Tang, 2019) or sparse (with certain strong caveats) (Gharibian & Le Gall,

2023), existing techniques appear insufficient to dequantize our full architecture. Moreover, as previously discussed, removing the final linear layer introduces receptive field problems, highlighting that it is not a purely artificial addition to the network. Nevertheless, it would be interesting to exploring dequantizing the architecture without the final linear layer (or perhaps replacing it with a low-rank one), and this could result in some interesting techniques to classical accelerate inference for certain architectures.

### E.2  REGIME 2: NETWORK STORED IN QRAM, INPUT LOADED WITHOUT QRAM

See the discussion in Section 4.1.

### E.3  REGIME 3: NO QRAM

To reiterate, in this regime, both the matrix weights and the network input are not given by QRAM. We will now prove the complexity of the Regime 3 architecture shown in Figure 1 (c), as discussed in Section 4.1. We note that there are many simple modifications which could be made to this architecture, for example by having a final low-rank linear layer with $O(N)$ parameters. Adopt the notation used in Theorem 2. Let the input be a $4 \times M \times M$ tensor, and define $N = M^2$, $n = \log_2(N)$, $m = \log_2(M)$. Thus, the vectorized input is of dimension $O(N)$. Let $d$ be the number of paths into the input tensor (i.e., the latent dimension will be $O(N^d)$), as per Figure 1 (c). $T_X$ is the access cost of the input; in the QRAM-free regime we assume a worst-case of $T_X \in O(N)$. Let $C$ be the number of output classes (or set of possible output tokens).

Assume $d = 2$. Let $\delta > 0$ be an error parameter used only in the proof. Directly from the proof of Theorem 2, we have $U_{\text{conv}}$, a $(1, 2^k(63 + n), \delta)$-VE (vector encoding) for the $\ell_2$-normalized output of the $k$ convolutional/residual block layers. $U_{\text{conv}}$ has $O(\log(N/\delta)^{2k}(n^2 + T_X))$ circuit depth. Note that in that proof, $N$ corresponds to the vectorized dimension of the latent space (i.e., if there is 1 input and output channel, $N$ corresponds to the dimension of the vector acted upon by the matrix-form of the 2D convolution), and thus corresponds to $N^d$ here.

Let $|\phi\rangle$ represent the exact vector output after the sequence of $k$ convolutional layers. This VE corresponds to a state $|\tilde{\phi}\rangle$ such that $\||\phi\rangle - |\tilde{\phi}\rangle\|_2 \leq \delta$. Consequently, by Lemma C.1, sampling this VE (and applying the binning-protocol) yields a sample from a vector $\text{pool}_C(|\tilde{\phi}\rangle)$ such that $\|\text{pool}_C(|\phi\rangle) - \text{pool}_C(|\tilde{\phi}\rangle)\|_2 \leq \frac{2N^2\delta}{\sqrt{C}}$. Noting that the correct output of the network is given by $\boldsymbol{y} = \text{pool}_C(|\phi\rangle)$, we can get an overall error of $\epsilon$, such that the vector we sample from satisfies $\|\boldsymbol{y} - \text{pool}_C(|\tilde{\phi}\rangle)\|_2 \leq \epsilon$ by setting $\epsilon = \frac{2N^2\delta}{\sqrt{C}} \implies \delta = \frac{\epsilon\sqrt{C}}{2N^2}$. By plugging this into the circuit complexity of $U_{\text{conv}}$, and noting that here we assume we pay the full input dimension cost (since there is no QRAM), $T_X \in O(N)$, and so this simplifies to $O(N \log(N^3/\epsilon\sqrt{C})^{2k}) \in \tilde{O}(N \log(1/\epsilon)^{2k})$ total circuit cost. As stated in the main text, since the dimension of the vector acted on by the 2D convolution is $O(N^2)$ (when d=2), the classical cost to compute this is $\Omega(N^2)$: showing **a quadratic speedup over an exact classical implementation**. The speedup can be made asymptotically larger by increasing $d$.

**Possible Limitations**   Here we will outline some of the possible limitations of the architecture shown in Figure 1 (c). Since there is no final linear layer (in the architecture as directly presented), the receptive field problems outlined in Section 4.1 may appear to apply. However, by virtue of taking the tensor product of the input paths, local information becomes immediately globally accessible circumventing this limitation. Moreover, another way that local information could be made global is from the processing that occurs along each path prior to the tensor product, since there is no limit on the classical processing that can occur (so long as the total compute is linear in the dimension of the input).

Moreover, our argument against dequantization in the first regime (see Section 4.1) relies on the final dense and full-rank linear layer. However, since this layer is not feasible without QRAM, this argument does not apply here. However, as we are only suggesting a polynomial speedup in Regime 3, we do not expect a dequantized algorithm to completely close the performance gap past quadratic, as we benefit from amplitude amplification. However, exploring dequantized algorithms based on the ideas in this paper appears to be interesting subsequent work.

Finally, if the network only contains the convolutional layers, it will likely be very under-parameterized making training challenging (see e.g., Allen-Zhu et al. (2019) for a discussion on overparameterized neural networks). However, where the dimension of the vectorized input is $N$, it would be easy to add $O(N)$ parameters, either in the classical paths prior to the tensor product, or as a final low-rank residual output block (prior to the $\ell_2$-norm-pooling), so long as the number of parameters in that block are $O(N)$.

**Alternative: Parameterized Quantum Circuits as Network Layers**  Alternatively, one could use parameterized quantum circuits as network layers (Peruzzo et al., 2014; Benedetti et al., 2019b; Cerezo et al., 2021), as the number of parameters in such circuits are usually polylogarithmic in the dimension of the operator. However, such circuits are often hard to train even on classical machines, due to under-parameterization, the barren plateau problem (McClean et al., 2018; Larocca et al., 2025), and the exponential amount of bad local minima in the optimization landscape (Anschuetz & Kiani, 2022). However, given good enough initializations and warm start assumptions (Mhiri et al., 2025), it may still possible to train such architectures, leading to potential speed-ups in inference.

**Other Possible Sources of Speedup**  In some cases, where the input can be efficiently prepared without paying a dimension-dependent cost (e.g., the input comes from quantum states which are easy to prepare, either via some other quantum algorithm, or via techniques like Rattew & Koczor (2022)) it may be possible to obtain better than quadratic speedups. However, we leave this as a topic for future investigation.

# F  TECHNICAL RESULTS

We now report a result on the efficient polynomial approximation to the error function due to Low & Chuang (2017), which builds on the results of Sachdeva & Vishnoi (2014). This result is an improvement over the approximation obtained by an integration of the series expansion for the Gaussian distribution.

**Lemma F.1** (Polynomial Approximation to Error Function due to Corollary 4 of Low & Chuang (2017)). *Let $m \geq 1/2$, $1 \geq \epsilon > 0$. There exists a degree $k \in O(m \log(1/\epsilon))$ polynomial $P_{k,m}(x)$ such that*

$$P_{k,m}(x) := \frac{2me^{-m^2/2}}{\sqrt{\pi}} \left( I_0(m^2/2)x + \sum_{j=1}^{(k-1)/2} I_j(m^2/2)(-1)^j \left( \frac{T_{2j+1}(x)}{2j+1} - \frac{T_{2j-1}(x)}{2j-1} \right) \right)$$
(F.1)

*and $\max_{x \in [-1,1]} |\operatorname{erf}(mx) - P_{k,m}(x)| \leq \epsilon$. Let $1 \geq c > 0$. Alternatively, if $k \in O(m \log(mc/\epsilon))$, then $\max_{x \in [-c,c]} |\operatorname{erf}(mx) - P_{k,m}(x)| \leq \epsilon$. Additionally, for all $k$, $\max_{x \in [-1,1]} |P_{k,m}(x)/x| \leq \frac{4m}{\sqrt{\pi}}$, and $P_{k,m}(0) = 0$. Finally, $\min_{x \in [-1,1]} |\operatorname{erf}(mx)/x| \geq 1/2$, and $\operatorname{erf}(mx)$ has Lipschitz constant $L = \frac{2m}{\sqrt{\pi}}$,*

*Proof.* For the case where $\max_{x \in [-1,1]} |\operatorname{erf}(mx) - P_{k,m}(x)| \leq \epsilon$, the result on the polynomial approximation is directly taken from Low & Chuang (2017). We will now prove the bound when the function is constrained to the interval $[-c, c]$. Let $\epsilon_1 := \max_{x \in [-c,c]} |\operatorname{erf}(mx) - P_{k,m}(x)|$. From Equation (71) of Corollary 4 of Low & Chuang (2017), for a degree $k$ polynomial approximation, we have the following error-bound,

$$\epsilon_1 \leq \frac{2me^{-m^2/2}}{\sqrt{\pi}} \left| \sum_{j=(k+1)/2}^{\infty} I_j(m^2/2)(-1)^j \left( \frac{T_{2j+1}(x)}{2j+1} - \frac{T_{2j-1}(x)}{2j-1} \right) \right|.$$
(F.2)

Using the identity $\left( \frac{T_{2j+1}(x)}{2j+1} - \frac{T_{2j-1}(x)}{2j-1} \right) = 2 \int_0^x T_{2j}(t) dt$, and using the fact that all Chebyshev polynomials of the form $T_{2j}$ are even, we can get the bound that $2 \left| \frac{T_{2j+1}(x)}{2j+1} - \frac{T_{2j-1}(x)}{2j-1} \right| \leq 2 \int_0^{|x|} |T_{2j}(t)| dt \leq 2|x| \leq 2 \max_{x \in [-c,c]} |x| = 2c$, since $\max_{x \in [-1,1]} |T_{2j}(x)| \leq 1$.

Then, applying the triangle inequality, Equation (F.2) becomes,

$$\epsilon_1 \leq \frac{4cme^{-m^2/2}}{\sqrt{\pi}} \sum_{j=(k+1)/2}^{\infty} \left| I_j(m^2/2) \right|. \tag{F.3}$$

Define $\epsilon_{gauss,\gamma,k}$ as per Corollary 3 of Low & Chuang (2017). Define some $\epsilon' > 0$. Note that $\epsilon_{gauss,\gamma,k} = 2e^{-\gamma^2/2} \sum_{j=\frac{n}{2}+1}^{\infty} |I_j(\gamma^2/2)|$, and that $\epsilon_{gauss,\gamma,k} \leq \epsilon'$ if $k \in O(\sqrt{(\gamma^2 + \log(1/\epsilon'))\log(1/\epsilon')})$. Thus, $\epsilon_1 \leq \frac{2cme'}{\sqrt{\pi}}$. To get an overall error-bound of at most $\epsilon$, we can set $\frac{2cme'}{\sqrt{\pi}} = \epsilon$, and so $\epsilon' = \frac{\sqrt{\pi}\epsilon}{2cm}$. Thus, if we set $k \in O(m\log(\frac{cm}{\epsilon}))$, we are guaranteed that $\max_{x \in [-c,c]} |P_{k,m}(x) - \text{erf}(mx)| \leq \epsilon$.

Next, $\frac{d}{dx}\text{erf}(mx) = \frac{2m}{\sqrt{\pi}}e^{-(mx)^2}$, and consequently the maximum value of the derivative of the function is when $x = 0$, i.e., $\max_{x \in [-1,1]} |\frac{d}{dx}\text{erf}(mx)| = \frac{2m}{\sqrt{\pi}}$.

We will now prove that $|P_{k,m}(x)/x| \leq \frac{4m}{\sqrt{\pi}}$ and $\min_{x \in [-1,1]} |\text{erf}(mx)/x| \geq 1/2$.

Noting that $P_{k,m}(0) = 0$, (since for $x = 0$, $T_{2j}(x) = \cos((2j+1)\arccos(0)) = \cos((2j+1)\pi/2) = 0$), by Lipschitz continuity we have that $|P_{k,m}(x)/x| \leq |\frac{d}{dx}P_{k,m}(x)|$. Noting that $\frac{d}{dx}\frac{1}{2}(\frac{T_{2j+1}(x)}{2j+1} - \frac{T_{2j-1}(x)}{2j-1}) = T_{2j}(x)$,

$$\max_{x \in [-1,1]} |P_{k,m}(x)/x| \leq \max_{x \in [-1,1]} \left| \frac{d}{dx}P_{k,m}(x) \right| \tag{F.4}$$

$$= \max_{x \in [-1,1]} \left| \frac{2me^{-m^2/2}}{\sqrt{\pi}} \left( I_0(m^2/2) + 2 \sum_{j=1}^{(k-1)/2} I_j(m^2/2)(-1)^j T_{2j}(x) \right) \right|. \tag{F.5}$$

A common identity for modified Bessel functions of the first kind states for $t \neq 0$, $e^{\frac{1}{2}y(t+t^{-1})} = \sum_{j=-\infty}^{\infty} t^j I_j(y)$. Setting $t = 1$, we find $e^y = \sum_{j=-\infty}^{\infty} I_j(y)$. Moreover, since $I_j(y) \geq 0$ for all $y > 0$, $\sum_{j=1}^{(k-1)/2} I_j(m^2/2) \leq e^{m^2/2}$. Thus, using that $\max_{x \in [-1,1]} |T_{2j}(x)| \leq 1$,

$$\max_{x \in [-1,1]} |P_{k,m}(x)/x| \leq \frac{4me^{-m^2/2}}{\sqrt{\pi}} \sum_{j=1}^{(k-1)/2} I_j(m^2/2) \leq \frac{4m}{\sqrt{\pi}}. \tag{F.6}$$

Thus, it is clear that this upper-bound is independent of the degree of the polynomial approximation, and thus applies to the whole interval $x \in [-1, 1]$ and not just $x \in [-c, c]$.

Finally, we must show that $\min_{x \in [-1,1]} |\text{erf}(mx)/x| \geq 1/2$. First, note that $|\text{erf}(mx)/x|$ is symmetrical, so we can simply consider the interval $x \in [0, 1]$. Moreover, it is monotonically decreasing, so we can take the endpoint $\min_{x \in [-1,1]} |\text{erf}(mx)/x| = \text{erf}(m)$. Since $m \geq 1/2$, $\text{erf}(m) \geq \text{erf}(1/2) \approx 0.52 > 1/2$. $\qquad \square$

