# OpenReview forum: "Accelerating Inference for Multilayer Neural Networks with Quantum Computers"
_ICLR.cc/2026/Conference — ICLR 2026 Poster_

### Official Review · Reviewer_JGKB · 2025-10-25

**Soundness:** 3
**Presentation:** 2
**Contribution:** 2
**Rating:** 2
**Confidence:** 4

**Summary:**

This paper proposes a fully quantum method for simulating the inference process of multi-layer neural networks. The authors claim that their approach can achieve exponential speedup when using QRAM, while still maintaining a quadratic speedup in QRAM-free scenarios. The work provides a potential pathway for leveraging fault-tolerant quantum computers to perform inference tasks. However, for the reasons outlined in the Weaknesses section, I do not believe this paper merits acceptance at the ICLR conference.

**Strengths:**

The authors address one of the most interesting topics in the field of quantum computation—using quantum computers to accelerate classical AI. They also provide a clear summary of previous results and explain how their work relates to them.

**Weaknesses:**

1. The manuscript is not well written: many complex definitions and results are deferred to the appendix. The paper would be more suitable for a journal without strict page limits.

2. Since the QRAM is a very strong assumption, which may require 2^n circuit depth to prepare a target state or the block-encoding of the matrix, all quantum speed-up results based on the QRAM assumption have very limited practical significance. For example, if a practical quantum computer with error-correction function were provided, can authors provide an analysis on how many single- double- quantum gates are required to achieve a QRAM oracle? Moreover, if the QRAM assumption is permitted, classical L2-norm based sampling methods (dequantization algorithm proposed by E. Tang, Nat. Rev. Phy. 4, 692–693, 2022) may achieve similar speedups for low-rank cases. Consequently, I am not convinced that the QRAM-based theoretical results in this paper make a sufficiently strong contribution to the community.

3. Following the above concern, the only potentially valuable contribution of the paper appears to be the QRAM-free scenario discussed in Lemma 5 and Section 4.1.3. However, I cannot follow the argument justifying this claim. In section E.3, the authors claimed that quantum circuit U_{conv} has approximate  O(log(N)T_X) circuit depth, which is central to the claimed quadratic speedup. Tracing this statement back to Theorem 2, I still find the justification unclear: the authors do not provide a convincing derivation or explanation for this depth bound. Please clarify the derivation and state precisely which assumptions are required.

4. A minor error: the author list for this reference is not correct: Towards provably efficient quantum algorithms for large-scale machine learning models. Nature Communications, 15:434, 2024.

**Questions:**

Please refer to the weakness section.

---

> ### Author Response · Authors · 2025-11-27
>
> We would like to thank the reviewer for taking the time to consider our paper and provide feedback. We also thank the reviewer for acknowledging that "The authors address one of the most interesting topics in the field of quantum computation -- using quantum computers to accelerate classical AI".
>
> We believe that the concerns about clarity are easily addressible, and we have outlined key changes that we have made to enhance clarity and general accessibility. We also believe that we have directly answered the other points raised, and thank the reviewer for their thoughtfulness.
>
>
> Weaknesses:
>
> 1. *The manuscript is not well written: many complex definitions and results are deferred to the appendix. The paper would be more suitable for a journal without strict page limits.*
>
>
> **Why many details are in the appendix.** As a theory-oriented submission, our goal was to keep the main paper focused on the algorithm ideas and complexity guarantees, while moving long technical proofs and auxiliary lemmas to the appendix. This is the standard organization for theoretical papers at ICLR and similar venues. The main text states the problem (Definition 1), describes the architectures at a high level (Figure 1 and Section 4), and presents the key results and their implications. The appendix simply contains the detailed derivations needed to formally prove these results. Therefore, we believe our work does fit the conference format, despite the length of the technical development. If a particular definition or result would be more appropriately placed in the main text (as opposed to the appendix), we would be happy to move it, and would welcome a specific pointer from the reviewer.
>
>
> **Concrete changes to improve readability.** In response to this and related comments, we have revised the manuscript to make the main narrative more accessible. In particular, we have provided additional intuition for some of the formal result statements, and have provided substantially more intuition regarding the derivations of the results for the three regimes. As an example, please see our following discussion responding to your question regarding our derivation of the circuit complexity of $U_{conv}$ for Regime 3. The updated document containing these changes (and other) will be uploaded soon.
>
>
> **Overall assessment of the writing.** We respectfully note that all the other reviewers rated our presentation as "Good", with Reviewer LfKb writing "Clear organization: The text is logically structured, with well-defined assumptions and computational models", and Reviewer yibn writing, "In terms of presentation, the paper is mostly very well-written and clearly presented (except for some ambiguous mathematical statements)".
>
> We hope that, together with the revisions outlined above, we have alleviated the concern that the paper is "not well written", or that it is not suitable for this venue.

---

> ### Author Response · Authors · 2025-11-27
> **Response to QRAM Assumption**
>
> 2. *Since the QRAM is a very strong assumption, which may require 2^n circuit depth to prepare a target state or the block-encoding of the matrix, all quantum speed-up results based on the QRAM assumption have very limited practical significance. For example, if a practical quantum computer with error-correction function were provided, can authors provide an analysis on how many single- double- quantum gates are required to achieve a QRAM oracle? Moreover, if the QRAM assumption is permitted, classical L2-norm based sampling methods (dequantization algorithm proposed by E. Tang, Nat. Rev. Phy. 4, 692–693, 2022) may achieve similar speedups for low-rank cases. Consequently, I am not convinced that the QRAM-based theoretical results in this paper make a sufficiently strong contribution to the community.*
>
> We thank the reviewer for raising the important issue of QRAM feasibility, and for pointing out the relation to dequantization. We discuss QRAM extensively in Appendix D, and we discuss dequantization in Appendix E.1 (we point to both in the main text). However, given the importance of these issues, we will expand the discussion of QRAM feasibility and dequantization in the main text, in addition to the more detailed treatment in Appendices D and E.1.
>
> In the following we will explicitly address each of the points raised by the reviewer.
>
>
> Our results in the "full QRAM" regime rely on the standard QRAM construction (e.g., bucket brigade) which (if implemented in the circuit model, which is not the efficient way to implement QRAM) has depth $O(\text{polylog}(N))$. However, circuit depth is not the main challenge for a fault-tolerant QRAM. As we discuss extensively in Appendix D, any fully error-corrected circuit implementation of an $N$-qubit QRAM requires $\Omega(N)$ physical operations in total (even in the highly parallelized and shallow implementations), which introduces substantial opportunity cost: the corresponding classical control and error-correction resources could instead be used to run a parallel classical algorithm. For this reason, we view a large-scale *practically useful QRAM* as more likely to be realized as a passive, noisy physical device with sublinear energy input (e.g., $O(\text{polylog}(N))$), coupled to an error-corrected processor via teleportation and distillation. For example, see the recent adapative-on-the-fly distilation-teleportation setup as shown by the recent Amazon AWS paper (arxiv:2505.20265). Additionally, the bucket-brigade architecture implies that any such noisy physical QRAM device, surprisingly, only needs physical error rates which scale as a polylogarithmic function of the total memory size. Nevertheless, getting such a passive QRAM device to work is an active area of research, but has by no means shown to be impossible. Consequently, we believe that our results in the regimes that use QRAM could still be of substantial potential importance, should such passive QRAM devices be realized. This realistic set of considerations and architectural constraints is exactly the picture we discuss in Appendix D, which underlies our interpation of the QRAM based speedups which we present, and are complementary to the more conservative QRAM-free results.
>
> On the topic of dequantization, we agree it is very important to consider comparisons to the best possible classical randomized algorithms. Indeed, we consider such a comparison in Appendix E.1. The key reason that we argue that our architectures in Regimes 1 and 2 cannot be dequantized with existing techniques is that all existing dequantization techniques require the encoded operator to be either low-rank or sparse. Our final architectural block, the full-rank and dense mat-vec-square block, by virtue of our new algorithmic contributions, is not compatible with any known dequantization technique, and moreover it may actually be possible to prove that it cannot be dequantized through a reduction to Fourier sampling. However, such a rigorous proof is beyond the scope of our current work, and we feel it is sufficient to observe that existing techniques cannot be immediately applied to our results.
>
> **In summary, we hope that we have shown that our results in Regimes 1 and 2 (the regimes using QRAM) still have substantial merit, and that importantly, they do not make any non-standard assumptions. There are a range of papers in ICLR and other similar venues with similar QRAM assumptions, and we believe that we have analyzed these assumptions at a much greater depth than is standard for such an algorithms paper.**

---

> ### Author Response · Authors · 2025-11-27
> **Additional High-Level Intuition and Derivation of Main Architectural Results [Part 1]**
>
> 3. *Following the above concern, the only potentially valuable contribution of the paper appears to be the QRAM-free scenario discussed in Lemma 5 and Section 4.1.3. However, I cannot follow the argument justifying this claim. In section E.3, the authors claimed that quantum circuit U_{conv} has approximate O(log(N)T_X) circuit depth, which is central to the claimed quadratic speedup. Tracing this statement back to Theorem 2, I still find the justification unclear: the authors do not provide a convincing derivation or explanation for this depth bound. Please clarify the derivation and state precisely which assumptions are required.*
>
> While we understand why the reviewer may feel that "the only potentially valuable contribution of the paper appears to be the QRAM-free scenario discussed in Lemma 5 and Section 4.1.3", we hope the above discussion has demonstrated why this is not the case -- if a passive QRAM (or a practically passive QRAM) can be realized, then the asymptotic speedups proven in the first two regimes could be of substantial impact.
>
> In the following argument, we emphasise that the proof in Regime 3 simply follows immediately by the intermediate result in Regime 1. We will now clearly provide all the high-level intuition required to understand the proofs yielding $U_{conv}$ (for all three regimes).
>
> First, at a high-level, the core operation in Regimes 1-3 is the sequence of $k$ residual blocks presented as Lemma 7 (which, when applied to a processed input vector, yields $U_{conv}$). This result just follows from invoking Lemma 6 (the general residual block result) $k$ times, combined with the QRAM-free convolution block-encoding (Lemma 5). Consequently, the key to understanding $U_{conv}$, is to understand the initial mapping of the input vector (which in Regime 1 is simply concatenation, and in Regime 3 is concatenation + efficient classical processing + tensor), and then to understand Lemmas 5 and 6 (which are identical when used for Regimes 1-3).
>
> Lemma 5 is technically involved but conceptually straightforward and hence the proof is well suited to being delegated to Appendix B.2. The key intuition regarding its efficiency is that the number of parameters for a given convolution is independent of the dimension of the vector that convolution is being applied to, and that the quantum complexity is polynomial in the number of parameters but polylogerithmic in the dimension of the vector. E.g., with UNet style architectures, the convolutional kernel is often e.g., $3\times 3$ (if there is 1 input and output channel), clearly a small constant. Thus, intuitively, it seems clear that without using QRAM, where the convolution (with a kernel with $P$ parameters) is applied to a vectorised input of dimension $N$, the complexity of implementing it should be lower-bounded by $\tilde\Omega(P)$. Indeed, our result follows this intuition closely. At a high-level, we show that one can efficiently construct a block-encoding corresponding to the matrix-form of the convolution corresponding to each individual parameter, then by loading the values of each parameter via a brute-force unitary which has cost linear in the number of parameters (and thus avoiding QRAM), a linear combination can then be taken to obtain the overall QRAM-free block-encoding. This intuition regarding the efficiency of our QRAM-free block encoding can probably be stated in a clearer form in the main text where we introduce Lemma 5, and we will do so if you believe it increases the clarity of our results.
>
> Lemma 6 follows simply by applying our results on summing encoding vectors to enact skip connections (Lemma 1), matrix-vector multiplication (Lemma 2), activation function application (Lemma B.19, a straight-forward improvement of the result in prior work), and $\ell_2$-normalisation (Lemma B.8, also a straight-forward improvement over a result from prior work). Noting that Lemma B.8 and Lemma B.19 are mild improvements over results from prior work, we feel it’s suitable leaving them in the appendix. Care is needed to ensure that norm lower-bounds are carefully tracked, but these are mechanical details which we also feel safe in delegating to Appendix C, as we provide the important intuition and ideas in the main text.

---

> > ### Author Response · Authors · 2025-11-27
> > **Additional High-Level Intuition and Derivation of Main Architectural Results [Part 2]**
> >
> > Then, since Lemma 7 follows by simply applying Lemma 6 to itself k times, to obtain $U_conv$ one needs only consider the input vector to the network, and then combine Lemma 7 with the convolution block-encoding from Lemma 5. Lemma 7 and Lemma 5 are combined by simply replacing the general weight matrix $W$ used in each residual block provided in Lemma 6 with the convolution block-encoding provided in Lemma 5. Combining this with the input state (which we will now describe) immediately yields $U_{conv}$. Consequently, given a state preparation unitary $U_{X}$ which yields the (vectorised) tensor input $|X\rangle$ with $O(T_X)$ circuit complexity, we then obtain another state preparation unitary $U_{X’}$ with $O(T_X)$ circuit complexity which corresponds to concatenating $|X\rangle$ with itself some constant number of times (4 times in Theorem 2). At a high-level (excluding any additional classical pre-processing performed on $|X\rangle$ in Regime 3, which by formulation does not change the asymptotic complexity), the only difference between $U_{conv}$ in Regimes 1 and 3, is that in Regime 3 $U_{X’}$ is then tensored with itself $d$ times, which for a constant $d$ still has $O(T_X)$ circuit complexity (by Lemma 3). Consequently, *the only difference* in the complexity of $U_{conv}$ between Regimes 1 and 3, is that with QRAM $T_X \in O(\log N)$, while without QRAM $T_X \in O(N)$.
> >
> > We will now combine all the above intuition to show that the proof yielding $U_{conv}$ is actually quite simple. Our only assumption is that the number of parameters in each 2D multi-filter convolution is constant (at least relative to the dimension of the vectorised input the convolution acts on), which is an accurate assumption for essentially every convolutional architecture used in machine learning. Then, noting that the parameters C and D in Lemma 5 are constants (in Theorem 2, $C=16$, and $D=3$), the circuit depth of the convolutional block-encoding circuit is simply O(\log M) where M is the dimension of the vectorized input acted upon by the sequence of k residual blocks. Combining this with Lemma 7, we find that the total circuit complexity of the sequence of k residual blocks is approximately $\tilde O(\text{polylog}(\sqrt{M}/\epsilon)^{k} T_X)$. The only piece missing is to consider the dimension of the input vector. Since the number of concatenations is constant, for Regime 1, $M \in O(N)$ (where $N$ is the vectorized dimension of $|X\rangle$). For Regime 3, the input is tensored $d$ times, and thus $M \in O(N^d)$, which when $d=2$ yields $M \in O(N^2)$. Plugging these values in gives the results stated in the paper. For example, for Regime 3 this immediately gives the $\tilde O(\text{polylog}(N/\epsilon)^{k} T_X)$ you refer to.
> >
> > **All this is to say that the proof yielding $U_{conv}$ in Regime 1 is identical to the technique which yields $U_{conv}$ in Regime 3, so long as the differences in the input vectors in the two regimes are carefully handled, which from a computational complexity perspective is simply a matter of accounting for the differences in their dimension due to the tensor product mapping.**
> >
> > As you can see, once the intuition is clear, the actual proof using our lemmas fits in a paragraph, and so we feel that this is suitable for this venue, however, we agree with you that we should state the intuition as described above more clearly in the main text, and we sincerely thank you for pointing out this opportunity to improve the clarity of our paper.

---

> ### Author Response · Authors · 2025-11-27
> **Correction of minor error**
>
> 4. *A minor error: the author list for this reference is not correct: Towards provably efficient quantum algorithms for large-scale machine learning models. Nature Communications, 15:434, 2024.*
>
> We sincerely thank the reviewer for pointing out this oversight. We have corrected this, and verified that we have not made any other errors inputting citation author names. We have also added some additional references.
>
> We will soon upload a version of the document containing all the corrections, additions and clarifications outlined above.

---

> > ### Comment · Reviewer_JGKB · 2025-11-28
> > **Reply to authors**
> >
> > I thank the authors for their effort in clarifying the proofs and revising the manuscript. I will consider to increase the score.
> >
> > Best!

---

> > > ### Author Response · Authors · 2025-12-03
> > >
> > > We would like to thank you for your review, and for your response. We appreciate your time, and your consideration in increasing our score.
> > >
> > > Kind regards,
> > >
> > > The Authors

---

### Official Review · Reviewer_yibn · 2025-10-28

**Soundness:** 2
**Presentation:** 3
**Contribution:** 3
**Rating:** 6
**Confidence:** 2

**Summary:**

The paper studies the important task of constructing quantum implementations of neural networks, with the goal of achieving provable speed-ups in neural network inference due to quantum phenomena. The paper considers convolutional ResNets and provides an extensive theoretical study, covering different types of assumptions about quantum data access.

**Strengths:**

While several previous works have studied this problem for different types of architectures (Kerenidis et al. for CNNs, Guo et al. for transformers), the paper provides important advances by allowing "coherent" multi-layer archiectures and non-linearities, where no readout or tomography is required. Employed methods build on previous work by Rattew & Rebentrost on Quantum Vector-Encodings, but extend these in various directions. Considered ResNet architectures are mostly practically relevant, thereby contributing to the building quantum subroutines for provable speed-ups for neural network-based learning.
In terms of presentation, the paper is mostly very well-written and clearly presented (except for some ambiguous mathematical statements).

**Weaknesses:**

There are still several important points related to relevance of assumptions and soundness of the claims / mathematical statements that I think need to be discussed thoroughly:

- Choice of activation function: the introduction and abstract state that the deep learning architectures use sigmoid activation functions. However, throughout the paper (Lemma 6, Theorem 2 and Figure 1) it seems that only "erf" is used as activation function. This is not a commonly used activation function due to lack of closed-form expression.
- Circuits depend on input: a fundamental issue appears to be that the circuits depend on $x$. This issue already appears in the problem statement in Definition 1. The statement suggests that for a fixed $x$ you aim to find $\hat{y}$ with $|y-\hat{y}| \leq \epsilon$? But in practice you would want to vary $x$. This seems to become a central issue in the main result in Theorem 2: the unitary matrix $U_X$ there depends on the value of the input, which then also seems to affect the overall circuit construction. Looking at the proof, lines 397-399 suggest that the circuit would need to change whenever a new input data point is processed in the network.
- Clarity of mathematical statements: the main findings of the paper are weakened by some unclear mathematical statements. For example,  in Definition 1 it is not clear what is meant by "return a sample from some probability vector $\hat{y}$ ...". Is this a random vector from which you randomly sample a realization? If yes, then with a certain probability the sample may not have the desired property. Can you quantify this probability? The same issue appears in the main result in Theorem 2: the statement does not relate the constructed quantum circuit with the vector $\tilde{y}$. Can you be more clear on how this vector is drawn? And what do you mean here by "we can draw a sample" (same question as before)?
- Assumptions on input: In Theorem 2, the input matrix is assumed to have a representation by a unitary matrix (see lines 378-380). Can you comment on this assumption? Is it easy to shown to be satisfied in certain situations?

**Questions:**

- Could you obtain the same results for more common choices of activation functions?
- Lemma 5: the Lemma requires that the convolutional matrix is normalized, which is rarely the case in practice. Is there a way of absorbing this, e.g., into the input?
- Can you provide more insights on what the Assumptions on the input mean in Theorem 2?
- From the statement of Theorem 2 it appears that for each new data point, you would need to construct a new circuit. Can you comment on this?
- Can you clarify on what you mean by "return a sample from some probability vector"? This is very important, otherwise the main statement is not clearly understandable.
- Can you clarify in Theorem 2 how the sample is obtained from the circuit?
- The neural network sizes assumed in Theorem 2 are very specific (you assume 16 input channels, output channels etc.) Why is this needed? It would be more convincing  to have a result for general sizes of channels.

A few minor points on clarity:

- The legend of figure 1 is a bit confusing, as it says "The figure shows the architectures we consider ... under thre different regimes of quantum data access assumptions". However, the figure does not contain any clues about how quantum data access differs in the different subfigures. Could you clarify this?
- The legend also states that "the input is assumed to be a rank-3 tensor (e.g. images with 4 channels)". Wouldn't this correspond to images with 3 channels?
- In lines 108/109 epsilon appears without having been introduced.
- In Definition 1 the role of the error epsilon is not mentioned.

---

> ### Author Response · Authors · 2025-11-27
>
> We sincerely thank the reviewer for taking the time to review our work. We also thank them for acknowledging that "the paper provides important advances".
>
>
> In the following, we will group the related weaknesses and questions raised and address them together. As a consequence of this review, we have made a number of important clarity improvements to the document which we will also now outline.
>
>
> Weaknesses:
>
> 1. *Choice of activation function.*
>
> Our results selected erf, a type of sigmoid activation function, because it has straight-forward analytical properties and polynomial approximations. However, our results are generally applicable to any activation function which admits efficient polynomial uniform approximations, satisfy $f(0)=0$, and can lower-bound the norm of any vector after application (although this last constraint is likely not necessary in practice, but is simply a helpful fact to enable analytic guarantees).
>
> There are a number of common activation functions which can immediately replace our usage of erf without requiring any other changes. For instance, we could immediately replace erf with the popular GeLU activation, noting that $\text{GeLU}(x) = \frac{x}{2}(1 + \text{erf}(x/\sqrt{2}))$. Our techniques also similarly immediately work with e.g., tanh.
>
>
>
> 2. *Circuits depend on the input $\vec x$.*
>
> These are great questions, and directly touch the key underlying idea which enables us to allow a unitary matrix to enact a non-linear transformation. In order to enact a non-linear transformation with a unitary matrix, the defintion of the unitary matrix *must* depend on the input it is being applied to (otherwise the operator is linear and cannot enact a non-linear transformation). As a simple example, given some vector $\vec x$, define $A = \text{diag}(\vec x)$, and note that $A \vec x = (\vec x)^2$ (where the square is applied element-wise), and so clearly a linear operator (a matrix) has enacted a non-linear transformation, because its definition depends on the vector it is being applied to. Consequently, the circuit enacted needs to be computed on the fly for each input it is applied to, however, since the circuits are compact this does not asymptotically change the complexity of the algorithm. In the absolute worst case, computing the circuit description would merely double the total compute performed by the algorithm. We briefly mention this in the main text, but will clarify this further.

---

> ### Author Response · Authors · 2025-11-27
>
> 3. *Clarifying the meaning of Definition 1 as drawing a sample from a probability vector, and clarifying how in Theorem 2 a sample is obtained from the circuit?*.
>
>
> First, we answer the question regarding Definition 1 as drawing a sample from a probability vector.
>
> Definition 1 assumes that we are given a neural network, which given an input $\vec x$ returns an output $\vec y$. Moreover, we assume that the output is a probability vector (i.e., all the entries of $\vec y$ are non-negative, and they sum to $1$). For instance, in the case of image classification, $\vec x$ would be a vectorized image tensor, and $\vec y$ would be a probability distribution over classes. E.g., for CIFAR10, $\vec y$ would have $10$ entries, with $y_0$ corresponding to the probability of assigning the class cat, $y_1$ corresponding to the probability of assigning the class dog, etc. Sampling from $\vec y$ simply means assigning a class with probability corresponding to the associated entry in the probability vector. In the case of autoregressive next token prediction, the probability distribution would instead be over the set of possible tokens, and sampling would mean selecting a specific next token with probability given by the output vector.
>
> The introduction of $\epsilon$ is intended to allow the algorithm to account for some error in the implementation of the underlying neural network. We note that since our algorithm has polylogarithmic dependence on $\epsilon$, one can treat $\tilde y$ and $y$ to be essentially equal (one can efficiently set $\epsilon$ to be extremely small), and so our network simply assigns classes corresponding to how the network was trained. To be clear, a polylogarithmic $\epsilon$ dependence is roughly equal to the asymptotic error dependence introduced by finite precision arithmetic.
>
> Our proof shows that our algorithm essentially draws the exact same samples as exactly running the network on a classical computer. The quality of these samples depends on how well the network was trained. Informally, for image classification, for a "well-trained" network, one can be confident in correctly assigning the correct class by simply drawing some constant number of samples and taking the majority vote. We can elaborate on this further, but note that these questions also apply to classical neural networks (e.g., once you've trained your network, what is your procedure for actually assigning a class from the output distribution over classes for a given input). Quantumly, one could use infinity-norm tomography techniques to e.g., output the class with the highest probability, but again such considerations are beyond the scope of the present work, as we are focused strictly on accelerating a given neural network architecture for inference as defined in the general statement of Definition 1.
>
> Now we sketch how Theorem 2 produces samples in accordance with Definition 1.
>
> The quantum algorithms in Theorem 2 produce a coherent quantum state which corresponds to the vectors in Figure 1 directly prior to the L2 pooling operation. Consequently, sampling is performed by measuring this quantum state, and then applying a simple binning protocal to map the measured basis vector to the output which corresponds to the true output after performing l2 norm pooling. This is proven to be directly equivalent to sampling (up to some exponentially suppressible error) to the exact output of the neural network. This allows us to perform the l2 norm pooling operation without increasing the complexity of the quantum circuit. Where the true output of the network is $\vec y$, our quantum sample corresponds to sampling from $\tilde y$.
>
>
>
> 4. *Assumptions on input: In Theorem 2, the input matrix is assumed to have a representation by a unitary matrix (see lines 378-380). Can you comment on this assumption? Is it easy to shown to be satisfied in certain situations?*
>
> This is fully general and imposes no restriction on the inputs (other than that they must be $\ell_2$-normalized). When QRAM is used for the input (Regime 1), an arbitrary input can be directly prepared. When QRAM is not used for the input (Regimes 2 and 3) a brute-force unitary can be constructed yielding any arbitrary input as a quantum state, and our results already take into account the complexity of this procedure.

---

> ### Author Response · Authors · 2025-11-27
>
> *We will now answer the questions which were not directly addressed above.*
>
>
> *Question 2. Lemma 5: the Lemma requires that the convolutional matrix is normalized, which is rarely the case in practice. Is there a way of absorbing this, e.g., into the input?*
>
> There are two options: the normalization factor of the convolution can be pre-computed after the network is pre-trained, and then the input to the next activation function could be scaled to cancel out the normalization factor (thus allowing any not necessarily normalized convolution to work) -- this is exactly the intuition of absorbing the normalization factor which you state. However, this somewhat complicates the theoretical analysis (although we expect would work in practice), and so for simplicity we assume the convolution is normalized. Alternatively, one can use a range of standard spectral regularization technniques during training to enforce this constraint.
>
>
> *Question 3. Can you provide more insights on what the Assumptions on the input mean in Theorem 2?*
>
> In Theorem 2, the only assumption placed on the input is that it is $\ell_2$-normalized, and that it is an image (with a transparancy -- alpha/null -- channel). If the image has no transparancy, then the null channel can simply be set to all 0, as we assume in the theorem. There are no other assumptions; *any possible image*, so long as it is normalized, works.
>
>
> *Question 7. The neural network sizes assumed in Theorem 2 are very specific (you assume 16 input channels, output channels etc.) Why is this needed? It would be more convincing to have a result for general sizes of channels.*
>
> These specific values were selected arbitrarily to simplify the theorem statement (all the intermediate results work in full generality). One can easily convert Theorem 2 to work with arbitrary channel sizes (and would have complexity polynomial in the channel size), and we can make a note of this in the paper. However, since the Theorem is already quite dense, by selecting a constant number of channels, we intended to make the result statement a bit more clear. The only real constraint is that we assume the number of channels are a power of 2. This could be relaxed, but would require some further work to modify the convolution block-encoding.
>
>
>
> **A few minor points on clarity:**
>
>
> We have updated the manuscript to incorporate all the raised clarity points. We thank you for raising them. As a small note responding to "The legend also states that "the input is assumed to be a rank-3 tensor (e.g. images with 4 channels)". Wouldn't this correspond to images with 3 channels?", an alpha RGB $H\times W$ image corresponds to a rank 3 tensor with dimensions $4 \times H \times W$. Here, the number of channels in the image (e.g., an alpha channel, a red channel, a green channel and a blue channel) corresponds to the dimension along the $0$ axis of the rank-3 tensor. Intuitively, this can be visualized as a stack of 4 $H\times W$ matrices.

---

> ### Comment · Reviewer_yibn · 2025-11-28
>
> I would like to thank the authors for carefully replying to my questions. This addresses all my concerns and I will increase my rating to recommend acceptance.

---

> > ### Author Response · Authors · 2025-12-03
> >
> > We would like to thank you for your time, and for reviewing our paper. We are glad that we have addressed your concerns.
> >
> > Kind regards,
> >
> > The Authors

---

### Official Review · Reviewer_LfKb · 2025-10-30

**Soundness:** 3
**Presentation:** 3
**Contribution:** 3
**Rating:** 8
**Confidence:** 4

**Summary:**

This paper presents a fully coherent quantum implementation of a multilayer neural network architecture inspired by classical ResNets. The authors design a family of quantum residual blocks that include multi-filter 2D convolutions, sigmoid activations, skip connections, and layer normalization. The key contribution is a fault-tolerant inference algorithm that, under different assumptions about quantum data access (inputs and weights), achieves up to a quartic speed-up over classical inference.

The analysis formalizes the quantum complexity of inference in three settings:
1. No quantum data access (quadratic speed-up).
2. Efficient quantum access to weights (quartic speed-up).
3. Efficient access to both weights and inputs (polylogarithmic scaling in N/ϵ).

The work focuses on theoretical algorithm design and complexity proofs, without empirical quantum-hardware demonstrations.

**Strengths:**

Strong theoretical grounding: The paper is mathematically rigorous and develops clear proofs for each claimed speed-up regime.

Novel construction: Presents, to my knowledge, the first coherent quantum implementation of a multilayer residual network with nonlinear activation and skip connections.

Solid literature placement: The work situates itself appropriately relative to prior quantum neural network and quantum linear-algebra algorithms, clarifying how block-encoding techniques generalize to deep architectures.

Potentially broad impact: The modular framework could be extended to other neural architectures or used as a building block for hybrid quantum–classical learning pipelines.

Clear organization: The text is logically structured, with well-defined assumptions and computational models.

Despite being theoretical, this paper is a well-constructed and rigorous contribution to the quantum machine learning literature. The results are technically sound, clearly presented, and potentially influential for future algorithmic developments in quantum deep learning.
While its immediate practical relevance is limited, the conceptual advancement—showing coherent quantum analogues of residual networks with formal complexity guarantees—justifies acceptance at ICLR, provided the venue continues to welcome high-theory work at the intersection of ML and quantum computation.

**Weaknesses:**

High theoretical abstraction: The paper is highly formal and primarily relevant to researchers in quantum algorithms. It lacks discussion of practical feasibility, especially the resource requirements for realistic network sizes.

Limited accessibility: The work assumes familiarity with block-encoding, and fault-tolerant quantum computation. This makes it difficult for deep-learning practitioners to appreciate its implications or limitations.

No discussion of training: The paper only treats inference and does not address how such quantum architectures could be trained, even conceptually. Since training dominates the computational cost in deep learning, the absence of any consideration of gradients, parameter updates, or trainability significantly narrows the scope and impact of the contribution.

**Questions:**

No empirical validation: While the results are theoretical, even small-scale simulations or resource estimates (for near-term or logical qubit counts) could help contextualize the practical significance of the proposed speed-ups.

Could this framework generalize beyond ResNet-style architectures, for example, to transformers or graph networks?

---

> ### Author Response · Authors · 2025-11-27
>
> We sincerely thank the reviewer for carefully considering our work in detail, and for providing constructive feedback. We additionally would like to thank them for recognizing that "Despite being theoretical, this paper is a well-constructed and rigorous contribution to the quantum machine learning literature. The results are technically sound, clearly presented, and potentially influential for future algorithmic developments in quantum deep learning."
>
> In the following, we will address the questions and weaknesses raised.
>
> 1. *High theoretical abstraction: The paper is highly formal and primarily relevant to researchers in quantum algorithms. It lacks discussion of practical feasibility, especially the resource requirements for realistic network sizes.* Also, *No empirical validation: While the results are theoretical, even small-scale simulations or resource estimates (for near-term or logical qubit counts) could help contextualize the practical significance of the proposed speed-ups.*
>
> With regards to the formality, in our revised version we have made efforts to emphasize the high-level concepts and intuition behind our key results, to increase the general accessibility of the work.
>
> As this is an initial theoretical work, we indeed do not provide resource estimates. We think this is an important question that requires careful consideration, and would be more appropriate for a follow-up self-contained work. With that said, all our proofs are constructive (and give exact circuit descriptions), and it should be straight-forward to convert these circuits into resource estimates. In our paper, we also note some areas (such as the convolution block-encoding) where further optimizations in circuit complexity are straight-forward. In practice, we don't anticipate the constants being that bad.
>
> 2. *Limited accessibility: The work assumes familiarity with block-encoding, and fault-tolerant quantum computation. This makes it difficult for deep-learning practitioners to appreciate its implications or limitations.*
>
> We agree, and in the updated version of the manuscript (which we will soon upload) we have increased the high-level intuition and exposition detailing these important quantum primitives. We also expand the intuition regarding the key aspects governing the complexity of our results (for example, see our high-level sketch of the algorithms in Theorem 2 as presented to Reviewer JGKB). Our goal is to enable general deep-learning practitioners to appreciate the impliciations of our work, while also enabling quantum algorithms experts to directly build on it.
>
> 3. *No discussion of training: The paper only treats inference and does not address how such quantum architectures could be trained, even conceptually. Since training dominates the computational cost in deep learning, the absence of any consideration of gradients, parameter updates, or trainability significantly narrows the scope and impact of the contribution.*
>
> We assume that the architectures we considered are trained entirely classically, and that the only objective of the quantum computer is to accelerate inference. Ideal settings where this would be useful include (as an example) the deployment of large language models, where the inference costs are substantial.
>
> Since our architectures are based on standard classical architectures, and they are trained classically, we don't suspect that they should have any trainability issues. However, an interesting point worth noting that we did not mention in the paper, is that by construction, our residual blocks likely have well-conditioned Jacobians, which may actually help in classical training. However, since this area of research is quite new, and given the scope of our work, exploring this in greater detail is left to subsequent work. We would be happy to make a note mentioning these points in the paper, if the reviewer believes these ideas would be of interest to the community.
>
>
> 4. *Could this framework generalize beyond ResNet-style architectures, for example, to transformers or graph networks?*
>
> The key techniques that we provide should easily generalize to a range of other architectures. The most interesting immediate application would be to expand the results to cover UNet style architectures, as this would enable the model to accelerate inference for the architectures used in many common diffusion models. Moreover, since such diffusion models can be distilled to allow few-step inference, this is precisely the regime where our techniques offer the most advantage (shallow but wide architectures). We think that this is a very promising direction for future investigation, and have updated the manuscript to mention this. It would also be interesting to try applying the novel techniques developed in this work to transformer architectures and graph neural networks, but we have not yet considered this in detail.
>
>
> Once again, we sincerely thank the reviewer for their time and for their input.

---

### Official Review · Reviewer_GrZJ · 2025-10-31

**Soundness:** 4
**Presentation:** 3
**Contribution:** 3
**Rating:** 6
**Confidence:** 4

**Summary:**

This paper presents a theoretical framework for accelerating inference in multi-layer neural networks, specifically those with ResNet-like architectures, using a fault-tolerant quantum computer. The authors introduce a set of quantum subroutines to coherently implement key components of modern deep neural networks, including multi-filter 2D convolutions, non-linear sigmoid activations, skip-connections, and layer normalizations. The work analyzes the inference complexity under three different quantum data access regimes, determined by the availability of Quantum Random Access Memory (QRAM) for inputs and weights. The main results claim a polylogarithmic cost in the input dimension. when QRAM is available for both inputs and weights, and polynomial (quartic and quadratic) speedups over classical methods in regimes with limited or no QRAM access.

**Strengths:**

The paper makes a significant theoretical contribution by providing a quantum algorithm for a highly relevant classical task—deep neural network inference—and offering rigorous proofs of an asymptotic speedup. This directly addresses the core goal of Quantum Machine Learning (QML).

A major achievement of this work is the construction of a fully coherent multi-layer architecture. By designing primitives that avoid intermediate measurements, the algorithm circumvents the readout and re-encoding bottleneck that has plagued previous proposals for deep QNNs.

 The authors' development of the vector-encoding (VE) framework is a key innovation. Critically, this framework enables the implementation of complex operations like 2D multi-filter convolutions without QRAM, directly addressing one of the most significant feasibility concerns surrounding many proposed quantum algorithms.

**Weaknesses:**

The framework presupposes a large-scale, fault-tolerant quantum computer, which is decades away. Therefore, an estimate of the quantum overhead of this algorithm for a specific application is critical for assessing its practical feasibility.

The paper is mathematically dense, making it challenging for a broader audience to grasp its core mechanics. The lack of high-level intuition and illustrative explanations of the core algorithmic mechanics hinders the work's ability to convey its conceptual breakthroughs and limits its overall impact.

Although the model reduces the dimension dependence, this comes at the cost of exponential scaling with respect to the number of layers. A discussion on the trade-off between system dimension and model depth could enhance the paper's theoretical depth.

The paper's analysis is asymptotic, and it lacks numerical simulation to validate the algorithm's behavior or performance characteristics. Some approximation of the model performance will significantly improve this paper.

**Questions:**

The paper's key contribution is reducing the dependency on the input dimension. However, this appears to come at the cost of an exponential scaling with respect to the number of network layers. To enhance the paper's theoretical depth, could the authors provide a dedicated discussion on this critical trade-off?

---

> ### Author Response · Authors · 2025-11-27
>
> We would like to thank the reviewer for taking the time to provide feedback on our work, and for their careful consideration. In the following we will outline our responses to the questions raised, and point out the resulting improvements we have made to the manuscript.
>
>
>
> 1. *The framework presupposes a large-scale, fault-tolerant quantum computer, which is decades away. Therefore, an estimate of the quantum overhead of this algorithm for a specific application is critical for assessing its practical feasibility.*, and *The paper's analysis is asymptotic, and it lacks numerical simulation to validate the algorithm's behavior or performance characteristics. Some approximation of the model performance will significantly improve this paper.*
>
> We completely agree with your assessment that an estimate of the quantum overhead for a specific application is an important consideration, and that a numerically investigation would also be valuable.
> As this is a very new area of research, our goal was to develop a theoretical framework for analysing how quantum computers can help existing deep learning pipelines. We believe that our work provides an essential foundation which will form the basis of such future investigations, which are important enough to investigate as self-contained works.
>
> We do note that our all the proofs in our paper are constructive, and thus provide all relevant circuit descriptions, making subsequent resource estimates straight-forward. We also outline the areas in our paper where numerical investigations could be used to improve complexity, and where the circuits could be further optimised (e.g., the convolution block-encoding could be improved by exploiting the fact that circulant convolutions are diagonalized by the Fourier transform).
>
> 2. *The paper is mathematically dense, making it challenging for a broader audience to grasp its core mechanics. The lack of high-level intuition and illustrative explanations of the core algorithmic mechanics hinders the work's ability to convey its conceptual breakthroughs and limits its overall impact.*
>
> We agree, and taking this feedback into account, we have made a number of clarity improvements to the manuscript. We have provided further intuition and clarifications for Definition 1, and have provided substantially more intuition for the proofs of the algorithms in all 3 regimes. Please see our response outlining the construction of the $U_{conv}$ to Reviewer JGKB as an example of the new intuition that we have added.
>
>
> 3. *Although the model reduces the dimension dependence, this comes at the cost of exponential scaling with respect to the number of layers. A discussion on the trade-off between system dimension and model depth could enhance the paper's theoretical depth.* Also, *The paper's key contribution is reducing the dependency on the input dimension. However, this appears to come at the cost of an exponential scaling with respect to the number of network layers. To enhance the paper's theoretical depth, could the authors provide a dedicated discussion on this critical trade-off?*
>
>
> We agree that this is a very important question which needs the appropriate consideration. We have added a "Future Work" section where we outline some thoughts on this topic. In particular, yes the exponential increase in complexity with respect to the number of non-linear activation function layers is the biggest limitation of our technique. However, we note that there are a number of popular architectures which are wide and shallow architectures (e.g., UNet based architectures, as used by Stable Diffusion) which fit exactly the paradigm our techniques excel at. We also note that there are potentially a number of ways to mitigate the exponential increase in cost with depth. For instance, a sequence of layers could be performed coherently, project onto a lower dimension subspace, perform tomography, and reinitialize the state reseting the depth cost (this would be a hybrid design between our techniques and the prior work, potentially getting the benefits of both designs). Additionally, one could explore enacting the non-linear transformations with a QPE based approach, where the circuit complexity may not compound exponentially with depth, at the cost of an exponentially increasing error (which numerically studies may show is not an issue for neural networks in practice, similarly to how numerical error doesn't usually compound exponentially with depth). However, as this is a new area of research, and our work provides the foundation for subsequent exploration of these ideas, we leave this as future research. We now sketch these arguments in "Future Work" section of the revision to the paper we will soon upload.
>
> Please let us know if there are any further clarifications that would be of help.

---

> > ### Comment · Reviewer_GrZJ · 2025-11-28
> >
> > I thank the authors for their response. However, my primary concerns regarding the numerical evidence have not been sufficiently addressed. Therefore, I am maintaining my current score.

---

> > > ### Author Response · Authors · 2025-12-03
> > >
> > > We would like to thank you for your time in considering our response, and for reviewing our paper.
> > >
> > > All the best,
> > > The authors

---

### Author Response · Authors · 2025-12-03

Dear AC(s),

The following message is intended to aid you in making a decision regarding our paper. We summarize the key strengths and weaknesses raised by each reviewer, our responses, and the score changes each reviewer indicated they intended to make.

In summary, prior to our rebuttal our scores were 8, 6, 6, 2. After rebuttal, our scores were indicated to be 8, 8, 6, 2, while the reviewer who gave a 2 indicated they were considering raising their score before further discussion was locked.

We will now post a summary for each reviewer.

Thank you for your time and consideration.

---

> ### Author Response · Authors · 2025-12-03
> **Reviewer GrZJ**
>
> ### **Reviewer GrZJ** Initial score: 6, Update Indicated: Score Unchanged
>
> **Strengths Highlighted:**
>
> - Recognizes a "significant theoretical contribution", which "directly addresses the core goal of Quantum Machine Learning (QML)"
>
> - Emphasizes the importance of our *fully coherent multi-layer architecture*
>
> - Notes the importance of the techniques we develop (the vector-encoding framework, a QRAM-free convolutional block-encoding, etc) which "directly addressing one of the most significant feasibility concerns surrounding many proposed quantum algorithms."
>
>
> **Main Concerns and Limitations:**
>
> - Lack of numerical simulations
>
> - Requested resource estimates
>
> - Accessibility and clarity: notes that the paper is mathematically dense and would benefit from more high-level intuition
>
> - Depth vs dimension tradeoff: requests further discussion of how this tradeoff impacts our complexity
>
>
> **Post-Rebuttal Status:**
>
> - We clarified that all our proofs are constructive, making a resource estimate a straightforward contribution for followup work
>
> - We added more high-level intuition to the paper, helping to explain our key results, and we added a "Future Work" section which expands on the depth vs width tradeoff and potential future improvements
>
> - **Reviewer's followup comment:** Thanked us for our response. Stated they planned to maintain their current score of borderline accept (6), as we did not sufficiently address their primary concern regarding the numerical simulations (which we believe are more appropriate for follow-up work, given the depth of our theoretical contributions)

---

> ### Author Response · Authors · 2025-12-03
> **Reviewer LfKb**
>
> ### **Reviewer LfKb** Initial score: 8, No Score Update Indicated
>
> **Strengths Highlighted:**
>
> - Notes strong theoretical grounding, with rigorous and clear proofs
>
> - Notes novelty, stating: "Presents, to my knowledge, the first coherent quantum implementation of a multilayer residual network with nonlinear activation and skip connections."
>
> - Notes a potentially broad impact (states, "potentially influential for future algorithmic developments in quantum deep learning"), and notes clear organization of the paper
>
> **Main Concerns and Limitations:**
>
> - Notes a high level of formality, and that we do not discuss practical feasibility (e.g., resource estimates)
>
> - Notes limited accessibility, and that we assume familiarity with fault-tolerant quantum computation
>
> - Notes that we only consider accelerating inference, and do not discuss training
>
> **Post-Rebuttal Status:**
>
> - We emphasised that this is an initial theory paper whose constructive proofs and explicit circuits are designed to make subsequent resource-estimation work straightforward, and we clarified where circuit constants could be improved
>
> - We made a number of changes to the manuscript (and detailed them in our rebuttals) increasing the general accessibility of our work, increasing intuition for the key proofs and key primitives used
>
> - We clarified that our focus is on classically trained networks with quantum-accelerated inference (e.g., for LLM or diffusion-model deployment), and sketched how our residual blocks are likely to have well-conditioned Jacobians that may actually be helpful during classical training
>
> - The reviewer did not post a further update, but **already recommends acceptance with a score of 8**

---

> ### Author Response · Authors · 2025-12-03
> **Reviewer yibn**
>
> ### **Reviewer yibn** Initial score: 6, New score indicated: 8 (accept)
>
> **Strengths Highlighted:**
>
> - Views the paper as providing "important advances" over prior work
>
> - Notes the architectures considered are practically relevant (ResNet)
>
> - Notes "the paper is mostly very well-written and clearly presented (except for some ambiguous mathematical statements)"; we have since corrected the ambiguous statements
>
> **Main Concerns and Limitations:**
>
> - They question our selection of the $\rm erf$ activation function claiming it is an uncommon activation
>
> - Point out our input-dependent circuits: they note that the unitary implementing each network appears to depend on the input $x$, raising questions about whether a new circuit must be constructed for each data point and how this impacts complexity
>
> - Clarity of mathematical statements: some ambiguity in Definition 1 and Theorem 2
>
> - Claim we have unclear assumptions on the inputs to our network
>
>
> **Post-Rebuttal Status:**
>
> - Their primary technical questions were directly addressed by providing the following clarifications:
>     - We are not limited to $\rm erf$ activation, common activations such as $\tanh$ and GELU also immediately work with our techniques
>     - Our input-dependent circuit definitions are a key part of our algorithm enabling non-linearities and we already carefully analyse the complexity this incurs
>     - We have no assumptions on our network inputs (any input works), other than it must be $\ell_2$-normalized
>
> - Their other concerns regarding clarity were directly addressed and incorporated in the revised paper
>
> - In their follow-up comment, the reviewer **explicitly states that we addressed all their concerns** and that **"I will increase my rating to recommend acceptance"**, i.e. moving the score from 6 to 8 (since only even scores are allowed).

---

> ### Author Response · Authors · 2025-12-03
> **Reviewer JGKB**
>
> ### **Reviewer JGKB** Initial score: 2, Update Indicated: Considering Increase (prior to discussion being closed)
>
> **Strengths Highlighted:**
>
> - Acknowledge that we "address one of the most interesting topics in the field of quantum computation"
>
> - Appreciate the clear positioning of our work over the prior literature
>
> **Main Concerns and Limitations:**
>
> - Percieved suitability / writing clarity: feels too many important definitions/ results are delegated to the appendix
>
> - QRAM assumptions:
>     - Views the QRAM assumption as being very strong, claiming that an $n$-bit QRAM oracle could take up to $2^n$ depth to implement
>     - Raises dequantization questions in the regimes with QRAM assumptions
>     - Questions the value of our results in the regimes where QRAM is used
>
> - QRAM-free derivations:
>     - Considers the QRAM-free regime as a potentially valuable contribution, but reports difficulty in following our derivations for this regime
>     - Asks for a clear derivation and explicit statement of all assumptions
>
> **Post-Rebuttal Status and Response:**
>
> - **On writing and structure:** We explained that the organisation (high-level narrative and main theorems in the primary text; long proofs and auxiliary lemmas in the appendix) follows common practice for theory papers at ICLR. We also pointed out that other reviewers rated the presentation as "good", while nonetheless agreeing to add more high-level explanations in the main text
>
> - **On QRAM assumptions and dequantization**:
>     - We emphasised that our QRAM usage follows standard constructions and analysed realistic implementation constraints in Appendix D, arguing that large-scale QRAM would most plausibly be realised as a passive, noisy device coupled to an error-corrected processor, rather than in the quantum circuit model
>     - We clarified the reviewer's claim that QRAM may require $2^n$ depth (in reality, QRAM can be implemented with ${\rm poly}(n)$ depth with standard techniques) -- we also point them to our discussion in Appendix D where we point out the challenges associated with QRAM beyond what the reviewer raised
>     - We pointed them to our comprehensive discussion of QRAM feasibility in Appendix D, and stress that we consider our QRAM assumptions with substantial rigour
>     - We compared our architectures to known dequantization techniques in Appendix E.1, highlighting that existing dequantization results require low-rank or sparse operators, and thus that our dense, full-rank mat–vec–square block cannot be dequantized with existing techniques. Moreover, we sketched why a reduction to Fourier sampling may even preclude such a dequantization (although this is left as future work)
>
> - **On the QRAM-free derivation:**
>     - We provided a detailed intuitive derivation in two follow-up comments ("Additional High-Level Intuition and Derivation of Main Architectural Results" Parts 1 and 2), explaining how Lemma 5 (QRAM-free convolution block-encoding) and Lemmas 6–7 (stacked residual blocks) combine. We have incorporated a version of this high-level intuition to the revised version of the paper
>     - We explained the intuition behind how our convolutional block-encodings are obtained with polylogarithmic dimension dependence *without QRAM*
>     - We now list all assumptions explicitly in the relevant lemmas/ theorems/ definitions
>
>
> - In their follow-up comment, the reviewer wrote "I thank the authors for their effort in clarifying the proofs and revising the manuscript", and that "**I will consider to increase the score.**" The ability for reviewers to change their score was locked shortly after, but their comment indicates that our clarifications softened their view.

---

### Meta-Review · Area_Chair_Mv1i · 2025-12-10

**Summary:**

The paper makes a technically strong and novel contribution by giving the first fully coherent quantum implementation of multilayer ResNet-style networks with provable speedups under several data-access regimes. Remaining limitations, lack of simulations/resource estimates and depth scaling, are clearly acknowledged and are reasonable for a foundational theory paper.

**Reviewer Concerns:**

The rebuttal successfully addressed the concerns raised by Reviewers yibn and LfKb regarding unclear mathematical statements, assumptions on inputs, the role of activation functions, and the dependence of circuits on the input; both reviewers indicated acceptance. The authors also provided substantially improved intuition for the QRAM-free regime, clarifying Lemma 5, the residual-block construction, and the depth bounds, which partially addressed Reviewer JGKB’s technical clarity concerns.

The outstanding issues are primarily those raised by Reviewer GrZJ, namely, the absence of numerical simulations and concrete resource estimates, and the broader QRAM practicality concerns from Reviewer JGKB, which the rebuttal contextualized but did not fully resolve.

**Reviewer Scores:**

2 reviewers have already indicated increasing the score.

---

### Decision · Program_Chairs · 2026-01-26

Accept (Poster)